# Composable Sparse Subnetworks via Maximum-Entropy Principle

**Francesco Caso** [*,1,2], **Samuele Fonio**[*,3], **Simone Monaco**[*,4], **Nicola Saccomanno**[*,5],
**Fabrizio Silvestri**[1]
[1]Sapienza University of Rome, [2]University of Cambridge, [3]University of Turin,
[4]Politecnico di Torino, [5]University of Udine

## Abstract

Neural networks implicitly learn class-specific functional modules. In this work, we ask: Can such modules be isolated and recombined? We introduce a method for training sparse networks that accurately classify only a designated subset of classes while remaining deliberately uncertain on all others, functioning as class-specific subnetworks. A novel KL-divergence-based loss trains only the functional module for the assigned set, and an iterative magnitude pruning procedure removes irrelevant weights. Across multiple datasets (MNIST, FMNIST, CIFAR-10, CIFAR-100, tabular and text classification data) and architectures (MLPs, CNNs, ResNet, VGG), we show that these subnetworks achieve high accuracy on their target classes with minimal leakage to others. When combined via weight summation or logit averaging, these specialized subnetworks act as functional modules of a composite model that often recovers generalist performance. For simpler models and datasets, we experimentally confirm that the resulting modules are mode-connected, which justifies summing their weights. Our approach offers a new pathway toward building modular, composable deep networks with interpretable functional structure.

## 1 Introduction

Modern neural networks (NNs) implicitly develop internal subgraphs of neurons and connections tuned to respond to specific classes. These structures, sometimes referred to as *circuits* (Olah et al., 2020; O'Neill and Bui, 2024), emerge during training but are difficult to isolate, reuse, or compose. Representations for different classes are often entangled, resulting in shared neurons or features—a phenomenon known as *superposition* (Mu and Andreas, 2020; Saphra and Wiegreffe, 2024)—which makes clean modularity elusive.

This lack of class-level modularity limits our ability to understand, edit, or compose networks. If we could reliably extract a *functional module*—a subnetwork that specializes in recognizing one class (or a small subset) while ignoring others—we could provide new tools both for modular NNs and for mechanistic interpretability. Crucially, such modules must not only function in isolation, but also be designed to compose smoothly, without fine-tuning or alignment (Ilharco et al., 2023; Hazimeh et al., 2024). In this work, we want to answer the following question:

> Can we train sparse, class-specialized subnetworks that remain ignorant outside their domain, and compose into accurate, generalist models?

We answer affirmatively, proposing a methodology that leverages two key concepts: the Maximum-Entropy (MaxEnt) principle and sparse training. The MaxEnt principle (Jaynes, 1957) advocates choosing the most uninformative distribution consistent with known constraints. Originally applied in statistical mechanics, it has become central to inference theory (Kuić, 2016; De Martino and De Martino, 2018).

We claim that MaxEnt can guide functional isolation: modules are trained to make confident predictions only for their class, and uniform predictions otherwise. This differs from standard entropy regularization used in calibration or selective prediction (Marczak et al., 2024; Pereyra et al., 2017).

---

[*]Alphabetical order. See Author Contributions section for more details. Email: fc580@cam.ac.uk

On the other hand, sparsity plays a crucial role in this context. Through pruning, the network discards weights that are irrelevant to the task, possibly mitigating interference during weight-space merging. We demonstrate its practical benefits by leveraging the Lottery Ticket Hypothesis (LTH). In general, LTH claims that sparse subnetworks can match dense models if trained with the right initialization (Frankle and Carbin, 2018; Zhou et al., 2019; Liu et al., 2024). In our scenario, pruning is essential for exposing the circuit and sparse subnetworks show improved modular compatibility.

In light of this, we propose a framework based on two main components: a novel **Maximum-Entropy loss** (ME) coupled with an **Iterative Magnitude Pruning** (IMP) strategy (Frankle and Carbin, 2018), and a model merging step to achieve generalist models. The goal of the ME is to enforce confident predictions on a subnetwork's rewarded class set and uniform predictions otherwise, while the IMP removes spurious capacity and reinforces specialization through sparsity (Burkholz, 2022; Girish et al., 2021). We call the resulting components *subnetwork modules*: sparse networks that specialize in a class or class subset, and can be combined to form more generalist models.

To the best of our knowledge, this is the first work that trains a NN by isolation and merging. Our results suggest a new approach to NN construction: rather than learning entangled solutions end-to-end, we can build systems from independently trained, entropy-regularized modules. To summarize, our contribution is threefold: *(i)* we introduce a novel MaxEnt loss able to train specialized modules; *(ii)* we propose a new training pipeline to achieve better isolation and model merging to get generalist models; *(iii)* we validate our findings through extensive experiments on several architectures and datasets.

## 2 RELATED WORK

In this section, due to lack of space, we put only the most relevant related work. An extended version of related work involving model merging, pruning methods, modular training and MaxEnt principle is available in Appendix A.

**Modular Neural Networks.** Several works have explored modular neural architectures that decompose computation across tasks (Kirsch et al., 2018; Han et al., 2021; Salem et al., 2023). Notably, Kirsch et al. (2018) introduced end-to-end learning of both modules and their composition via a controller. They rely on subsequent modules for learning broader tasks, while using expectation-maximization for learning a specific module. While sharing a similar motivation to our work, we build *class-specific functional modules*, trained to focus on a single class while producing high-entropy on different classes. This specific goal is hindered by their structure of subsequent modules, that are chosen by a controller and reused across tasks, making it difficult to isolate class-specific functionality. Additionally, our approach relies on model merging, rather than leveraging subsequent modules. Interestingly, Malakarjun Patil et al. (2023) highlight the importance of pruning methods in unveiling the hierarchical structure of NNs. In particular, they propose an iterative approach for hierarchy detection, where the hierarchy is the underlying hierarchy of sub-functions in a specific task, delivering interesting network analysis. Our work partially shares the goal, but focuses more on a complete training pipeline and on submodules isolation without seeking a hierarchical structure, rather than network analysis about submodules.

**Interpretability.** Our work inevitably aligns with the goal of producing interpretable networks. Among the methods that tackle this challenge, we can distinguish between post-hoc and interpretable-by-design methods. While post-hoc interpretability methods (Ribeiro et al., 2016; Lundberg and Lee, 2017) are constrained to work with trained models, our methodology is applied *during training*, yielding units with clear semantics and compositional behavior. In particular, our approach aligns with mechanistic interpretability methods (Saphra and Wiegreffe, 2024; Olah et al., 2020), where subnetworks (or circuits) are studied as functional entities. Interestingly, Liu et al. (2023) propose a brain-inspired modular training procedure, relying on network embedding into geometric spaces, and neurons swapping. Instead of leveraging pruning methodologies, they rely mainly on regularization. However, they mainly use synthetic datasets and never scale to CNN, ResNet, or VGG, while we show that our technique is applicable also to a broader range of tasks.
Finally, our work also complements neuron-level analyses like Mu and Andreas (2020), which characterize the concepts encoded by individual neurons from a compositional point of view. Rather than probing what individual neurons encode post-hoc, we directly train subnetworks to be functionally specialized by construction, obtaining modular units with unambiguous class semantics. Our work is also conceptually related to Marchetti et al. (2024), where symmetry constraints shape functional structure—our modules may reflect class-wise symmetry components in such a framework.

# 3 LEARNING COMPOSABLE CLASS-SPECIFIC SUBNETWORK MODULES

We propose a method for constructing composable subnetwork modules, each specialized in classifying a specific subset of classes while remaining unresponsive to others. These modules are trained to produce peaked predictions on their target classes and uniform predictions otherwise, in line with the MaxEnt principle. Our proposed procedure consists of two key elements that we are going to discuss: a custom loss function and an IMP schedule. Our final goal is to merge trained modules efficiently, resulting in a more powerful model, built by simple merging. Figure 1 provides an high level overview of our method and its components.

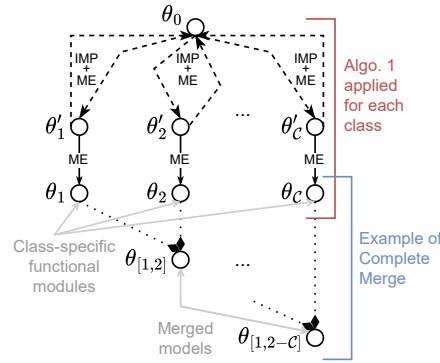

## 3.1 MAXIMUM-ENTROPY LOSS

Let $\mathcal{C}$ be the full set of classes and $\mathcal{R} \subseteq \mathcal{C}$ the set of *rewarded classes* for a given subnetwork module. For a training sample $(x, y)$, where $y \in \mathcal{C}$, we define a *target distribution* $\tilde{y} \in \mathbb{R}^{|\mathcal{C}|}$ as:

Figure 1: Overview of our method. Starting from common initialization $\theta_0$ we find a specialized subnetwork for each class in $\mathcal{C}$, leveraging IMP and our ME loss (see Algorithm 1).

$$\tilde{y}_i = \begin{cases} \delta_{i=y} & \text{if } y \in \mathcal{R} \\ \frac{1}{|\mathcal{C}|} & \text{otherwise} \end{cases}. \tag{1}$$

For example, if $\mathcal{C} = \{0, 1, 2\}$ and $\mathcal{R} = \{0\}$ we will use $\tilde{y} = (1, 0, 0)$ for class 0, $\tilde{y} = (0.33, 0.33, 0.33)$ for classes 1 and 2. Let $\hat{y} = \text{softmax}(f_\theta(x))$ be the predicted class probability distribution produced by the model. The *MaxEnt loss* (ME) is then defined as the Kullback-Leibler (KL) divergence between the target distribution $\tilde{y}$ and the predicted distribution $\hat{y}$:

$$\mathcal{L}_{\text{ME}}(x, y) = \text{KL}(\tilde{y} \parallel \hat{y}) = \sum_{i=1}^{|\mathcal{C}|} \tilde{y}_i \log \left( \frac{\tilde{y}_i}{\hat{y}_i} \right). \tag{2}$$

This formulation encourages the model to output *peaked predictions* (low entropy) for samples in rewarded classes and *uniform predictions* (high entropy) for non-rewarded ones. In doing so, it promotes *functional isolation*, ensuring that each subnetwork module specializes only in its intended class subset and remains maximally uncertain elsewhere.

Unlike one-vs-all formulations, our approach enforces *neuron-permutation invariance* on non-rewarded samples—a crucial property for composability. In a one-vs-all setup, training a module for class 0 would implicitly push other output neurons (e.g., neuron 1) to represent "not class 0." However, if another module uses neuron 1 to represent "class 1," summing the two leads to a semantic collision: the same neuron would encode both "class 1" and "not class 0", which includes many other classes.

## 3.2 ITERATIVE MAGNITUDE PRUNING

To enhance specialization and remove spurious capacity, we apply **Iterative Magnitude Pruning** (IMP) (Frankle and Carbin, 2018) as in Algorithm 1. At each iteration, we train with ME, prune a fixed percentage of low-magnitude weights, and reinitialize the remaining ones. After several rounds, a final training phase is applied to the resulting pruned subnetwork. This process encourages class-specific specialization.

---

**Algorithm 1** Iterative Magnitude Pruning with Maximum-Entropy Loss

---

**Require:** Initial weights $\theta_0$, training data $\mathcal{D}$, rewarded classes $\mathcal{R}$, number of pruning iterations $N$, pruning percentage $P$, epochs per iteration $E$
1: $\theta \leftarrow \theta_0$
2: **for** $i = 1$ to $N$ **do**
3:     Train model $f_\theta$ on $\mathcal{D}$ using maximum-entropy loss with rewarded classes $\mathcal{R}$ for $E$ epochs
4:     Prune $K\%$ of weights in $\theta$ with smallest absolute value, with $K = 1 - (1 - P)^{\frac{1}{N}}$
5:     Reset remaining weights in $\theta$ to their initial values from $\theta_0$
6: **end for**
7: Train the final pruned subnetwork on $\mathcal{D}$ using maximum-entropy loss with rewarded classes $\mathcal{R}$ for $E$ epochs

---

### 3.3    MODEL MERGING

We investigate two strategies for composing a generalist model from class-specific submodules identified by ME training and eventually pruned through IMP. These strategies operate in *weight-space* and *logit-space*, respectively, and reflect two distinct regimes revealed by our experiments.

#### 3.3.1    VIA WEIGHT SUMMATION

The first approach consists of composing a generalist model by summing submodules' weights: $\theta_{\text{merged}} = \sum_i \theta_i$. This operation is enabled by our design: submodules are trained to specialize on disjoint subsets of classes and to behave identically—via uniform predictions—on all others, minimizing interference.

We consider both pairwise merges ($\theta_1 + \theta_2$) and complete merge ($\sum_i \theta_i$), and evaluate compositionality using the same metrics used for individual modules. To understand when and why summation preserves performance, we analyze the loss landscape through the lens of mode connectivity. Crucially, unlike typical mode connectivity studies, where $\theta_1$ and $\theta_2$ solve the *same task*, our submodules are specialized for *different sets of classes*. The merged model is intended to solve the union task, and is evaluated using the *MaxEnt loss* over the combined rewarded classes. This distinction makes it essential to assess whether the merged weights lie in a low-loss region for the composite task.

Following Frankle et al. (2020) and Lubana et al. (2023), we say that $\theta_1$ and $\theta_2$ are *mode connected* along a path $\gamma(t)$ if:

$$\forall t \in [0, 1], \quad \mathcal{L}(f_{\gamma(t)}(\mathcal{D})) \leq ((1 - t) \cdot \mathcal{L}(f_{\theta_1}(\mathcal{D})) + t \cdot \mathcal{L}(f_{\theta_2}(\mathcal{D})) + \epsilon), \tag{3}$$

where $\epsilon$ is a small margin, set to $2\%$ of the first term on the r.h.s. following Frankle et al. (2020), and $\mathcal{L}$ is our ME loss evaluated on a dataset $\mathcal{D}$ with labels restricted to $\mathcal{R}_1 \cup \mathcal{R}_2$, the union of the rewarded classes for the two modules.

To test this, we define the following *piecewise-linear* path, designed such that $\theta_1 + \theta_2$ appears as an intermediate point:

$$\gamma_{\theta_1 \rightarrow \theta_2}(t) = \begin{cases} \theta_1 + 2t \cdot \theta_2 & \text{if } t \leq 0.5 \\ 2(1 - t) \cdot \theta_1 + \theta_2 & \text{if } t > 0.5 \end{cases}. \tag{4}$$

This path satisfies: $\gamma(0) = \theta_1$, $\gamma(1) = \theta_2$, and $\gamma(0.5) = \theta_1 + \theta_2$. We evaluate the loss along $\gamma(t)$ and interpret low-barrier profiles as evidence that the modules are composable by construction.

#### 3.3.2    VIA LOGIT AVERAGING

In addition to weight summation, we consider a second compositional strategy, i.e., merging modules directly in *logit-space*. Let $f_{\theta_i}$ be a submodule trained with ME on rewarded classes $\mathcal{R}_i \subseteq \mathcal{C}$. For an input $x$, define its logits as $z^{(i)} = f_{\theta_i}(x) \in \mathbb{R}^{|\mathcal{C}|}$, and thus $\hat{y}^{(i)} = \text{softmax}(z^{(i)})$. By construction, ME encourages each $f_{\theta_i}$ to output a peaked distribution on samples in $\mathcal{R}_i$ and a uniform distribution on $\mathcal{C} \backslash \mathcal{R}_i$. Hence, each submodule behaves as a specialist expert, informative only on its rewarded set. We merge $N$ such experts through a convex combination of their logits (specifically, their average),

$$\bar{z} = \sum_{i=1}^{N} w_i \, z^{(i)}, \qquad w_i \geq 0, \sum_i w_i = 1, \tag{5}$$

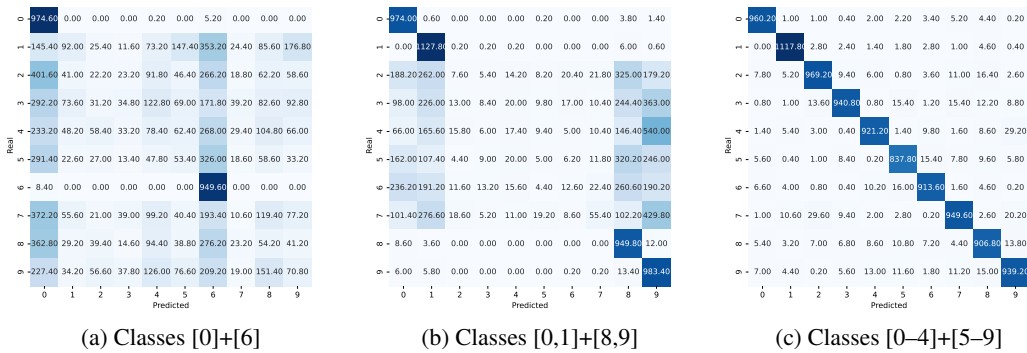

(a) Classes [0]+[6]  (b) Classes [0,1]+[8,9]  (c) Classes [0–4]+[5–9]

Figure 2: Confusion matrices (total predicted counts per class) for three representative pairwise merges in weight-space of shallow MLP submodules on MNIST. Even after merging, each rewarded class remains highly separable, and predictions on non-rewarded classes remain nearly uniform.

and predict $\bar{y} = \mathrm{softmax}(\bar{z})$. This operation has a direct probabilistic interpretation: since softmax is an exponential normalization, Eq. 5 is equivalent to a *logarithmic opinion pool* (Genest and Zidek, 1986), i.e., a product-of-experts aggregation (Hinton, 2002),

$$\bar{y}_k = \mathrm{softmax}(\bar{z})_k \propto \exp \sum_{i=1}^{N} w_i\, z_k^{(i)} = \prod_{i=1}^{N} \exp(z_k^{(i)})^{w_i} \propto \prod_{i=1}^{N} \mathrm{softmax}(z^{(i)})_k^{w_i}. \qquad (6)$$

Under ME, any expert that is non-rewarded for $x$ outputs nearly uniform $\hat{y}^{(i)}$, which contributes only a class-independent factor in Eq.6 and thus cancels after normalization. Therefore, logit averaging automatically ignores irrelevant experts and preserves only the specialists that match $x$, explaining the strong union-task performance observed in our experimental evaluation.

## 4 EXPERIMENTS

We empirically evaluate our method on multiple datasets, aiming to answer the following questions: (i) Can ME training enforce meaningful class-specific specialization? (ii) How well do merged submodules recover generalist performance? (iii) Does iterative pruning improve the quality of submodules and composability? (iv) Does our training procedure induce mode connectivity between independently trained modules and does it scale with the number of modules?

### 4.1 TRAINING AND EVALUATION PROTOCOL

We apply our ME and IMP independently to each class or subset $\mathcal{R} \subseteq \mathcal{C}$. Each submodule is trained as in Algorithm 1, and evaluated on: (i) **rewarded accuracy**, i.e., classification accuracy on samples from $\mathcal{R}$; (ii) **non-rewarded entropy**, i.e., the average predictive entropy on inputs from $\mathcal{C} \setminus \mathcal{R}$; and (iii) the **confusion matrix**, used qualitatively to assess specialization and leakage. In particular, it is worth mentioning that in our training procedure we simulate a scenario in which only the classes in $\mathcal{R}$ are labeled, but we still have access to the other samples. Formal definitions and implementation details are provided in Appendix B.

To evaluate composability, we used $|\mathcal{R}| \in \{1, 2, 5\}$ to test pairwise merging. For a specific cardinality, the model is trained to recognize solely a selected set of classes, of the given cardinality and sampled among the ones available for that specific dataset. We sample 10 pairs of class sets $\{(\mathcal{R}_{2k}, \mathcal{R}_{2k+1})\}_{k=0}^{9}$, with $\mathcal{R}_{2k} \bigcap \mathcal{R}_{2k+1} = \emptyset$ and $\mathcal{R}_{2k}, \mathcal{R}_{2k+1}$ spanning in the set of combinations for the dataset's class set. For each pair, we perform 5 independent random seeds and merge a pair into a single model. For example, with $|\mathcal{R}| = 1$, the procedure consists of: training a NN using our ME rewarding one specific class $\mathcal{R}_0$, then a different NN is trained on a different single class $\mathcal{R}_1$, and then the two models are merged and tested. The resulting model should perform well on the 2 classes. This procedure is repeated for 5 seeds and 10 paired sets to ensure statistical evidence. When $|\mathcal{R}| = n > 1$, two NNs are trained to specialize on two different sets of $n$ classes at the same time, and there is no overlap between the sets. Figure 2 depicts an example of resulting confusion matrices for these pairwise-merging experiments.

Table 1: Rewarded accuracy and non-rewarded entropy of sub-modules obtained using ME with and without IMP. Results are averaged over 5 runs and across classes. Number of classes for dataset: 10 for MNIST and FMNIST, 6 for HAR and a balanced subset of 4 for Yeast. Accuracy is high on rewarded classes and predictions are nearly uniform on excluded ones.

| Model | IMP | MNIST | | FMNIST | | HAR | | Yeast | |
|---|---|---|---|---|---|---|---|---|---|
| | | Entropy | Rewarded Acc | Entropy | Rewarded Acc | Entropy | Rewarded Acc | Entropy | Rewarded Acc |
| Shallow MLP | No | 2.296 (0.003) | 0.998 (0.002) | 2.296 (0.003) | 0.998 (0.002) | 1.762 (0.017) | 0.997 (0.007) | 1.298 (0.062) | 0.995 (0.009) |
| | Yes | 2.293 (0.004) | 0.999 (0.001) | 2.293 (0.004) | 0.999 (0.001) | 1.757 (0.023) | 0.996 (0.008) | 1.297 (0.059) | 0.998 (0.006) |
| Deep MLP | No | 2.298 (0.002) | 0.997 (0.003) | 2.285 (0.013) | 0.995 (0.004) | 1.772 (0.014) | 0.992 (0.013) | 1.302 (0.064) | 0.996 (0.009) |
| | Yes | 2.300 (0.001) | 0.998 (0.002) | 2.291 (0.008) | 0.991 (0.007) | 1.762 (0.023) | 0.999 (0.005) | 1.302 (0.056) | 1.000 (0.000) |
| CNN | No | 2.302 (0.000) | 0.998 (0.004) | 2.302 (0.000) | 0.996 (0.004) | - | - | - | - |
| | Yes | 2.302 (0.000) | 0.994 (0.005) | 2.302 (0.000) | 0.992 (0.005) | - | - | - | - |

## 4.2 EXPERIMENTAL SETUP

We evaluate our approach on four model classes: MLPs (denoted as shallow for 1 hidden layer and as deep for 2 hidden layers), CNNs (LeNet-style), ResNet18, and VGG11 (with or without Batch-Norm). For image data, we use MNIST (train/test 60K/10K ) (LeCun et al., 1998), Fashion MNIST (FMNIST, 60K/10K) (Xiao et al., 2017), and CIFAR-10 (60K/10K) (Krizhevsky et al., 2009), with 10 classes for each dataset. These datasets are in line with similar works (Liu et al., 2023; Malakar-jun Patil et al., 2023), but to get a broader impact we include two tabular classification datasets from the UCI repository: *Human Activity Recognition* (HAR, 7K/3K) (Reyes-Ortiz et al., 2012) (6 classes), and *Yeast* (1K/250) (Dua and Graff, 2017) (a balanced subset of 4 classes). For each model-dataset pair we train the modules both with and without IMP and merge them both in weight-space and logit-space.

We compare our ME mainly with two baselines, showing that entropy maximization plays a crucial role in isolation. In particular, we compare ME with: Quasi-MaxEnt loss (QME), a loss function that on non-rewarded classes ignores the rewarded classes and produces a uniform distribution solely on non-rewarded classes; and CrossEntropy loss (XE), the standard classification loss, where labels are not modified but the model is trained only on the classes in $\mathcal{R}$. We additionally consider XE with label-smoothing ($\lambda = 0.1$), which replaces the one-hot target distribution with a smoothed version; and XE with confidence-penalty which adds an entropy-regularization term. For the formal definitions, we refer to Appendix B. Due to lack of space and underperforming behavior, results related to label-smoothing and confidence penalty are shown in Table 9, Appendix C. Full dataset, preprocessing and experimental details are reported in Appendix B. Extensive and detailed experimental results are provided in Appendix C. Next we describe the main outcomes.

## 4.3 SUBMODULE BEHAVIOR WITH MAXENT LOSS

We begin by evaluating submodules trained solely with ME, *without pruning*. Each subnetwork is tasked with recognizing one specific class from the original dataset, and to output uniform predictions on all others. For each of 5 random seeds, we train a separate submodule per class. The results in Table 1 are aggregated across both classes and seeds, providing average values and standard deviations for overall trends.

Table 1 reports the accuracy on rewarded classes and the average entropy on non-rewarded samples. Accuracy is consistently high (close to $100\%$) across all classes, confirming that submodules correctly specialize on their class. The entropy, meanwhile, approaches the theoretical value of $\log(N_{\text{classes}})$ for a uniform distribution, e.g. $\log(10) \approx 2.30$ for MNIST and FMNIST, suggesting that predictions on excluded classes are nearly uniform. These results suggest that our proposed submodules can effectively recognize individual classes and exhibit the theoretical behavior they were designed to emulate, across different architectures.

## 4.4 PAIRWISE MERGING

We further evaluate the effectiveness of our method through pairwise merging experiments across different configurations. In Figure 3, we show the effectiveness of IMP on MNIST and FMNIST on three architectures (Shallow MLP, Deep MLP, and CNN) and all cardinalities, while merging in weight-space. We can see in particular that IMP improves the performances consistently for a Deep MLP, while benefiting more modestly the Shallow MLP and the CNN.

Table 2: Average rewarded accuracy of pairwise-merged sub-modules, merging either via logit averaging or weight summation, across architectures, datasets, cardinalities $|\mathcal{R}|$, and loss functions: CrossEntropy (XE), Quasi-MaxEnt (QME), and MaxEnt (ME), with IMP applied. Results are averaged over 5 seeds and 10 pairs. For each group model-cardinality-dataset, the best approach is highlighted in bold. ME consistently achieves the highest rewarded accuracy.

| Model | $|\mathcal{R}|$ | Loss | FMNIST Logit | FMNIST Weight | MNIST Logit | MNIST Weight | HAR Logit | HAR Weight | Yeast Logit | Yeast Weight |
|---|---|---|---|---|---|---|---|---|---|---|
| Shallow MLP | 1 | XE | 0.859 (0.181) | 0.813 (0.209) | 0.798 (0.162) | 0.679 (0.188) | 0.882 (0.164) | 0.866 (0.181) | 0.667 (0.100) | 0.660 (0.094) |
| | | QME | 0.962 (0.035) | 0.868 (0.147) | 0.984 (0.006) | 0.973 (0.011) | 0.955 (0.023) | 0.954 (0.041) | 0.734 (0.127) | 0.737 (0.132) |
| | | ME | **0.983** (0.029) | **0.980** (0.031) | **0.992** (0.004) | **0.991** (0.005) | **0.982** (0.026) | **0.983** (0.026) | **0.844** (0.115) | **0.843** (0.116) |
| | 2 | XE | 0.813 (0.093) | 0.777 (0.111) | 0.852 (0.070) | 0.831 (0.067) | 0.870 (0.068) | 0.864 (0.059) | 0.439 (0.056) | 0.419 (0.039) |
| | | QME | 0.953 (0.021) | 0.918 (0.043) | 0.982 (0.006) | 0.963 (0.015) | 0.937 (0.010) | 0.919 (0.020) | 0.474 (0.018) | 0.475 (0.021) |
| | | ME | **0.960** (0.023) | **0.952** (0.025) | **0.983** (0.005) | **0.980** (0.006) | **0.949** (0.014) | **0.945** (0.016) | **0.565** (0.065) | **0.549** (0.064) |
| | 5 | XE | 0.655 (0.021) | 0.434 (0.084) | 0.855 (0.006) | 0.831 (0.018) | - | - | - | - |
| | | QME | 0.865 (0.002) | 0.752 (0.036) | 0.950 (0.001) | 0.909 (0.012) | - | - | - | - |
| | | ME | **0.867** (0.002) | **0.827** (0.014) | **0.952** (0.001) | **0.945** (0.001) | - | - | - | - |
| Deep MLP | 1 | XE | 0.889 (0.150) | 0.813 (0.212) | 0.834 (0.128) | 0.668 (0.176) | 0.887 (0.160) | 0.840 (0.214) | 0.689 (0.095) | 0.674 (0.097) |
| | | QME | 0.956 (0.039) | 0.858 (0.152) | 0.971 (0.009) | 0.960 (0.029) | 0.943 (0.026) | 0.923 (0.091) | 0.800 (0.113) | 0.807 (0.112) |
| | | ME | **0.978** (0.034) | **0.973** (0.040) | **0.992** (0.004) | **0.991** (0.005) | **0.982** (0.029) | **0.982** (0.030) | **0.844** (0.117) | **0.839** (0.119) |
| | 2 | XE | 0.790 (0.098) | 0.748 (0.121) | 0.841 (0.060) | 0.724 (0.085) | 0.857 (0.068) | 0.821 (0.065) | 0.400 (0.031) | 0.372 (0.027) |
| | | QME | 0.947 (0.023) | 0.846 (0.113) | 0.979 (0.007) | 0.910 (0.055) | 0.926 (0.020) | 0.868 (0.085) | 0.485 (0.024) | 0.486 (0.026) |
| | | ME | **0.955** (0.024) | **0.930** (0.030) | **0.983** (0.005) | **0.975** (0.010) | **0.942** (0.014) | **0.870** (0.098) | **0.608** (0.013) | **0.598** (0.015) |
| | 5 | XE | 0.624 (0.014) | 0.500 (0.094) | 0.799 (0.017) | 0.687 (0.069) | - | - | - | - |
| | | QME | 0.850 (0.003) | 0.649 (0.068) | 0.941 (0.003) | 0.807 (0.040) | - | - | - | - |
| | | ME | **0.852** (0.002) | **0.800** (0.018) | **0.944** (0.003) | **0.919** (0.008) | - | - | - | - |
| CNN | 1 | XE | 0.633 (0.182) | 0.576 (0.158) | 0.539 (0.103) | 0.520 (0.070) | - | - | - | - |
| | | QME | 0.955 (0.031) | 0.896 (0.131) | 0.986 (0.007) | 0.952 (0.081) | - | - | - | - |
| | | ME | **0.983** (0.024) | **0.966** (0.037) | **0.997** (0.002) | **0.984** (0.020) | - | - | - | - |
| | 2 | XE | 0.568 (0.110) | 0.630 (0.162) | 0.675 (0.139) | 0.638 (0.135) | - | - | - | - |
| | | QME | 0.959 (0.017) | 0.898 (0.065) | 0.992 (0.003) | 0.945 (0.054) | - | - | - | - |
| | | ME | **0.972** (0.017) | **0.926** (0.047) | **0.994** (0.003) | **0.964** (0.040) | - | - | - | - |
| | 5 | XE | 0.625 (0.052) | 0.493 (0.144) | **0.901** (0.025) | 0.776 (0.061) | - | - | - | - |
| | | QME | 0.912 (0.004) | 0.790 (0.046) | 0.986 (0.000) | 0.909 (0.034) | - | - | - | - |
| | | ME | **0.915** (0.002) | **0.791** (0.083) | **0.988** (0.001) | **0.943** (0.034) | - | - | - | - |

For what it concerns IMP itself, for each model we have tried different level of pruning. Interestingly, Shallow MLP and Deep MLP did not show specific harm with pruning, enabling us to use a percentage of 99%. For the Convolutional Networks (CNN, ResNet18, VGG11), the optimal pruning ratio was 60%. More details about this are provided in Appendix B.7. Notice that each model is pruned over two iterations, followed by a final training phase.

Table 2 reports the aggregated metrics over seeds and merged pairs when using IMP. The full results are in Table 4 and Table 5 of Appendix C. Rewarded accuracy remains high (often $\geq 0.90$), indicating that each merged module retains its classification ability. The mean entropy on non-rewarded classes stays close to the theoretical maximum of $\log(N_{\text{classes}})$, e.g. it is around 2.28 for MNIST and FMNIST, confirming minimal interference (see Table 4 and Table 5 in the Appendix C).

Moreover, we can see that ME outperforms QME in all cases. Besides being such a small change in the learning phase (ME enforces uniformity over all neurons for non-rewarded inputs, while QME enforces uniformity over all but the rewarded ones), this loss function loses in performance with respect to ours. This surprising result is probably due to the leakage of information on non-rewarded classes. In particular, the QME incentivizes changes in the flow of information even for non-rewarded classes, which violates the MaxEnt principle. In contrast, ME en-

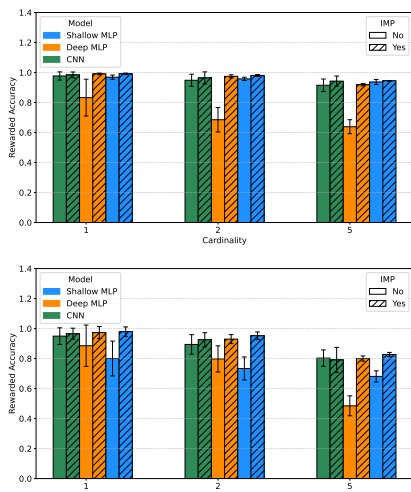

Figure 3: Impact of IMP on pairwise merging in weight-space for different architectures and cardinalities, considering the MNIST (top) and FMNIST (bottom) datasets, and using the ME loss. IMP consistently benefits the Deep MLP, while its impact is more modest for the Shallow MLP and the CNN.

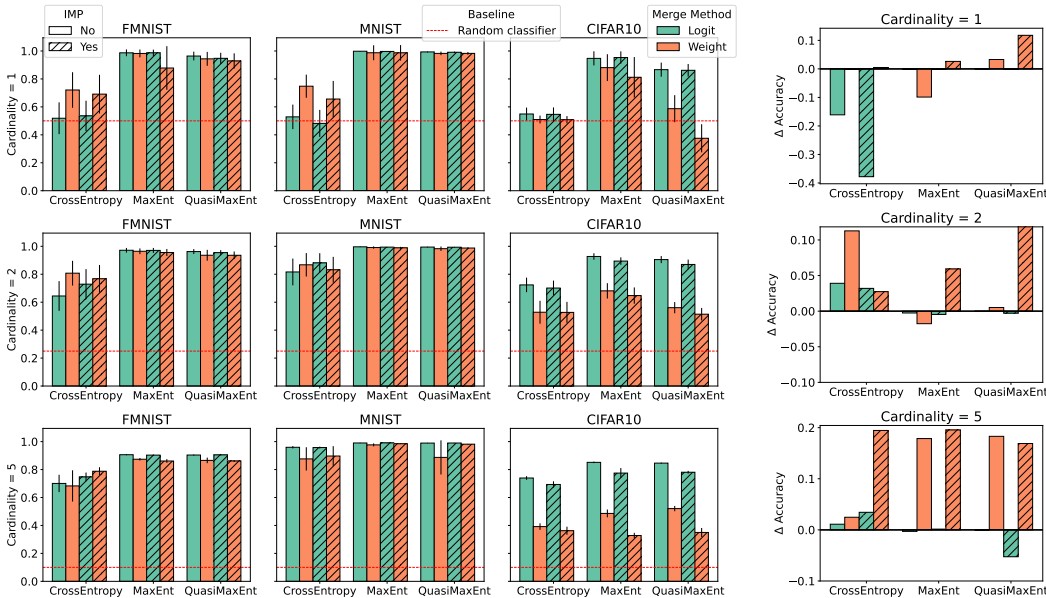

Figure 4: Left (3x3 grid): rewarded accuracy of ResNet18 on FMNIST, MNIST and CIFAR-10 across cardinalities, loss functions, and merge strategies (logit vs. weight), with and without IMP, confirming that our framework extends to larger architectures. On simpler datasets (FM-NIST, MNIST), logit- and weight-space merging yield comparable results, whereas on CIFAR-10 logit-space merging is substantially more effective. Right (vertical stack of 3 plots): difference in rewarded accuracy between VGG11 with and without BatchNorm on FMNIST ($\Delta$ = VGG11-BN $-$ VGG11), showing that BatchNorm re-estimation consistently improves weight-space merging with IMP. Results are averaged over 5 seeds and 10 pairs.

courages only information from the rewarded classes, improving the specialization and better avoiding spurious activations. Our method also outperforms standard XE. This was expected, since XE is intended to train with all the labels at once, while in this case its effectiveness is hindered by the reduced number of classes on which it is optimized.

Additionally, in Table 2 we report both the results of merging in weight-space and in logit-space. We can clearly see that logit merge outperforms the weight merge in almost every case (the only exception is CNN with $|\mathcal{R}| = 2$ with XE). Moreover, logit merge is mostly unaffected by the IMP methodology, as there is no need of pruning when the submodules are not going to be merged in weight-space, where otherwise there could be interferences.

Focusing on larger models, Figure 4 (left) shows the results on ResNet18 trained within our framework on FMNIST, MNIST and CIFAR-10 (full numerical results are in Table 6, Appendix C). These experiments demonstrate that our technique is extendible to larger architectures and suggest different analysis for simple and complex datasets. On one hand, for FMNIST and MNIST, results are similar for both logit- and weight-space merging: our ME loss consistently outperforms XE and QME, especially at cardinality 1, while IMP provides limited benefits in this setting, yielding performances roughly on par with training without IMP. These results suggest that pruning removes irrelevant weights, thus reinforcing the functional isolation induced by the loss. On the other hand, for more complex datasets like CIFAR-10, logit-space merging is much more effective, suggesting that in this setting it is not trivial to avoid potential interference between weights.

Investigating the applicability of our technique to larger networks, we have also considered the role of BatchNorm (BN). When merging in weight-space models with BN, it suffices to re-estimate the running statistics on data from the union of the rewarded classes, keeping all learned affine parameters frozen. This simple step is enough to enable weight-space merges for ResNet18 and VGG11 with BN, with a clear impact on the performance. In particular, in Figure 4 (right) we show the difference in accuracy between VGG11 with and without BN on FMNIST (full numerical results are in Table 7 and 8 in the Appendix C). Clearly, no issues arise when performing logits-space merging of models with BN. In Appendices C.8 and C.10, we further validate pairwise merging on

a large-scale image dataset (Imagenette) and on text classification (IMDB and 20 Newsgroups), confirming that the same conclusions hold in these settings.

## 4.5 MERGING ANALYSIS

In this section, we further evaluate our method in the more challenging scenario of a complete merge and provide additional analysis based on mode connectivity.

**Complete Merge.** We perform a *complete merge* experiment. For each dataset, we randomly generate 10 permutations of the class labels. For each of 5 random seeds, we train one submodule per class using either ME alone or in combination with IMP. We then merge the submodules incrementally, following the order dictated by the permutation: at step $k$, the merged model contains submodules for the first $k$ classes in the permuted list.

Figure 5 reports the average rewarded accuracy at each merge step, aggregated across permutations and seeds. We can see that submodules trained with IMP (solid lines) consistently outperform or match those trained without pruning (dashed lines) when merging in weight-space. On Yeast, performance degrades more rapidly as modules accumulate. Shallow MLPs show good composability on MNIST even in weight-space, where accuracy stays above 90% up to the full merge. Deep MLPs do not outperform shallower ones overall. CNNs also face a performance degradation, but IMP importantly benefits their performance for weight merge.

The specific comparison between Shallow and Deep MLP suggests that additional depth and narrower layers do not benefit composability in weight-space, augmenting the possibility of intereference. This suggests that width has a crucial role in enabling composability in weight-space. Moreover, CNNs face a significant degradation while ResNet18 (a much deeper model) degrades only on CIFAR-10. This further indicates the key role of BN in enabling a smooth merge even in weight-space. Additionally, we can see that logit merge significantly limits the degradation, recovering almost always the best performances on each dataset. The difference is particularly evident for CIFAR-10, for which weight merge

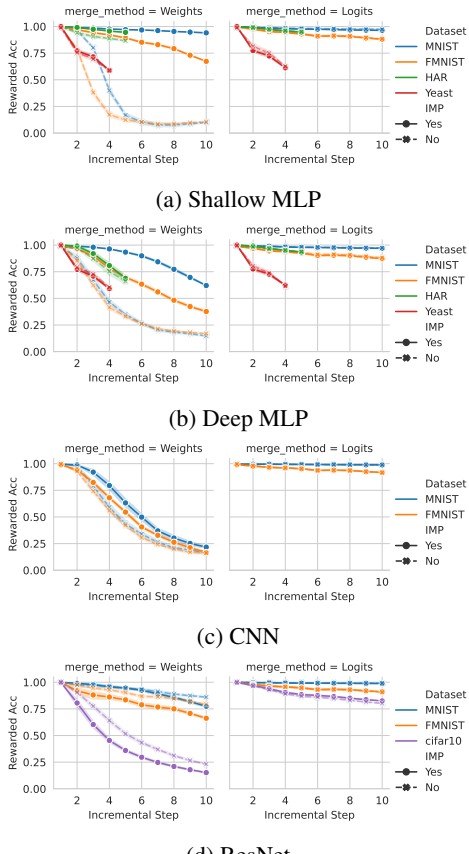

(a) Shallow MLP

(b) Deep MLP

(c) CNN

(d) ResNet

Figure 5: Average rewarded accuracy across incremental submodule merging across architectures, aggregated over 5 seeds and 10 class permutations. Solid lines denote models trained with IMP, dashed lines without. Logit-space merging (right column) consistently limits accuracy degradation across all architectures and datasets, while weight-space merging (left column) benefits substantially from IMP.

fails dramatically, while logit merge achieves good performances. As a stress test, in Appendix C.9 we extend the complete merge to 100 modules on CIFAR-100, finding that logit merging degrades gracefully across all 100 steps without a sudden compositionality bottleneck.

Overall, these results confirm that ME trains composable modules across a variety of architectures and datasets. For weight-space merging, while we have shown that it is possible to train in our proposed way, the suboptimal results indicate that further study should be conducted on pruning and merging techniques or the role of the width. Nevertheless, the strong performance of logit-space merging demonstrates that ME is always able to train composable modules, regardless of the complexity of the architectures or the dataset.

**Mode Connectivity.** To better understand why merging submodules via weight summation is often effective, we study the connectivity of models in weight-space. Specifically, we follow the formulation of Lubana et al. (2023), and evaluate whether the merged model lies on a low-loss path between its components, using the ME loss as criterion. We analyze complete merges on the

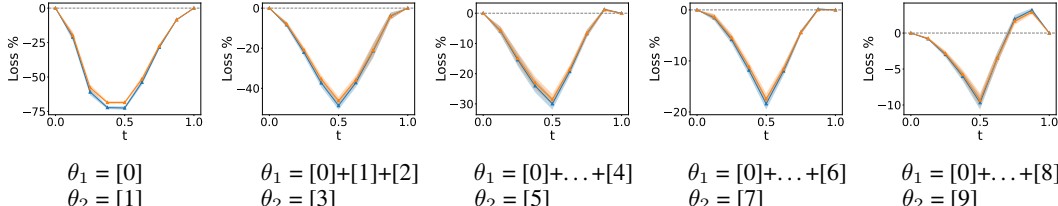

$$\theta_1 = [0] \qquad \theta_1 = [0]+[1]+[2] \qquad \theta_1 = [0]+\ldots+[4] \qquad \theta_1 = [0]+\ldots+[6] \qquad \theta_1 = [0]+\ldots+[8]$$
$$\theta_2 = [1] \qquad\qquad \theta_2 = [3] \qquad\qquad\quad \theta_2 = [5] \qquad\qquad\quad \theta_2 = [7] \qquad\qquad\quad \theta_2 = [9]$$

Figure 6: Loss barrier along the linear interpolation path $\gamma_{\theta_1 \to \theta_2}(t)$ considering multiple steps of the complete merge starting from 0 to 9 for the FMNIST dataset (train and test errors, in blue and orange respectively, mean and std across 5 runs). Loss values at each $t$ are relativized with respect to the corresponding interpolated errors of $\theta_1$ and $\theta_2$ and rescaled as percentages. Values close to or lower than 0 indicate mode connectivity. Barriers remain near zero throughout most of the merge sequence, supporting the hypothesis that ME training induces weight-space composability, with only mild interference emerging in the later steps.

FMNIST dataset using shallow MLPs. For each step $k$ in the merge sequence—e.g., combining a submodule obtained from the merge $[0] + \cdots + [k-1]$ and one trained on class $k$—we evaluate ME along a piecewise linear path between the previous model and the new submodule, with the midpoint corresponding to their weight summation. Figure 6 shows the relative loss barrier for each step, computed as the gap between the loss at the midpoint (merged model) and the linear interpolation of the endpoint losses. Values near or below zero indicate mode connectivity. In the majority of cases, merged models correspond to local minima or lie along smooth, low-loss trajectories, with no significant barriers emerging along the merge path. This supports the hypothesis that the training procedure induces functional composability in weight-space. However, small barriers (around 2%) begin to emerge in the later steps of the complete merge—when many submodules have already been added—suggesting mild interference.

## 5 DISCUSSION AND LIMITATIONS

Our proposed modular design might open new avenues, but also presents limitations and open challenges. IMP seems to help submodules isolation and thus weight-space merging, but the overlap between submodules should be carefully demistified, in order to optimize the best IMP strategy to isolte and merge subnetworks in this setting. While we rely on simple summation for the weight-space merge, more sophisticated merging techniques—e.g., Git-Rebasin or PLeas (Ainsworth et al., 2023; Nasery et al., 2025)—may further improve composability, particularly in challenging architectures. However, merge in logit-space already allows to compose our modules regardless of the architecture.

Furthermore, we believe effective modular training can find application in machine unlearning (e.g., removing or re-training just the single module), NN verification (where formally verifying an entire network is often intractable), federated learning (where merging different networks is a key step), mechanistic interpretability by design, and many other settings. Showing concrete downstream uses is indeed valuable, and we plan to explore this in multiple future works.

## 6 CONCLUSIONS

We introduced a principled framework for training sparse neural submodules that specialize in recognizing a designated subset of classes while deliberately remaining ignorant of others. This behavior is encouraged via a novel KL-based MaxEnt loss. In addition, we propose to use an IMP procedure to select only the relevant weights, i.e. the circuit. Together, they yield compact, high-performing subnetworks that with simple datasets and architectures can be composed via simple weight summation—a procedure supported by empirical evidence showing that such modules are mode connected. For complex datasets and architectures, we show that the modules are composable in logit-space, producing product of experts, each one specialized on a specific class, and thus highly interpretable.

Our approach strengthens the agenda of modular NNs by providing submodules with well-defined functional roles. At the same time, it creates a controlled testbed for mechanistic interpretability, where circuits are available by design rather than discovered post-hoc. In the long run, we envision functional submodules as a key abstraction for building interpretable neural systems.

REPRODUCIBILITY STATEMENT

To ensure reproducibility of our results, all code necessary to reproduce the experiments presented in this paper is provided at the following link: https://github.com/FrancescoCaso/Composable-Sparse-Subnetworks-MaxEnt. Additional details are provided in Appendix B, which thoroughly documents the experimental setting: training configurations and pipeline, evaluation metrics, mode connectivity tests, datasets, pruning schedules, and hyperparameter choices.

AUTHOR CONTRIBUTIONS

This paper is the result of a real team effort, in which each author has been fundamental to the final outcome. Specifically, all authors contributed to refining the methodology, programming and extending the codebase, conducting experiments, analyzing the results, and revising the manuscript. Among the authors, Francesco Caso stands out since he originated the main research idea, defined the theoretical motivation based on the maximum entropy principle and coordinated the project. Fabrizio Silvestri supervised the work, provided guidance, and contributed to shaping the research direction.

ACKNOWLEDGMENTS

Francesco Caso, Samuele Fonio, Simone Monaco, and Nicola Saccomanno sincerely thank Prof. Pietro Liò for hosting them during a research visit in 2024, which laid the foundation for this work. Moreover, they also thank Lorenzo Petrosino for valuable discussions during the initial stages of the project. Francesco Caso acknowledges support from Renaissance Philanthropy through the AI for Math Fund for research conducted at the Department of Computer Science and Technology, University of Cambridge. Nicola Saccomanno acknowledges travel support from the European Union's Horizon 2020 research and innovation programme under grant agreement No 951847, and the support from the Interconnected Nord-Est Innovation Ecosystem (iNEST), which received funding from the European Union Next-GenerationEU (PIANO NAZIONALE DI RIPRESA E RESILIENZA (PNRR) – MISSIONE 4 COMPONENTE 2,INVESTIMENTO 1.5 – D.D. 1058 23/06/2022, ECS00000043). Simone Monaco acknowledges travel support from ELIAS (GA no 101120237).

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

# COMPOSABLE SPARSE SUBNETWORKS VIA MAXIMUM-ENTROPY PRINCIPLE - APPENDIX

## APPENDIX STRUCTURE

The appendix is structured as follows:

- Appendix A describes the extended **related work**;

- Appendix B focuses on detailing the **experimental setting** of our work. Specifically, we further describe the **training configurations and pipeline** (Appendix B.1), the **metrics** employed to evaluate the proposed methods (Appendix B.2), the formal definitions of the **baseline loss functions** (Appendix B.3), the procedure we developed to test for presence or absence of **mode connectivity** (Appendix B.4), the **datasets** considered for our experiments (Appendix B.5), the **pruning ratio per iteration** (Appendix B.6), and the **hyperparameters** of our proposed procedure (Appendix B.7);

- Appendix C focuses on providing additional **experimental results** to support the findings in the main text. We report extended **confusion matrices** for individual submodules without (Appendix C.1) and with pruning (Appendix C.2), as well as for **pairwise** (Appendix C.3) and **incremental merges** (Appendix C.4). We also include **loss landscape plots** from the mode connectivity analysis (Appendix C.5), and a comprehensive table comparing merging results **with and without pruning across all merge strategies** (Appendix C.6), as well as a full **baseline comparison** including Confidence Penalty and Label Smoothing (Appendix C.7). Finally, to further analyze possible extension of our framework, we present experimental results on Imagenette, demonstrating generalisation to **large-scale image datasets** (Appendix C.8), experimental results on CIFAR-100, probing the limits of compositionality with **up to 100 merged modules** (Appendix C.9), and experimental results on **text-classification** benchmarks (IMDB and 20 Newsgroups), showing that the method transfers beyond image classification (Appendix C.10).

## A    EXTENDED RELATED WORK

**Modular Neural Networks and Interpretability.**    Several works have explored modular neural architectures that learn to decompose computation across tasks (Kirsch et al., 2018; Han et al., 2021; Salem et al., 2023). In particular, Kirsch et al. (2018) introduced a method to jointly learn both the modules and their composition end-to-end, showing interpretable specialization through controller-based selection. However, their approach does not isolate class-specific knowledge: modules are reused across multiple tasks and selected based on input context, making it difficult to extract independently meaningful functional units.

In contrast, our work explicitly aims to construct *class-specific functional modules*, where each module is trained to specialize in a single class and produce maximum-entropy outputs for all others. This sharp functional separation not only promotes compositionality via weight merging, but also improves *interpretability*: submodules can be understood in terms of their response to a single concept. Unlike post-hoc analyses, we impose structure *during training*, yielding functional units with clear class semantics and compositional behavior. This aligns with the goals of mechanistic interpretability (Saphra and Wiegreffe, 2024), where subnetworks (or "circuits" (Olah et al., 2020)) are analyzed as self-contained structures responsible for identifiable sub-tasks.

Our method can be seen as a complement to neuron-level analysis techniques such as those in Mu and Andreas (2020), which decompose single neuron activations into logical or perceptual components. However, such analyses often suffer from *polysemanticity*—the same neuron responding to unrelated stimuli—limiting their explanatory power. Instead, we enforce functional modularity at the network level, allowing the identification of interpretable, composable subgraphs. This modular decomposition is also conceptually connected to recent work on *harmonics of learning* (Marchetti et al., 2024), where symmetry structures constrain the function space; our class-specific modules may correspond to symmetry-aligned components in this broader framework.

**Maximum Entropy Principle.**    The principle of maximum entropy (MaxEnt), introduced by Jaynes (1957), asserts that when making inferences under partial knowledge, one should prefer the probability distribution that satisfies known constraints while remaining otherwise maximally uninformative. Originally formulated to connect statistical mechanics and information theory, MaxEnt has since become a foundational tool in probabilistic modeling and statistical inference (Kuić, 2016; De Martino and De Martino, 2018). Its generalization to complex-valued functions has also been explored, for example in image reconstruction tasks (Bajkova, 1992).

In our setting, MaxEnt serves as a guiding principle for functional specialization: we train submodules to be highly confident only on their assigned class and maximally uncertain (i.e., high-entropy) elsewhere. This interpretation differs from standard entropy regularization, which is typically used to smooth predictions or calibrate confidence scores (Marczak et al., 2024). Rather than encouraging mild uncertainty, our maximum-entropy loss enforces *uniformity* outside the rewarded class set—actualizing the MaxEnt principle to promote class exclusion and module isolation. To our knowledge, this use of entropy as a strict compositional prior is novel within the context of modular deep learning.

**Lottery Ticket Hypothesis and Sparse Training.**    Our approach builds on the Lottery Ticket Hypothesis (LTH), which states that sparse subnetworks in randomly initialized networks can be trained to match the performance of the full dense model (Frankle and Carbin, 2018; Zhou et al., 2019; Liu et al., 2024). This idea has been extended theoretically to show that such subnetworks exist with high probability even in convolutional architectures with ReLU activations (Burkholz, 2022). Empirical studies also demonstrate that global unstructured pruning outperforms structured pruning in high-sparsity regimes (Girish et al., 2021). We adopt global pruning in our setting to retain the most functionally relevant connections across the entire network. LTH has been further explored in transfer learning (Van Soelen and Sheppard, 2019; Burkholz et al., 2022) and federated learning (Itahara et al., 2020). In contrast to prior work that uses pruning primarily for compression or transfer, we leverage it to enforce *functional isolation*: pruning removes spurious capacity, encouraging submodules to specialize only on their rewarded classes.

**Mode Connectivity and Model Merging.**    Recent work on model merging has explored how independently trained networks can be combined through linear or non-linear paths in weight-space,

a property known as *mode connectivity* (Ilharco et al., 2023). This has inspired techniques for task arithmetic (Ortiz-Jimenez et al., 2023; Yang et al., 2024; Wang et al., 2024), which define task vectors as differences between fine-tuned and base models, enabling operations like unlearning and multi-task interpolation. These methods typically assume large pre-trained models and alignment in parameter space, and focus on editing rather than modular design.

Other approaches, such as Git-Rebasin or PLeas (Hazimeh et al., 2024; Zeng et al., 2025), address the challenge of merging models with different initializations via weight alignment. While these techniques focus on how to combine models *after* training, our method focuses on how to construct modules that are composable *by design*. Specifically, we show that submodules trained with maximum-entropy loss and pruning often exhibit linear mode connectivity—allowing for effective merging via weight summation—while also identifying cases where destructive interference occurs.

Our results suggest that mode connectivity may emerge naturally when modules are trained under entropy-based functional isolation, without requiring explicit alignment. This positions our work as complementary to task arithmetic and merging strategies: we do not assume pre-trained baselines, but instead provide a procedure to build sparse, specialized components that are compatible by construction.

## B  EXTENDED EXPERIMENTAL SETTING

### B.1  TRAINING CONFIGURATIONS AND PIPELINE

We report here the key training configurations used in our experiments. All models were trained using the maximum-entropy loss (ME) and iterative magnitude pruning (IMP), with two pruning iterations and a final sparsity of 99% for MLPs and 60% for CNNs, unless stated otherwise. The optimizer used was Adam. We use the highest accuracy achieved on the validation set as our checkpointing strategy. All experiments are conducted on a workstation equipped with an Intel Core i9-10940X (14-core CPU running at 3.3GHz), 256GB of RAM, and a single Nvidia RTX A6000 GPU with 48GB of VRAM.

Temperature scaling for the softmax was fixed to $t$. Inputs were normalized when appropriate, and all datasets were used with standard training/validation splits. Each experiment was run with 5 random seeds.

*All YAML configuration files are released alongside the code repository for full reproducibility.*

**Model Architectures and Hyperparameters**

**Common Settings.**  All models use ReLU activations, are trained with a batch size of 64, and are evaluated using the same training and pruning pipeline.

Table 3: Architecture configurations for the MLP and CNN models used in our experiments, including hidden layer sizes, dropout rate, and the temperature scaling for the softmax $t$.

| Model | Hidden Layers | Dropout | $t$ |
|---|---|---|---|
| Shallow MLP | [512] | 0.5 | 5.0 |
| Deep MLP | [512, 256] | 0.5 | 5.0 |
| CNN (LeNet-style) | Conv + [120, 64] | 0. | 1.0 |

The learning rate was set to $10^{-3}$ for MNIST and FMNIST across all architectures, and to $10^{-4}$ for Yeast and Human Activity Recognition, where only Shallow and Deep MLPs were evaluated.

**Dataset-specific Input Dimensions and Classes**

- **MNIST / FMNIST:** 784 input features (flattened $28 \times 28$ images), 10 classes.
- **Human Activity Recognition (HAR):** 561 input features, 5 classes.
- **Yeast:** 8 input features, a balanced subset of 4 classes.

**Training Pipeline.**  Here we present an overview of a generic experiment. For each experiment:

- Given the set of classes $\mathcal{C}$ in the dataset, choose a subset of classes $\mathcal{C}' \subset \mathcal{C}$.
- Train one subnetwork module per class or per subset $\mathcal{R} \subseteq \mathcal{C}'$, using ME and IMP.
- Evaluate each subnetwork using:
    - **Accuracy** on rewarded classes;
    - **Mean entropy** on non-rewarded classes;
    - **Confusion matrix** on the validation set.

### B.2  FORMAL DEFINITION OF METRICS

**Accuracy on Rewarded Classes.**  Let $\mathcal{R} \subseteq \mathcal{C}$ be the set of rewarded classes, and let $\mathcal{D}_{\text{val}}^{\text{R}}$ denote the subset of validation data whose labels belong to $\mathcal{R}$. The accuracy over rewarded classes is defined as:

$$\text{Accuracy}_{\text{rewarded}} = \frac{1}{|\mathcal{D}_{\text{val}}^{\text{R}}|} \sum_{(x,y) \in \mathcal{D}_{\text{val}}^{\text{R}}} \mathbb{1}\left[\hat{y}(x) = y\right] \quad , \tag{7}$$

where $\mathbb{1}[\cdot]$ is the indicator function. This measures how well the subnetwork performs on the class subset it is trained to recognize.

**Mean Entropy on Non-Rewarded Classes.** Let $\mathcal{D}_{\text{val}}^{\text{non-R}}$ be the validation subset containing samples from non-rewarded classes. We define the average entropy of the model's predictions over this set as:

$$\text{Entropy}_{\text{non-rewarded}} = \frac{1}{|\mathcal{D}_{\text{val}}^{\text{non-R}}|} \sum_{x \in \mathcal{D}_{\text{val}}^{\text{non-R}}} H\left(\hat{y}(x)\right) \quad , \tag{8}$$

where $\hat{y}(x)$ is the predicted probability distribution for input $x$, and $H(p)$ denotes the entropy of a distribution $p$, defined as:

$$H(p) = -\sum_{i=1}^{|\mathcal{C}|} p_i \log p_i \quad . \tag{9}$$

This metric quantifies the uncertainty the model exhibits on inputs it is not supposed to classify, with higher values indicating closer adherence to the maximum-entropy principle.

**Confusion Matrix.** Given a validation dataset $\mathcal{D}_{\text{val}}$ and a trained model $f_\theta$, we define the confusion matrix $\mathbf{M} \in \mathbb{N}^{|\mathcal{C}| \times |\mathcal{C}|}$ such that each entry $M_{i,j}$ counts the number of samples with true label $i$ that are predicted as class $j$:

$$M_{i,j} = \sum_{(x,y) \in \mathcal{D}_{\text{val}}} \mathbb{1}[y = i] \cdot \mathbb{1}[\hat{y}(x) = j] \quad , \tag{10}$$

where $\hat{y}(x) = \arg\max_k f_\theta(x)_k$ is the predicted class. This matrix allows us to qualitatively inspect specialization and interference across class predictions.

### B.3 FORMAL DEFINITION OF BASELINES

- Quasi-MaxEnt loss (QME): a loss function that on non-rewarded classes ignores the rewarded classes and produces a uniform distribution solely on non-rewarded classes. Formally referring to equation 1:

$$\tilde{y}_i = \begin{cases} \delta_{i=y} & \text{if } y \in \mathcal{R}, \\ \delta_{i \neq j, j \in \mathcal{R}} \frac{1}{|\mathcal{C} \backslash \mathcal{R}|} & \text{otherwise.} \end{cases} \tag{11}$$

- CrossEntropy loss (XE): the standard loss used in standard classification tasks. In this case, we do not modify the labels, which remain standard, but the model is exposed only to the classes in $\mathcal{R}$, rather than all available classes.

- XE with Label-Smoothing ($\lambda = 0.1$), which replaces the one-hot target distribution with a smoothed version

$$y_k^{\text{LS}} = \begin{cases} 1 - \lambda, & \text{if } k = y, \\ \frac{\lambda}{K-1}, & \text{otherwise.} \end{cases}$$

- XE with Confidence-Penalty, which adds an entropy-regularization term

$$\mathcal{L} = \text{XE}(p, y) - \beta\, H(p) \quad .$$

### B.4 Additional explanations on Mode Connectivity

To further shed light on why merging via summation works (as demonstrated in the main paper) for simple datasets, we study also the loss landscape through the lens of mode connectivity.

Mode connectivity refers to a phenomenon observed in the loss landscape of neural networks. When a neural network is trained, the optimization process (e.g. SGD) typically finds a set of weights that minimizes the loss function. This set of weights is called a *mode* or a *local minimum* in the high-dimensional loss landscape. Several contributions (Frankle et al., 2020; Lubana et al., 2023) pointed out that these different modes (solutions found by, e.g., training the same model architecture multiple times with different initializations or training procedures) are often not isolated; rather, they can frequently be connected by paths along which the loss value remains consistently low.

Many paths of different shapes can connect two models, $\theta_1$ and $\theta_2$, in weight-space. We are specifically interested in testing for a path without significant loss barriers that includes the model $\theta_{[1,2]} = \theta_1 + \theta_2$, which results from merging the original models via summation. A key aspect of our setup, diverging from typical mode connectivity studies, is that $\theta_1$ and $\theta_2$ are initially specialized for different sets of rewarded classes. The merged model $\theta_{[1,2]}$ is intended to operate on the union of these class sets. Consequently, we evaluate for barriers using our *maximum-entropy loss* function. This loss function is consistently applied across the path and for the endpoint models ($\theta_1, \theta_2$), and it considers the full union of rewarded classes relevant to $\theta_{[1,2]}$.

Formally, in line with Lubana et al. (2023), let $\theta_1$ and $\theta_2$ be two sets of weights, $\mathcal{D}$ a dataset, $f_\theta$ a neural network parametrized by weights $\theta$, and $\mathcal{L}$ our specific *maximum-entropy loss* function (evaluated on $\mathcal{D}$ considering the union of rewarded classes as described above). We say that $\theta_1$ and $\theta_2$ are mode connected along the path $\gamma_{\theta_1 \to \theta_2}(t)$ if $\forall t \in [0,1] \, \mathcal{L}(f_{\gamma_{\theta_1 \to \theta_2}(t)}(\mathcal{D})) \leq (1-t) \cdot \mathcal{L}(f_{\theta_1}(\mathcal{D})) + t \cdot \mathcal{L}(f_{\theta_2}(\mathcal{D})) + \epsilon$, where $\epsilon$ is a small margin, set to 2% of the first term on the r.h.s. following Frankle et al. (2020).

Each point along the path $\gamma_{\theta_1 \to \theta_2}(t)$ represents a valid set of weights for the network $f(\cdot)$. Standard *linear* mode connectivity considers the path $\gamma^{\text{LMC}}_{\theta_1 \to \theta_2}(t) = (1-t) \cdot \theta_1 + t \cdot \theta_2$. For our case, we require a path $\gamma_{\theta_1 \to \theta_2}(t)$ that satisfies: (i) for $\gamma_{\theta_1 \to \theta_2}(0) = \theta_1$, (ii) for $\gamma_{\theta_1 \to \theta_2}(1) = \theta_2$, and (iii) $\exists t' \in [0,1] \, s.t. \, \gamma_{\theta_1 \to \theta_2}(t') = \theta_1 + \theta_2$ . As a consequence, we define the mode connectivity with respect to the following *piecewise linear* path:

$$\gamma_{\theta_1 \to \theta_2}(t) = \begin{cases} \theta_1 + 2t \cdot \theta_2 & \text{if } t \leq 0.5 \\ 2(1-t) \cdot \theta_1 + \theta_2 & \text{if } t > 0.5 \end{cases} \tag{12}$$

It is trivial to see that our definition fulfill our properties, especially (iii) for $t = 0.5$.

### B.5 Dataset Details

**MNIST and Fashion-MNIST.**  Both are standard 10-class image classification benchmarks with grayscale $28 \times 28$ images. We normalize pixel values to $[0,1]$ and use the standard train/validation splits (60,000 train, 10,000 test).

**Human Activity Recognition (HAR).**  This dataset contains sensor data from smartphones, labeled with six different physical activities. Each example has 561 standardized features. We adopt the original train/test split (7,352 train, 2,947 test) and reserve a validation portion from the training set.

**Yeast.**  A protein localization dataset with 10 classes and 8 features. We normalize each feature to zero mean and unit variance, and split the dataset using a 65/15/20 ratio (on 1,299 samples). Since the class distribution is highly imbalanced, we focused on the most represented 4 classes (i.e., CYT, NUC, MIT, ME3).

**Preprocessing.**  For all tabular datasets, we apply z-score normalization and encode labels as integers. We use early stopping on the validation set and batch sizes of 64 during training.

### B.6   ITERATIVE PRUNING SCHEDULE DERIVATION

We follow the Iterative Magnitude Pruning (IMP) framework, in which a neural network is pruned over multiple iterations. At each pruning step, a fraction $K$ of the remaining weights (those with the smallest absolute value) is removed. This process is repeated for $N$ iterations until a target sparsity level $P \in (0, 1)$ is reached, corresponding to retaining only a $1 - P$ fraction of the original parameters.

**Pruning Rate Schedule.**   Let $S_i$ denote the fraction of weights remaining after the $i$-th pruning iteration. The pruning process is multiplicative, meaning that:

$$S_i = (1 - K)^i,$$

assuming $S_0 = 1$ (i.e., all weights are initially present). After $N$ pruning iterations, we desire a final sparsity $P$, which implies a remaining fraction $S_N = 1 - P$. Therefore, we solve:

$$(1 - K)^N = 1 - P.$$

Solving for $K$, we obtain the pruning rate per iteration:

$$K = 1 - (1 - P)^{1/N}.$$

This schedule ensures that pruning a fixed fraction $K$ of the remaining weights at each of the $N$ steps results in an overall sparsity of $P$ at the end of the iterative process.

**Pruning Algorithm.**   Algorithm 2 summarizes the full procedure used in our experiments.

---
**Algorithm 2** Iterative Magnitude Pruning with Maximum-Entropy Loss

---
**Require:** Initial weights $\theta_0$, dataset $\mathcal{D}$, rewarded classes $\mathcal{R}$, number of pruning iterations $N$, target sparsity $P$, epochs per iteration $E$
1: $\theta \leftarrow \theta_0$
2: $K \leftarrow 1 - (1 - P)^{1/N}$
3: **for** iteration $i = 1$ to $N$ **do**
4:    Train model $f_\theta$ on $\mathcal{D}$ using maximum-entropy loss with reward set $\mathcal{R}$ for $E$ epochs
5:    Prune the fraction $K$ of weights in $\theta$ with the smallest absolute value
6:    Reset remaining weights in $\theta$ to their initial values from $\theta_0$
7: **end for**
8: Train the final pruned subnetwork on $\mathcal{D}$ using the maximum-entropy loss with $\mathcal{R}$ for $E$ epochs

---

### B.7   HYPERPARAMETER TUNING

We perform a small-scale hyperparameter study with respect to two key components of the training pipeline: the softmax temperature used in the maximum-entropy loss, and the pruning ratio applied in iterative magnitude pruning. The temperature analysis is conducted on MNIST using a shallow MLP in the *complete merge* setting. For the pruning ratio, we provide two complementary studies: one on the shallow MLP in the same complete merge scenario, and one on CNNs (LeNet-style) in the *pairwise merge* setting, using both MNIST and FashionMNIST. All results are averaged over 5 random seeds.

**Temperature.**   We study the effect of the temperature hyperparameter $t \in \{1, 3, 5, 7\}$, which scales the logits before the softmax activation and thus controls the sharpness of the output distribution. Lower temperatures produce more peaked distributions, while higher values encourage uniformity.

Figure 7a and 7b show the average entropy on non-rewarded classes and the rewarded accuracy across merge steps for different temperatures.

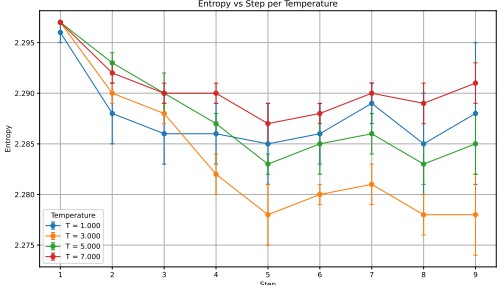 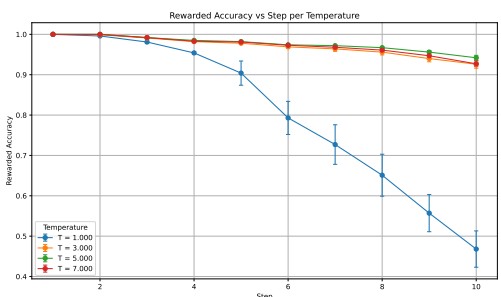

(a) Entropy on non-rewarded samples at each incremental merge step for different temperatures.

(b) Rewarded accuracy at each incremental merge step for different temperatures.

Figure 7: Effect of temperature scaling on specialization and composability. Temperature $t$ controls the confidence of the softmax output; $t = 5$ achieves a good trade-off between high entropy on non-rewarded classes and accuracy on rewarded ones.

**Pruning Ratio.** We evaluate the impact of pruning ratio on submodule performance and merge behavior for two architectures: a shallow MLP and a CNN (LeNet-style). For the shallow MLP, we focus on the *complete merge* setting, testing three final sparsity levels—90%, 99%, and 99.9%—adjusted across two IMP iterations. As shown in Figure 8, pruning to 99% yields the best trade-off between specialization and compositionality: it maintains high entropy on non-rewarded inputs and preserves rewarded accuracy throughout the merge process. Lower sparsity (90%) retains excessive capacity, causing interference when modules are merged. Higher sparsity (99.9%) leads to underfitting and degraded performance.

For CNNs, we instead study pruning in the *pairwise merge* setting, focusing on class subset sizes $|\mathcal{R}| = 2$ and $5$, across MNIST and FashionMNIST. The results, summarized in Figure 9, show that a pruning ratio of 0.6 consistently yields the highest average rewarded accuracy across both datasets and cardinalities. This indicates that, unlike MLPs, CNNs benefit from less aggressive sparsification.

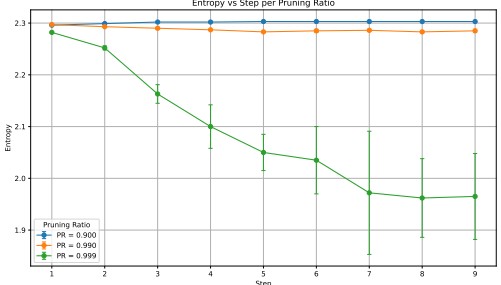 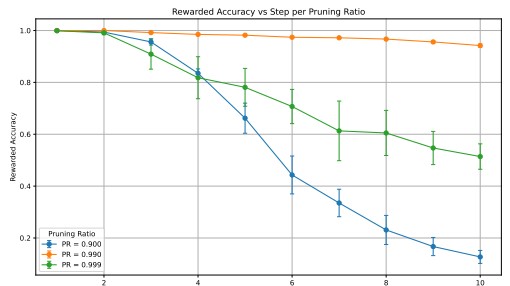

(a) Entropy on non-rewarded samples across merge steps for different pruning ratios.

(b) Rewarded accuracy across merge steps for different pruning ratios.

Figure 8: Effect of pruning ratio on submodule specialization and merge stability. A pruning ratio of 0.99 yields a good balance between maintaining accuracy and increasing entropy on non-rewarded classes.

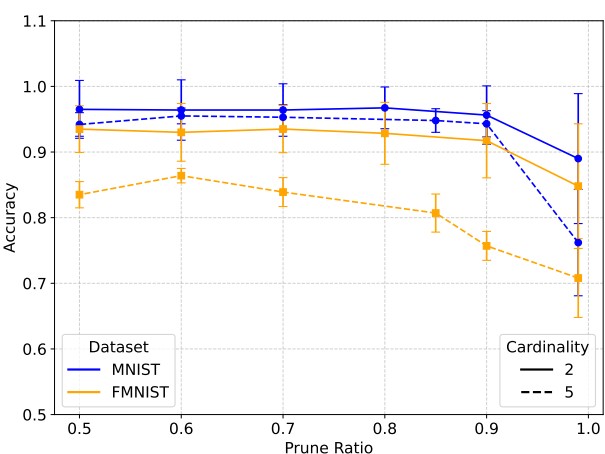

Figure 9: Effect of pruning ratio on CNN accuracy after pairwise merging on MNIST and Fashion-MNIST, with cardinality $|\mathcal{R}| = 2$ and $5$. The optimal pruning ratio is $0.6$.

## C    EXTENDED EXPERIMENTS

In this section, we present additional experimental results and qualitative insights.

First, we report representative examples for three settings:

- **Shallow MLP on MNIST**
- **Deep MLP on Human Activity Recognition**
- **CNN (LeNet-style) on FashionMNIST**

Then, we provide the following extended results, each complementing the findings reported in the main paper:

- a comprehensive set of tables and figures reporting the full pairwise submodule merging results across all configurations—varying model architectures, pruning settings (with and without IMP), merge strategies (logit vs. weight), cardinalities $|\mathcal{R}|$, and loss functions (XE, ME, QME). These extend the main paper, where only pruned results were shown (Appendix C.6);
- a complete baseline comparison including Confidence Penalty (CP) and Label Smoothing (LS), showing that ME is not a re-parameterisation of standard regularisers (Appendix C.7);
- pairwise and complete merge results on **Imagenette**, demonstrating generalisation to large-scale image data (Appendix C.8);
- pairwise and complete merge results on **CIFAR-100** with up to 100 modules, stress-testing the limits of compositionality (Appendix C.9);
- pairwise merge results on **IMDB** and **20 Newsgroups**, showing that the framework transfers beyond vision tasks (Appendix C.10).

### C.1    CONFUSION MATRICES (NO PRUNING)

In this subsection, we show submodules trained using the maximum-entropy loss without applying iterative magnitude pruning (IMP). For each model-dataset pair, we show the confusion matrix of a submodule trained on a single rewarded class. These visualizations allow us to assess the degree of specialization and the presence of residual structure in the predictions for non-rewarded classes.

In all cases, the models exhibit strong activation on the rewarded class, while showing varying degrees of non-uniformity on the excluded classes—especially evident in the confusion matrix rows corresponding to non-rewarded labels.

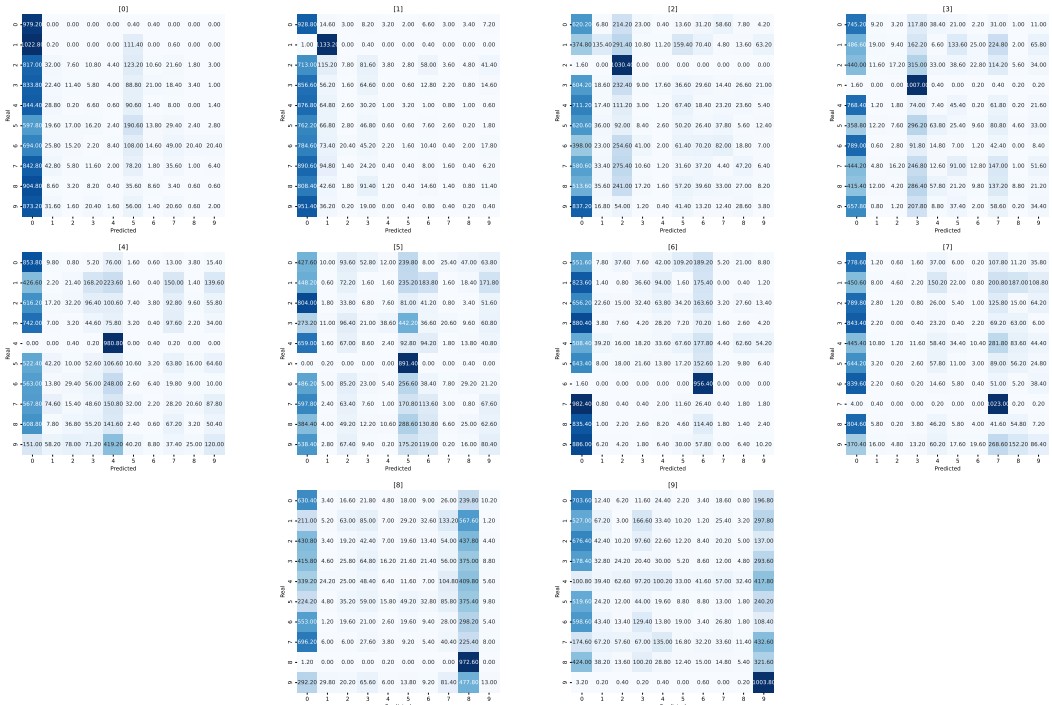

Figure 10: Confusion matrices (total predicted counts per class) for submodules trained on classes 0–9 without pruning, using the maximum-entropy loss. Each model is trained to specialize on one class and suppress predictions on others. Results refer to the **Shallow MLP** architecture applied to the **MNIST** dataset.

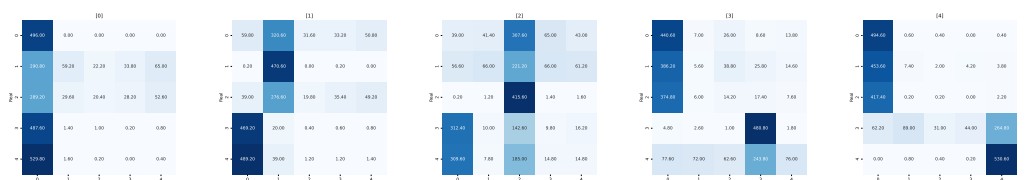

Figure 11: Confusion matrices (total predicted counts per class) for submodules trained on classes 0–4 without pruning, using the maximum-entropy loss. Each model is trained to specialize on one class and suppress predictions on others. Results refer to the **Deep MLP** architecture applied to the **HAR** dataset.

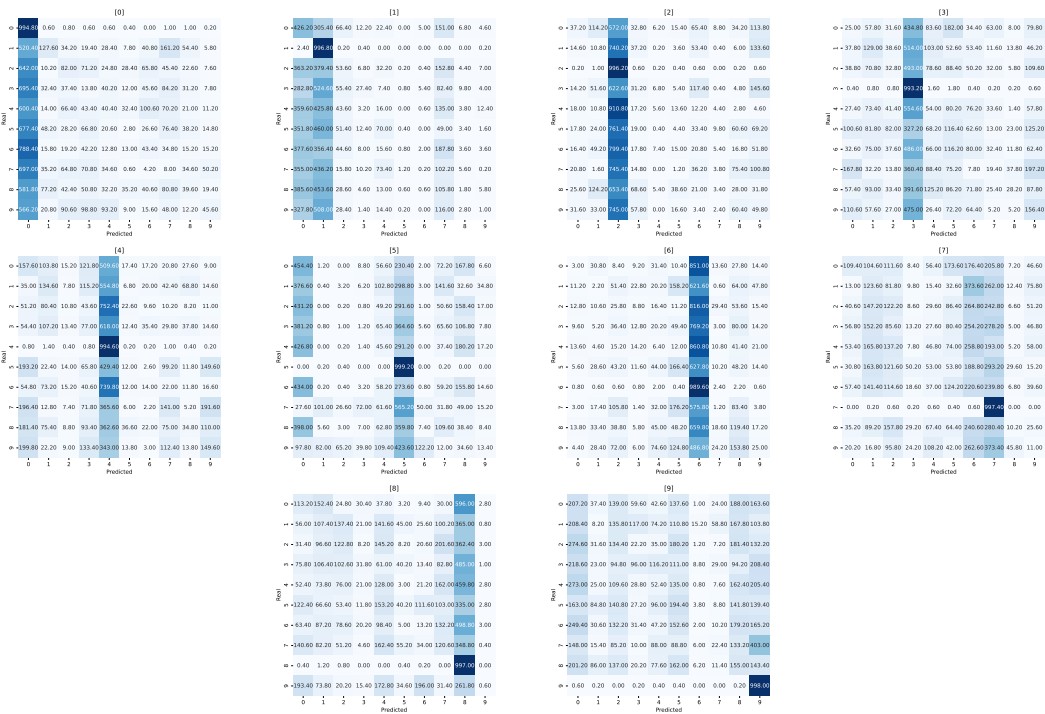

Figure 12: Confusion matrices (total predicted counts per class) for submodules trained on classes 0–9 without pruning, using the maximum-entropy loss. Each model is trained to specialize on one class and suppress predictions on others. Results refer to the **CNN** architecture applied to the **FMNIST** dataset. The CNN was pruned with a 99% pruning ratio.

## C.2 CONFUSION MATRICES AFTER PRUNING

We report confusion matrices for submodules trained with both the maximum-entropy loss and iterative magnitude pruning (IMP). These models are expected to exhibit stronger functional isolation, with reduced leakage across non-rewarded classes. Comparisons with Section C.1 highlight the impact of pruning on output uniformity and specialization.

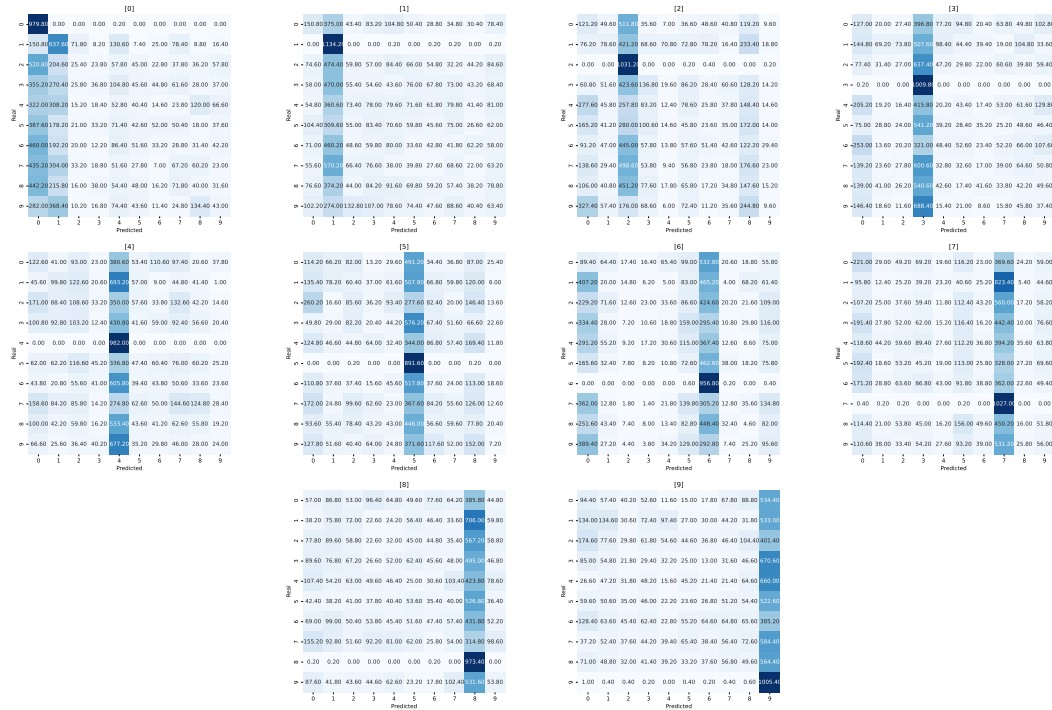

Figure 13: Confusion matrices (total predicted counts per class) for submodules trained on classes 0–9, using the maximum-entropy loss and pruning. Each model is trained to specialize on one class and suppress predictions on others. Results refer to the **Shallow MLP** architecture applied to the **MNIST** dataset.



Figure 14: Confusion matrices (total predicted counts per class) for submodules trained on classes 0–4, using the maximum-entropy loss and pruning. Each model is trained to specialize on one class and suppress predictions on others. Results refer to the **Deep MLP** architecture applied to the **HAR** dataset.

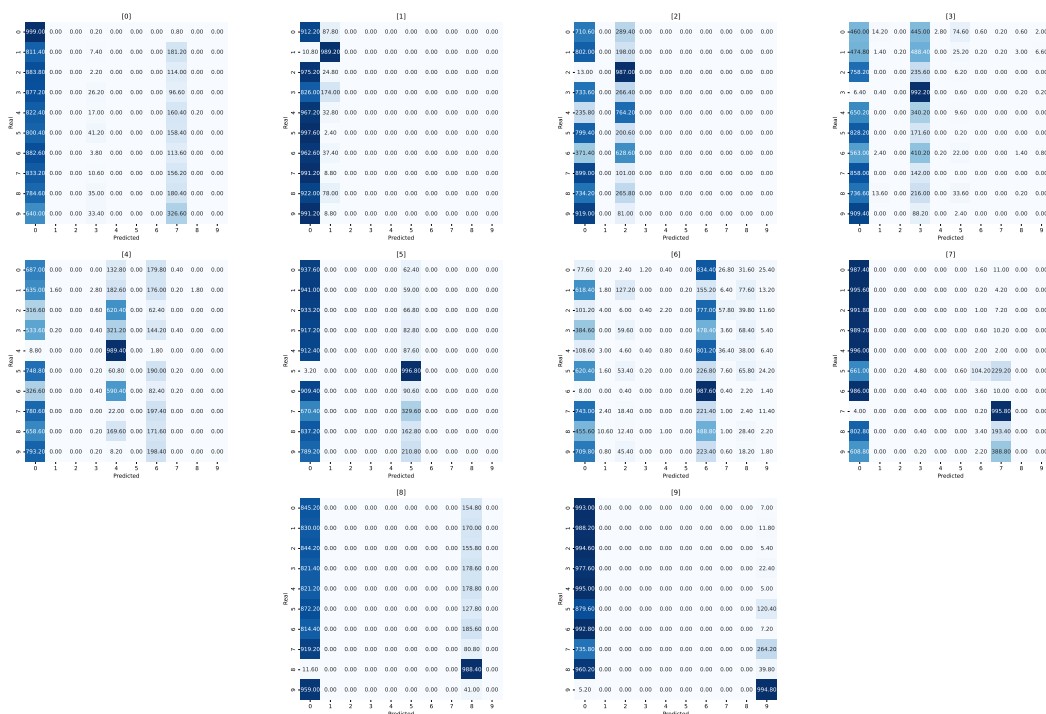

Figure 15: Confusion matrices (total predicted counts per class) for submodules trained on classes 0–9, using the maximum-entropy loss and pruning. Each model is trained to specialize on one class and suppress predictions on others. Results refer to the **CNN** architecture applied to the **FMNIST** dataset. The CNN was pruned with a 99% pruning ratio.

## C.3 CONFUSION MATRICES FOR PAIRWISE MERGES

We show representative confusion matrices for pairwise merges of submodules, each specialized on a distinct class. These examples illustrate that, even after weight summation, the merged model retains high confidence on rewarded classes and preserves uniform predictions on others—confirming compatibility and lack of interference.

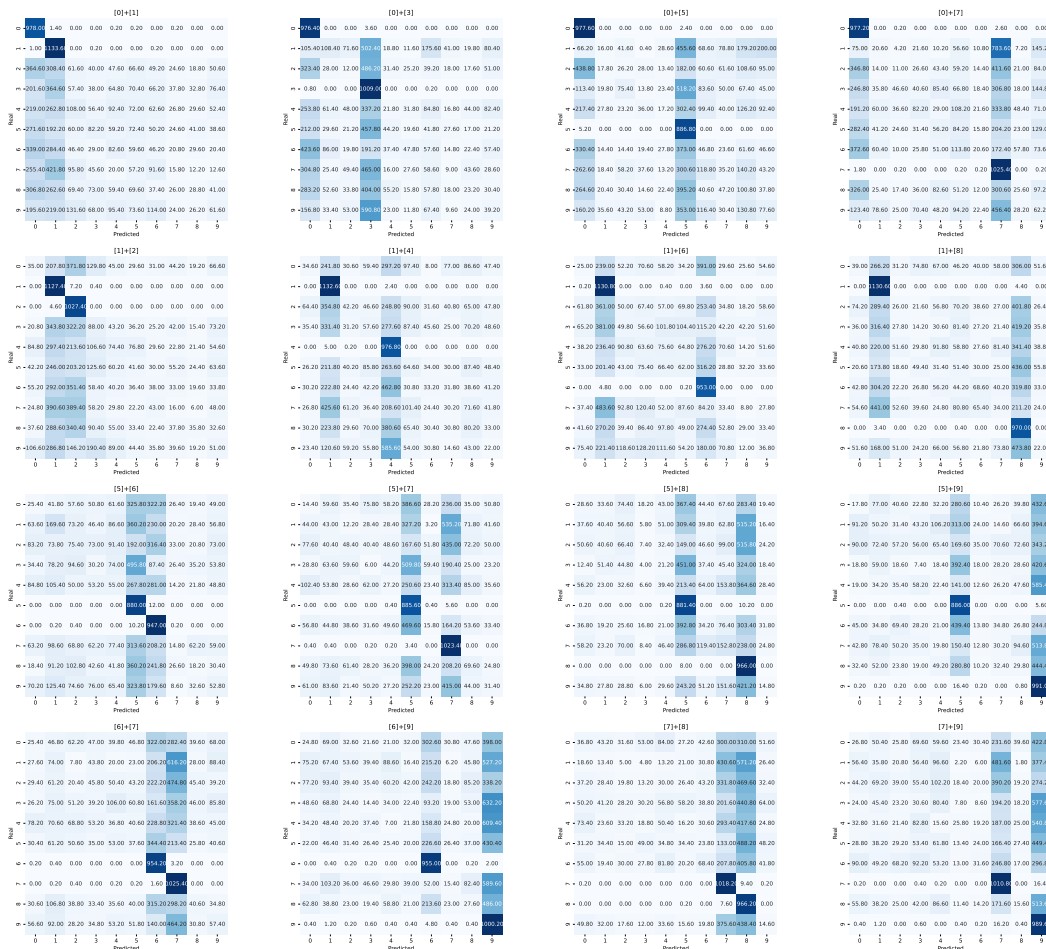

Figure 16: Confusion matrices (total predicted counts per class) for 16 pairwise merges of **Shallow MLP** submodules on **MNIST**. Each cell shows the average across 5 seeds.

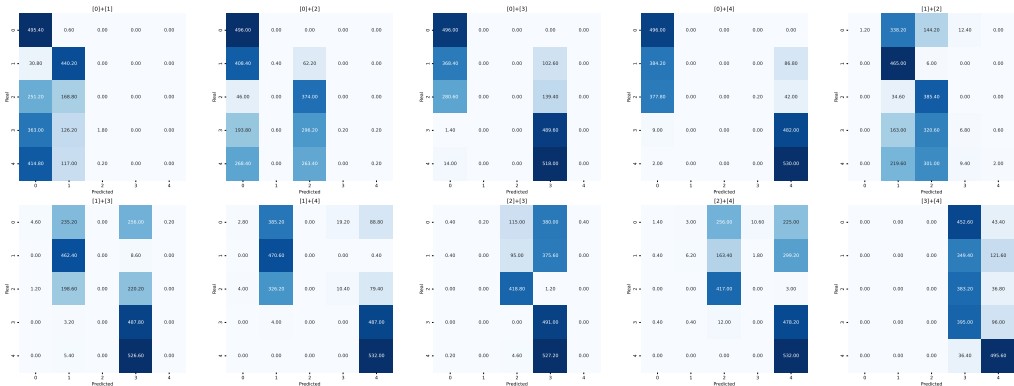

Figure 17: Confusion matrices (total predicted counts per class) for 10 pairwise merges of **Deep MLP** submodules on **HAR**. Each cell shows the average across 5 seeds.

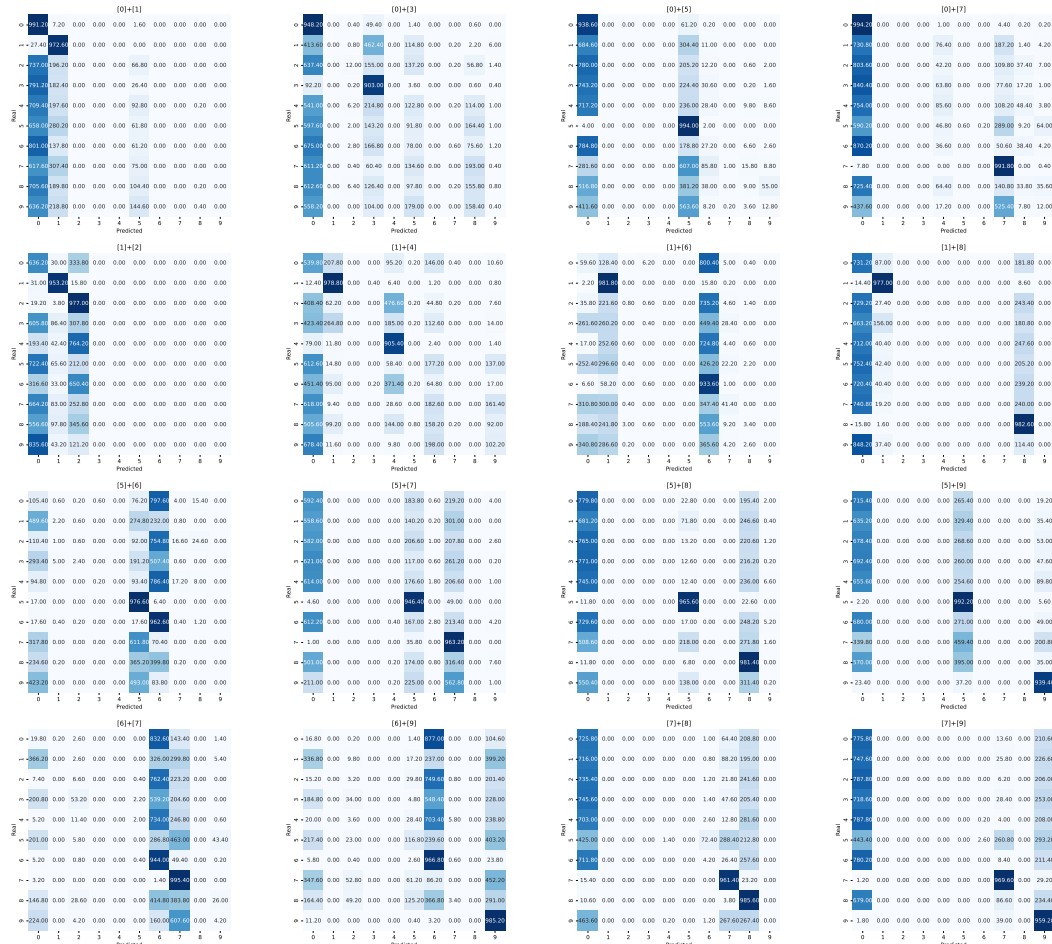

Figure 18: Confusion matrices (total predicted counts per class) for 16 pairwise merges of **CNN** submodules on **FMNIST**. Each cell shows the average across 5 seeds. The CNN was pruned with a 99% pruning ratio.

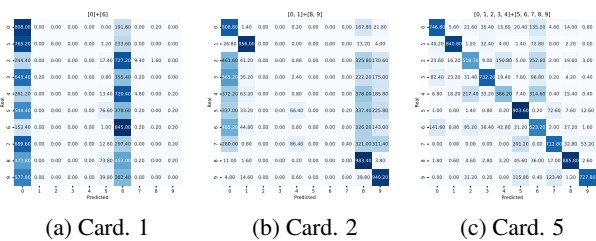

(a) Card. 1          (b) Card. 2          (c) Card. 5

Figure 19: Confusion matrices (total predicted counts per class) for pairwise merges of **CNN** submodules on **FMNIST**, for exemplary modules of cardinality 1, 2, and 5 each. Each cell shows the average across 5 seeds. The CNN was pruned with a 99% pruning ratio.

## C.4 CONFUSION MATRICES FOR INCREMENTAL MERGES

This section presents confusion matrices from various stages of incremental merging, where multiple submodules are combined one by one. The results demonstrate how compositional behavior is maintained across merge steps, and how the model continues to correctly isolate rewarded class behavior while suppressing predictions on excluded ones

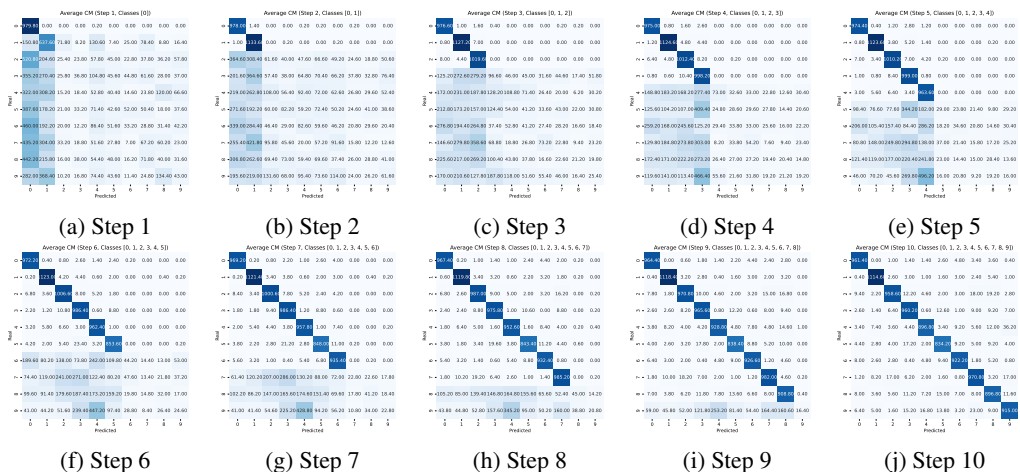

Figure 20: Confusion matrices (total predicted counts per class) for incremental merge steps of a **Shallow MLP** on **MNIST**. Each submodule is trained independently and merged following increasing order. Results are averaged over 5 seeds. The structure of predictions remains clean and interpretable throughout the merge.

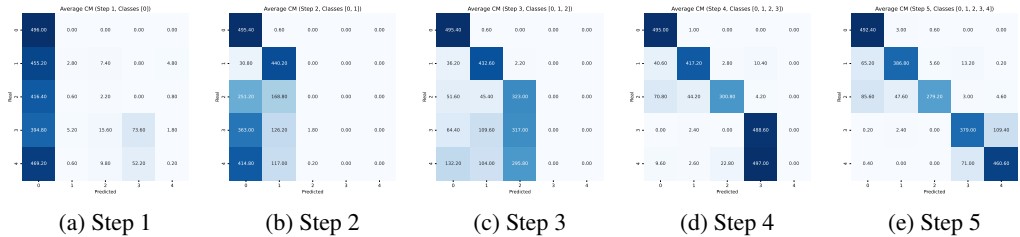

Figure 21: Confusion matrices (total predicted counts per class) for incremental merge steps of a **Deep MLP** on **HAR**. Each submodule is trained independently and merged following increasing order. Results are averaged over 5 seeds.

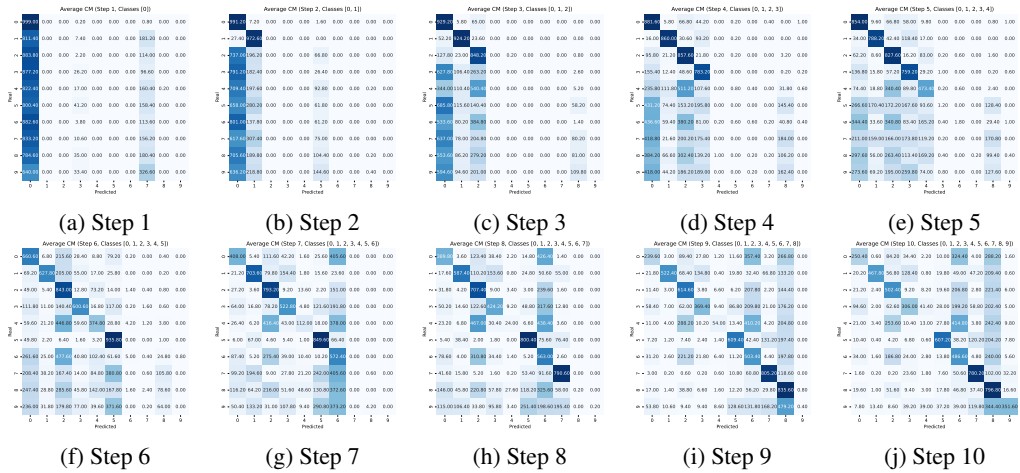

(a) Step 1  (b) Step 2  (c) Step 3  (d) Step 4  (e) Step 5

(f) Step 6  (g) Step 7  (h) Step 8  (i) Step 9  (j) Step 10

Figure 22: Confusion matrices (total predicted counts per class) for incremental merge steps of a **CNN** on **FMNIST**. Each submodule is trained independently and merged following increasing order. Results are averaged over 5 seeds. The CNN was pruned with a 99% pruning ratio.

### C.5 MODE CONNECTIVITY

In this section, we present additional plots analyzing the loss landscape between merged submodules. Following the framework described in Appendix B.4, we evaluate the relative difference in loss along the piecewise linear path connecting two modules and their sum.

We show incremental merging results for a **Deep MLP** on the **HAR** dataset and for a **CNN** on **FMNIST**, for both incremental and pairwise merging settings. In the majority of the cases, the pictures describe the desired behaviour (no or limited increase above 0). As already mentioned in the main paper, it can also be observed that for the CNN, after many incremental merges (e.g., steps 9 and 10), the difficulty in achieving an optimal merge is reflected in the presence of barriers in the loss function landscape.

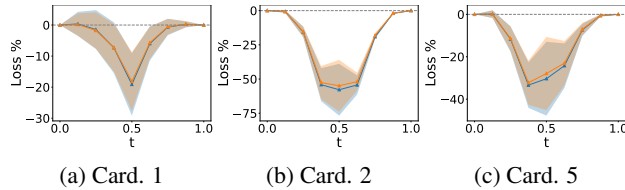

Figure 23: Values for $\gamma_{\theta_1 \to \theta_2}(t)$ for pairwise merges of **CNN** submodules on **FMNIST**, for exemplary modules of cardinality 1, 2, and 5 each (same as Figure 19). Results are averaged over 5 seeds. The CNN was pruned with a 99% pruning ratio.

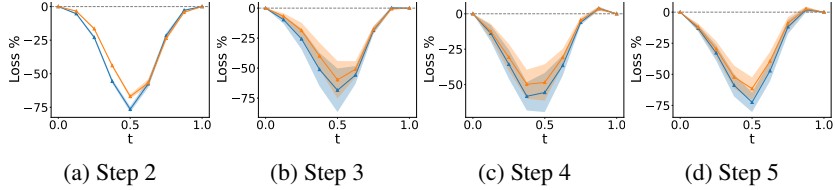

Figure 24: Values for $\gamma_{\theta_1 \to \theta_2}(t)$ for incremental merge steps of a **Deep MLP** on **HAR** (same as Figure 21). Each submodule is trained independently and merged following increasing order. Results are averaged over 5 seeds.

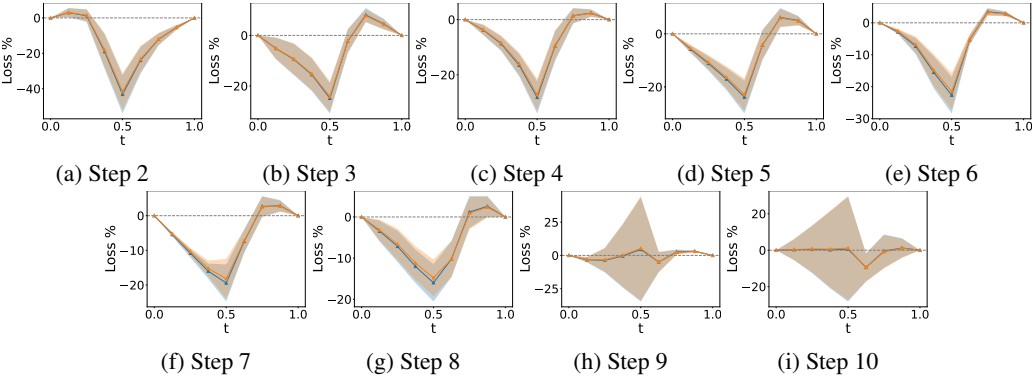

Figure 25: Values for $\gamma_{\theta_1 \to \theta_2}(t)$ for incremental merge steps of a **CNN** on **FMNIST** (same as Figure 22). Each submodule is trained independently and merged following increasing order. Results are averaged over 5 seeds. The CNN was pruned with a 99% pruning ratio.

### C.6 COMPLETE RESULTS FOR PAIRWISE SUBMODULE MERGING

In the main paper (Section 4), we report the results of pairwise submodule merging under the Iterative Magnitude Pruning (IMP) regime, as this generally led to the best overall performance. For completeness, here we provide the full set of results, including both pruned and non-pruned configurations. Table 2 reports the average rewarded accuracy and average entropy on non-rewarded classes for all model families (Shallow MLP, Deep MLP, CNN), across multiple datasets (MNIST, FMNIST, HAR, Yeast), cardinalities $|\mathcal{R}| \in 1, 2, 5$, and loss functions (CrossEntropy, Quasi-MaxEnt, and MaxEnt).

We observe that:

- Models trained with IMP consistently outperform their non-pruned counterparts in most settings, particularly when using the MaxEnt loss.

- The MaxEnt loss yields the most reliable and composable modules across all model types and datasets, both with and without pruning.

- Non-pruned models trained with standard CrossEntropy struggle to maintain entropy on non-rewarded classes, often resulting in degraded compositional performance.

- The benefit of pruning is especially marked in deep MLPs, where without IMP the merging process suffers from significant degradation in rewarded accuracy.

These extended results support the conclusions drawn in the main paper: pruning strengthens functional specialization and enhances composability, especially when combined with entropy-based objectives.

Table 4: Performance of pairwise-merged sub-modules with Shallow MLP and Deep MLP on FM-NIST, MNIST, HAR and Yeast, merged either via logit averaging or weight summation, across cardinalities $|\mathcal{R}|$, and loss functions: CrossEntropy (XE), MaxEnt (ME), and Quasi-MaxEnt (QME), with and without Iterative Magnitude Pruning (IMP). Results are averaged over 5 seeds and 10 pairs.

| Model | $|\mathcal{R}|$ | IMP | loss | Merge | FMNIST Entropy | FMNIST Rewarded Acc | MNIST Entropy | MNIST Rewarded Acc | HAR Entropy | HAR Rewarded Acc | Yeast Entropy | Yeast Rewarded Acc |
|---|---|---|---|---|---|---|---|---|---|---|---|---|
| Shallow MLP | 1 | No | XE | Logit | 0.308 (0.280) | 0.852 (0.197) | 0.580 (0.085) | 0.806 (0.156) | 0.582 (0.616) | 0.893 (0.157) | 1.099 (0.077) | 0.692 (0.042) |
| | | | | Weight | 0.045 (0.039) | 0.723 (0.219) | 0.001 (0.003) | 0.492 (0.024) | 0.318 (0.399) | 0.811 (0.235) | 0.176 (0.246) | 0.559 (0.146) |
| | | | ME | Logit | 2.294 (0.004) | 0.984 (0.027) | 2.299 (0.001) | 0.994 (0.003) | 1.592 (0.012) | 0.984 (0.021) | 1.349 (0.013) | 0.862 (0.118) |
| | | | | Weight | 2.295 (0.006) | 0.800 (0.116) | 2.300 (0.002) | 0.969 (0.014) | 1.512 (0.052) | 0.966 (0.033) | 0.804 (0.190) | 0.829 (0.150) |
| | | | QME | Logit | 2.107 (0.009) | 0.962 (0.033) | 2.105 (0.011) | 0.986 (0.004) | 1.160 (0.030) | 0.955 (0.020) | 1.259 (0.058) | 0.792 (0.090) |
| | | | | Weight | 2.038 (0.060) | 0.909 (0.088) | 2.040 (0.091) | 0.968 (0.021) | 0.902 (0.167) | 0.974 (0.041) | 0.828 (0.092) | 0.792 (0.096) |
| | | Yes | XE | Logit | 0.561 (0.451) | 0.859 (0.181) | 0.667 (0.043) | 0.798 (0.162) | 0.836 (0.491) | 0.882 (0.164) | 1.384 (0.001) | 0.667 (0.100) |
| | | | | Weight | 0.269 (0.294) | 0.813 (0.209) | 0.452 (0.159) | 0.679 (0.188) | 0.571 (0.625) | 0.866 (0.181) | 1.367 (0.011) | 0.660 (0.094) |
| | | | ME | Logit | 2.294 (0.003) | 0.983 (0.029) | 2.299 (0.001) | 0.992 (0.004) | 1.590 (0.011) | 0.982 (0.026) | 1.366 (0.005) | 0.844 (0.115) |
| | | | | Weight | 2.279 (0.011) | 0.980 (0.031) | 2.288 (0.004) | 0.991 (0.005) | 1.555 (0.029) | 0.983 (0.026) | 1.287 (0.036) | 0.843 (0.116) |
| | | | QME | Logit | 2.109 (0.010) | 0.962 (0.035) | 2.102 (0.009) | 0.984 (0.006) | 1.183 (0.035) | 0.955 (0.023) | 1.366 (0.007) | 0.734 (0.127) |
| | | | | Weight | 2.047 (0.127) | 0.868 (0.147) | 2.081 (0.004) | 0.973 (0.011) | 1.076 (0.056) | 0.954 (0.041) | 1.300 (0.024) | 0.737 (0.132) |
| | 2 | No | XE | Logit | 0.382 (0.138) | 0.685 (0.135) | 0.662 (0.102) | 0.771 (0.087) | 0.356 (0.268) | 0.918 (0.032) | – | 0.598 (0.010) |
| | | | | Weight | 0.064 (0.053) | 0.555 (0.076) | 0.133 (0.075) | 0.564 (0.085) | 0.152 (0.137) | 0.772 (0.189) | – | 0.437 (0.003) |
| | | | ME | Logit | 2.282 (0.005) | 0.963 (0.022) | 2.295 (0.001) | 0.988 (0.004) | 1.570 (0.022) | 0.953 (0.010) | – | 0.611 (0.007) |
| | | | | Weight | 2.288 (0.007) | 0.734 (0.077) | 2.297 (0.002) | 0.957 (0.011) | 1.521 (0.062) | 0.945 (0.015) | – | 0.549 (0.009) |
| | | | QME | Logit | 2.108 (0.012) | 0.955 (0.020) | 2.099 (0.009) | 0.987 (0.004) | 1.129 (0.036) | 0.952 (0.012) | – | 0.583 (0.011) |
| | | | | Weight | 1.999 (0.072) | 0.881 (0.058) | 1.696 (0.252) | 0.954 (0.022) | 0.286 (0.257) | 0.929 (0.036) | – | 0.556 (0.012) |
| | | Yes | XE | Logit | 0.610 (0.183) | 0.813 (0.093) | 0.714 (0.124) | 0.852 (0.070) | 0.691 (0.400) | 0.870 (0.068) | – | 0.439 (0.056) |
| | | | | Weight | 0.309 (0.110) | 0.777 (0.111) | 0.334 (0.068) | 0.831 (0.067) | 0.411 (0.318) | 0.864 (0.059) | – | 0.419 (0.039) |
| | | | ME | Logit | 2.283 (0.004) | 0.960 (0.023) | 2.290 (0.002) | 0.983 (0.005) | 1.568 (0.016) | 0.949 (0.014) | – | 0.565 (0.065) |
| | | | | Weight | 2.252 (0.012) | 0.952 (0.025) | 2.264 (0.007) | 0.980 (0.006) | 1.509 (0.034) | 0.945 (0.016) | – | 0.549 (0.064) |
| | | | QME | Logit | 2.114 (0.012) | 0.953 (0.021) | 2.101 (0.011) | 0.982 (0.006) | 1.149 (0.061) | 0.937 (0.010) | – | 0.474 (0.018) |
| | | | | Weight | 1.773 (0.341) | 0.918 (0.043) | 1.698 (0.180) | 0.963 (0.015) | 0.577 (0.344) | 0.919 (0.020) | – | 0.475 (0.021) |
| | 5 | No | XE | Logit | – | 0.516 (0.045) | – | 0.849 (0.021) | – | – | – | – |
| | | | | Weight | – | 0.451 (0.061) | – | 0.554 (0.056) | – | – | – | – |
| | | | ME | Logit | – | 0.883 (0.003) | – | 0.974 (0.001) | – | – | – | – |
| | | | | Weight | – | 0.681 (0.037) | – | 0.937 (0.017) | – | – | – | – |
| | | | QME | Logit | – | 0.877 (0.004) | – | 0.972 (0.001) | – | – | – | – |
| | | | | Weight | – | 0.713 (0.085) | – | 0.891 (0.025) | – | – | – | – |
| | | Yes | XE | Logit | – | 0.655 (0.021) | – | 0.855 (0.006) | – | – | – | – |
| | | | | Weight | – | 0.434 (0.084) | – | 0.831 (0.018) | – | – | – | – |
| | | | ME | Logit | – | 0.867 (0.002) | – | 0.952 (0.001) | – | – | – | – |
| | | | | Weight | – | 0.827 (0.014) | – | 0.945 (0.001) | – | – | – | – |
| | | | QME | Logit | – | 0.865 (0.002) | – | 0.950 (0.001) | – | – | – | – |
| | | | | Weight | – | 0.752 (0.036) | – | 0.909 (0.012) | – | – | – | – |
| Deep MLP | 1 | No | XE | Logit | 0.151 (0.175) | 0.864 (0.184) | 0.420 (0.128) | 0.780 (0.159) | 0.419 (0.512) | 0.888 (0.167) | 0.631 (0.110) | 0.724 (0.082) |
| | | | | Weight | 0.016 (0.034) | 0.544 (0.115) | 0.000 (0.000) | 0.490 (0.022) | 0.218 (0.277) | 0.797 (0.250) | 0.019 (0.051) | 0.509 (0.169) |
| | | | ME | Logit | 2.294 (0.003) | 0.982 (0.028) | 2.300 (0.001) | 0.994 (0.003) | 1.590 (0.012) | 0.977 (0.022) | 1.349 (0.014) | 0.859 (0.126) |
| | | | | Weight | 2.293 (0.015) | 0.886 (0.138) | 2.300 (0.004) | 0.833 (0.123) | 1.305 (0.277) | 0.979 (0.033) | 0.865 (0.157) | 0.850 (0.132) |
| | | | QME | Logit | 2.108 (0.010) | 0.960 (0.032) | 2.112 (0.011) | 0.983 (0.006) | 1.156 (0.030) | 0.949 (0.021) | 1.236 (0.062) | 0.777 (0.084) |
| | | | | Weight | 1.836 (0.152) | 0.916 (0.107) | 1.996 (0.114) | 0.975 (0.023) | 0.792 (0.175) | 0.965 (0.036) | 0.594 (0.104) | 0.787 (0.109) |
| | | Yes | XE | Logit | 0.460 (0.403) | 0.889 (0.150) | 0.656 (0.029) | 0.834 (0.128) | 0.731 (0.553) | 0.887 (0.160) | 1.386 (0.000) | 0.689 (0.095) |
| | | | | Weight | 0.216 (0.181) | 0.813 (0.212) | 0.486 (0.149) | 0.668 (0.176) | 0.543 (0.633) | 0.840 (0.214) | 1.376 (0.008) | 0.674 (0.097) |
| | | | ME | Logit | 2.298 (0.002) | 0.978 (0.034) | 2.301 (0.000) | 0.992 (0.004) | 1.593 (0.013) | 0.982 (0.029) | 1.356 (0.008) | 0.844 (0.117) |
| | | | | Weight | 2.237 (0.073) | 0.973 (0.040) | 2.294 (0.009) | 0.991 (0.005) | 1.081 (0.432) | 0.982 (0.030) | 1.201 (0.072) | 0.839 (0.119) |
| | | | QME | Logit | 2.108 (0.011) | 0.956 (0.039) | 2.098 (0.008) | 0.971 (0.009) | 1.166 (0.034) | 0.943 (0.026) | 1.334 (0.028) | 0.800 (0.113) |
| | | | | Weight | 1.844 (0.282) | 0.858 (0.152) | 2.062 (0.028) | 0.960 (0.029) | 0.779 (0.237) | 0.923 (0.091) | 1.178 (0.077) | 0.807 (0.112) |
| | 2 | No | XE | Logit | 0.271 (0.095) | 0.655 (0.133) | 0.564 (0.108) | 0.799 (0.067) | 0.380 (0.338) | 0.794 (0.150) | – | 0.527 (0.023) |
| | | | | Weight | 0.066 (0.045) | 0.622 (0.116) | 0.133 (0.087) | 0.580 (0.123) | 0.141 (0.157) | 0.668 (0.169) | – | 0.438 (0.035) |
| | | | ME | Logit | 2.284 (0.005) | 0.964 (0.022) | 2.297 (0.002) | 0.989 (0.003) | 1.569 (0.011) | 0.948 (0.011) | – | 0.619 (0.013) |
| | | | | Weight | 2.291 (0.014) | 0.798 (0.087) | 2.295 (0.007) | 0.685 (0.082) | 1.219 (0.214) | 0.935 (0.033) | – | 0.619 (0.006) |
| | | | QME | Logit | 2.113 (0.011) | 0.954 (0.020) | 2.107 (0.009) | 0.986 (0.004) | 1.134 (0.028) | 0.949 (0.016) | – | 0.601 (0.021) |
| | | | | Weight | 1.966 (0.256) | 0.862 (0.066) | 1.825 (0.197) | 0.900 (0.075) | 0.238 (0.187) | 0.919 (0.044) | – | 0.582 (0.019) |
| | | Yes | XE | Logit | 0.435 (0.168) | 0.790 (0.098) | 0.498 (0.100) | 0.841 (0.060) | 0.601 (0.369) | 0.857 (0.068) | – | 0.400 (0.031) |
| | | | | Weight | 0.226 (0.116) | 0.748 (0.121) | 0.194 (0.059) | 0.724 (0.085) | 0.400 (0.255) | 0.821 (0.065) | – | 0.372 (0.027) |
| | | | ME | Logit | 2.290 (0.003) | 0.955 (0.024) | 2.298 (0.001) | 0.983 (0.005) | 1.570 (0.023) | 0.942 (0.014) | – | 0.608 (0.013) |
| | | | | Weight | 2.225 (0.064) | 0.930 (0.030) | 2.272 (0.040) | 0.975 (0.010) | 0.672 (0.541) | 0.870 (0.098) | – | 0.598 (0.015) |
| | | | QME | Logit | 2.118 (0.011) | 0.947 (0.023) | 2.100 (0.011) | 0.979 (0.007) | 1.120 (0.049) | 0.926 (0.020) | – | 0.485 (0.024) |
| | | | | Weight | 1.347 (0.469) | 0.846 (0.113) | 1.158 (0.368) | 0.910 (0.055) | 0.205 (0.192) | 0.868 (0.085) | – | 0.486 (0.026) |
| | 5 | No | XE | Logit | – | 0.586 (0.070) | – | 0.882 (0.014) | – | – | – | – |
| | | | | Weight | – | 0.556 (0.042) | – | 0.578 (0.133) | – | – | – | – |
| | | | ME | Logit | – | 0.882 (0.005) | – | 0.975 (0.002) | – | – | – | – |
| | | | | Weight | – | 0.485 (0.067) | – | 0.639 (0.047) | – | – | – | – |
| | | | QME | Logit | – | 0.881 (0.003) | – | 0.975 (0.002) | – | – | – | – |
| | | | | Weight | – | 0.601 (0.090) | – | 0.784 (0.039) | – | – | – | – |
| | | Yes | XE | Logit | – | 0.624 (0.014) | – | 0.799 (0.017) | – | – | – | – |
| | | | | Weight | – | 0.500 (0.094) | – | 0.687 (0.069) | – | – | – | – |
| | | | ME | Logit | – | 0.852 (0.002) | – | 0.944 (0.003) | – | – | – | – |
| | | | | Weight | – | 0.800 (0.018) | – | 0.919 (0.008) | – | – | – | – |
| | | | QME | Logit | – | 0.850 (0.003) | – | 0.941 (0.003) | – | – | – | – |
| | | | | Weight | – | 0.649 (0.068) | – | 0.807 (0.040) | – | – | – | – |

Table 5: Performance of pairwise-merged sub-modules with CNN on FMNIST and MNIST, merged either via logit averaging or weight summation, across cardinalities $|\mathcal{R}|$, and loss functions: CrossEntropy (XE), MaxEnt (ME), and Quasi-MaxEnt (QME), with and without Iterative Magnitude Pruning (IMP). Results are averaged over 5 seeds and 10 pairs.

| Model | $\|\mathcal{R}\|$ | IMP | loss | Merge | FMNIST Entropy | FMNIST Rewarded Acc | MNIST Entropy | MNIST Rewarded Acc |
|---|---|---|---|---|---|---|---|---|
| CNN | 1 | No | XE | Logit | 0.240 (0.204) | 0.588 (0.158) | 0.338 (0.262) | 0.517 (0.083) |
| | | | | Weight | 0.045 (0.097) | 0.592 (0.175) | 0.077 (0.172) | 0.517 (0.064) |
| | | | ME | Logit | 2.296 (0.003) | 0.987 (0.021) | 2.301 (0.000) | 0.997 (0.002) |
| | | | | Weight | 2.226 (0.107) | 0.950 (0.056) | 2.102 (0.273) | 0.977 (0.027) |
| | | | QME | Logit | 2.106 (0.010) | 0.969 (0.028) | 2.097 (0.006) | 0.992 (0.003) |
| | | | | Weight | 1.272 (0.324) | 0.906 (0.098) | 1.210 (0.301) | 0.970 (0.047) |
| | | Yes | XE | Logit | 0.151 (0.151) | 0.633 (0.182) | 0.296 (0.254) | 0.539 (0.103) |
| | | | | Weight | 0.022 (0.043) | 0.576 (0.158) | 0.023 (0.104) | 0.520 (0.070) |
| | | | ME | Logit | 2.294 (0.004) | 0.983 (0.024) | 2.301 (0.001) | 0.997 (0.002) |
| | | | | Weight | 2.189 (0.112) | 0.966 (0.037) | 2.021 (0.357) | 0.984 (0.020) |
| | | | QME | Logit | 2.091 (0.009) | 0.955 (0.031) | 2.085 (0.003) | 0.986 (0.007) |
| | | | | Weight | 1.096 (0.287) | 0.896 (0.131) | 0.969 (0.371) | 0.952 (0.081) |
| | 2 | No | XE | Logit | 0.354 (0.164) | 0.603 (0.122) | 0.630 (0.165) | 0.753 (0.109) |
| | | | | Weight | 0.081 (0.050) | 0.664 (0.164) | 0.095 (0.050) | 0.711 (0.130) |
| | | | ME | Logit | 2.287 (0.004) | 0.973 (0.017) | 2.299 (0.001) | 0.995 (0.002) |
| | | | | Weight | 2.147 (0.093) | 0.894 (0.065) | 1.981 (0.204) | 0.948 (0.040) |
| | | | QME | Logit | 2.110 (0.013) | 0.964 (0.017) | 2.094 (0.006) | 0.993 (0.002) |
| | | | | Weight | 1.188 (0.318) | 0.909 (0.055) | 0.743 (0.376) | 0.936 (0.061) |
| | | Yes | XE | Logit | 0.164 (0.126) | 0.568 (0.110) | 0.438 (0.165) | 0.675 (0.139) |
| | | | | Weight | 0.047 (0.037) | 0.630 (0.162) | 0.062 (0.032) | 0.638 (0.135) |
| | | | ME | Logit | 2.281 (0.007) | 0.972 (0.017) | 2.297 (0.002) | 0.994 (0.003) |
| | | | | Weight | 2.055 (0.190) | 0.926 (0.047) | 1.649 (0.466) | 0.964 (0.040) |
| | | | QME | Logit | 2.089 (0.010) | 0.959 (0.017) | 2.083 (0.003) | 0.992 (0.003) |
| | | | | Weight | 0.694 (0.372) | 0.898 (0.065) | 0.366 (0.260) | 0.945 (0.054) |
| | 5 | No | XE | Logit | – | 0.632 (0.045) | – | 0.830 (0.041) |
| | | | | Weight | – | 0.462 (0.034) | – | 0.708 (0.087) |
| | | | ME | Logit | – | 0.914 (0.002) | – | 0.988 (0.001) |
| | | | | Weight | – | 0.803 (0.055) | – | 0.915 (0.042) |
| | | | QME | Logit | – | 0.908 (0.002) | – | 0.987 (0.001) |
| | | | | Weight | – | 0.767 (0.043) | – | 0.884 (0.043) |
| | | Yes | XE | Logit | – | 0.625 (0.052) | – | 0.901 (0.025) |
| | | | | Weight | – | 0.493 (0.144) | – | 0.776 (0.061) |
| | | | ME | Logit | – | 0.915 (0.002) | – | 0.988 (0.001) |
| | | | | Weight | – | 0.791 (0.083) | – | 0.943 (0.034) |
| | | | QME | Logit | – | 0.912 (0.004) | – | 0.986 (0.000) |
| | | | | Weight | – | 0.790 (0.046) | – | 0.909 (0.034) |

Table 6: Performance of pairwise-merged sub-modules with ResNet18 on FMNIST, MNIST and CIFAR-10, merged either via logit averaging or weight summation, across cardinalities $|\mathcal{R}|$, and loss functions: CrossEntropy (XE), MaxEnt (ME), and Quasi-MaxEnt (QME), with and without Iterative Magnitude Pruning (IMP). Results are averaged over 5 seeds and 10 pairs.

| Model | $|\mathcal{R}|$ | IMP | loss | Merge | FMNIST Entropy | FMNIST Rewarded Acc | MNIST Entropy | MNIST Rewarded Acc | CIFAR-10 Entropy | CIFAR-10 Rewarded Acc |
|---|---|---|---|---|---|---|---|---|---|---|
| ResNet | 1 | No | XE | Logit | 0.804 (0.008) | 0.519 (0.114) | 0.804 (0.010) | 0.529 (0.088) | 0.705 (0.004) | 0.549 (0.046) |
| | | | | Weight | 0.388 (0.089) | 0.721 (0.127) | 0.423 (0.047) | 0.748 (0.083) | 0.650 (0.034) | 0.510 (0.028) |
| | | | ME | Logit | 2.296 (0.003) | 0.987 (0.024) | 2.301 (0.001) | 0.998 (0.002) | 2.287 (0.003) | 0.947 (0.050) |
| | | | | Weight | 2.208 (0.056) | 0.982 (0.028) | 2.282 (0.013) | 0.987 (0.054) | 1.803 (0.219) | 0.881 (0.096) |
| | | | QME | Logit | 2.132 (0.008) | 0.964 (0.031) | 2.124 (0.005) | 0.993 (0.003) | 2.184 (0.009) | 0.866 (0.050) |
| | | | | Weight | 2.030 (0.033) | 0.944 (0.050) | 2.072 (0.006) | 0.983 (0.013) | 1.979 (0.060) | 0.587 (0.097) |
| | | Yes | XE | Logit | 0.809 (0.008) | 0.536 (0.109) | 0.808 (0.008) | 0.482 (0.097) | 0.686 (0.006) | 0.547 (0.049) |
| | | | | Weight | 0.487 (0.073) | 0.691 (0.138) | 0.500 (0.064) | 0.656 (0.130) | 0.637 (0.043) | 0.508 (0.026) |
| | | | ME | Logit | 2.292 (0.006) | 0.987 (0.023) | 2.300 (0.005) | 0.997 (0.002) | 2.286 (0.006) | 0.953 (0.044) |
| | | | | Weight | 2.249 (0.037) | 0.878 (0.156) | 2.293 (0.007) | 0.987 (0.056) | 1.504 (0.309) | 0.812 (0.145) |
| | | | QME | Logit | 2.113 (0.005) | 0.948 (0.040) | 2.112 (0.004) | 0.991 (0.005) | 2.155 (0.009) | 0.862 (0.045) |
| | | | | Weight | 2.031 (0.029) | 0.929 (0.055) | 2.072 (0.004) | 0.983 (0.009) | 1.941 (0.102) | 0.375 (0.101) |
| | 2 | No | XE | Logit | 0.902 (0.122) | 0.644 (0.106) | 1.058 (0.154) | 0.816 (0.096) | 1.005 (0.069) | 0.724 (0.052) |
| | | | | Weight | 0.136 (0.046) | 0.808 (0.089) | 0.136 (0.028) | 0.868 (0.084) | 0.306 (0.106) | 0.528 (0.082) |
| | | | ME | Logit | 2.285 (0.006) | 0.972 (0.017) | 2.300 (0.001) | 0.996 (0.002) | 2.273 (0.006) | 0.927 (0.023) |
| | | | | Weight | 2.110 (0.080) | 0.964 (0.020) | 2.276 (0.016) | 0.991 (0.008) | 1.484 (0.305) | 0.681 (0.055) |
| | | | QME | Logit | 2.126 (0.011) | 0.962 (0.018) | 2.119 (0.006) | 0.994 (0.002) | 2.128 (0.015) | 0.905 (0.025) |
| | | | | Weight | 1.830 (0.097) | 0.936 (0.039) | 1.978 (0.062) | 0.983 (0.016) | 1.245 (0.241) | 0.561 (0.040) |
| | | Yes | XE | Logit | 0.842 (0.052) | 0.729 (0.108) | 0.908 (0.072) | 0.881 (0.069) | 0.963 (0.053) | 0.701 (0.053) |
| | | | | Weight | 0.101 (0.043) | 0.768 (0.099) | 0.113 (0.018) | 0.832 (0.092) | 0.565 (0.138) | 0.527 (0.075) |
| | | | ME | Logit | 2.276 (0.008) | 0.971 (0.018) | 2.299 (0.001) | 0.994 (0.002) | 2.269 (0.011) | 0.894 (0.027) |
| | | | | Weight | 2.164 (0.060) | 0.955 (0.025) | 2.286 (0.011) | 0.989 (0.007) | 1.619 (0.183) | 0.647 (0.058) |
| | | | QME | Logit | 2.101 (0.011) | 0.955 (0.018) | 2.108 (0.003) | 0.994 (0.002) | 2.146 (0.016) | 0.870 (0.035) |
| | | | | Weight | 1.806 (0.106) | 0.936 (0.027) | 1.833 (0.122) | 0.988 (0.005) | 1.406 (0.163) | 0.514 (0.045) |
| | 5 | No | XE | Logit | – | 0.701 (0.062) | – | 0.959 (0.008) | – | 0.740 (0.013) |
| | | | | Weight | – | 0.683 (0.112) | – | 0.876 (0.083) | – | 0.392 (0.023) |
| | | | ME | Logit | – | 0.906 (0.004) | – | 0.990 (0.001) | – | 0.852 (0.005) |
| | | | | Weight | – | 0.874 (0.009) | – | 0.977 (0.011) | – | 0.486 (0.028) |
| | | | QME | Logit | – | 0.904 (0.004) | – | 0.989 (0.001) | – | 0.846 (0.004) |
| | | | | Weight | – | 0.866 (0.020) | – | 0.887 (0.122) | – | 0.521 (0.019) |
| | | Yes | XE | Logit | – | 0.747 (0.031) | – | 0.958 (0.006) | – | 0.693 (0.023) |
| | | | | Weight | – | 0.788 (0.029) | – | 0.897 (0.071) | – | 0.362 (0.029) |
| | | | ME | Logit | – | 0.903 (0.005) | – | 0.991 (0.001) | – | 0.775 (0.036) |
| | | | | Weight | – | 0.861 (0.013) | – | 0.985 (0.002) | – | 0.328 (0.019) |
| | | | QME | Logit | – | 0.905 (0.004) | – | 0.990 (0.001) | – | 0.781 (0.010) |
| | | | | Weight | – | 0.861 (0.007) | – | 0.982 (0.002) | – | 0.350 (0.032) |

Table 7: Performance of pairwise-merged sub-modules with VGG on FMNIST, merged either via logit averaging or weight summation, across cardinalities $|\mathcal{R}|$, and loss functions: CrossEntropy (XE), MaxEnt (ME), and Quasi-MaxEnt (QME), with and without Iterative Magnitude Pruning (IMP). Results are averaged over 5 seeds and 10 pairs.

| Model | $|\mathcal{R}|$ | IMP | loss | Merge | FMNIST Entropy | Rewarded Acc |
|---|---|---|---|---|---|---|
| VGG | 1 | No | XE | Logit | 0.000 (0.000) | 0.501 (0.006) |
| | | | | Weight | 0.000 (0.000) | 0.500 (0.000) |
| | | | ME | Logit | 2.296 (0.003) | 0.986 (0.023) |
| | | | | Weight | 1.732 (0.712) | 0.938 (0.101) |
| | | | QME | Logit | 2.108 (0.017) | 0.971 (0.027) |
| | | | | Weight | 2.148 (0.059) | 0.713 (0.201) |
| | | Yes | XE | Logit | 0.275 (0.253) | 0.558 (0.138) |
| | | | | Weight | 0.000 (0.000) | 0.500 (0.000) |
| | | | ME | Logit | 2.293 (0.005) | 0.989 (0.023) |
| | | | | Weight | 2.174 (0.174) | 0.914 (0.116) |
| | | | QME | Logit | 2.095 (0.023) | 0.957 (0.035) |
| | | | | Weight | 2.004 (0.249) | 0.663 (0.214) |
| | 2 | No | XE | Logit | 0.143 (0.092) | 0.707 (0.133) |
| | | | | Weight | 0.271 (0.214) | 0.575 (0.138) |
| | | | ME | Logit | 2.287 (0.005) | 0.969 (0.018) |
| | | | | Weight | 2.047 (0.346) | 0.762 (0.089) |
| | | | QME | Logit | 2.112 (0.017) | 0.964 (0.017) |
| | | | | Weight | 2.101 (0.215) | 0.704 (0.121) |
| | | Yes | XE | Logit | 0.090 (0.058) | 0.690 (0.143) |
| | | | | Weight | 0.240 (0.201) | 0.524 (0.140) |
| | | | ME | Logit | 2.280 (0.008) | 0.971 (0.018) |
| | | | | Weight | 2.005 (0.274) | 0.689 (0.100) |
| | | | QME | Logit | 2.088 (0.011) | 0.959 (0.018) |
| | | | | Weight | 2.003 (0.235) | 0.611 (0.104) |
| | 5 | No | XE | Logit | – | 0.680 (0.056) |
| | | | | Weight | – | 0.452 (0.046) |
| | | | ME | Logit | – | 0.913 (0.003) |
| | | | | Weight | – | 0.432 (0.079) |
| | | | QME | Logit | – | 0.911 (0.003) |
| | | | | Weight | – | 0.411 (0.060) |
| | | Yes | XE | Logit | – | 0.628 (0.052) |
| | | | | Weight | – | 0.330 (0.052) |
| | | | ME | Logit | – | 0.912 (0.004) |
| | | | | Weight | – | 0.388 (0.121) |
| | | | QME | Logit | – | 0.911 (0.004) |
| | | | | Weight | – | 0.356 (0.083) |

Table 8: Performance of pairwise-merged sub-modules with VGG equipped with BatchNorm on FMNIST, merged either via logit averaging or weight summation, across cardinalities $|\mathcal{R}|$, and loss functions: CrossEntropy (XE), MaxEnt (ME), and Quasi-MaxEnt (QME), with and without Iterative Magnitude Pruning (IMP). Results are averaged over 5 seeds and 10 pairs.

| Model | $|\mathcal{R}|$ | IMP | loss | Merge | FMNIST Entropy | Rewarded Acc |
|---|---|---|---|---|---|---|
| VGGbn | 1 | No | XE | Logit | 0.003 (0.001) | 0.340 (0.096) |
| | | | | Weight | 0.011 (0.074) | 0.499 (0.007) |
| | | | ME | Logit | 2.296 (0.003) | 0.985 (0.020) |
| | | | | Weight | 2.238 (0.104) | 0.840 (0.165) |
| | | | QME | Logit | 2.106 (0.015) | 0.971 (0.027) |
| | | | | Weight | 1.605 (0.279) | 0.746 (0.173) |
| | | Yes | XE | Logit | 0.180 (0.071) | 0.180 (0.120) |
| | | | | Weight | 0.200 (0.230) | 0.505 (0.027) |
| | | | ME | Logit | 2.294 (0.006) | 0.987 (0.021) |
| | | | | Weight | 2.134 (0.250) | 0.940 (0.052) |
| | | | QME | Logit | 2.091 (0.008) | 0.956 (0.033) |
| | | | | Weight | 1.475 (0.358) | 0.781 (0.141) |
| | 2 | No | XE | Logit | 0.112 (0.057) | 0.746 (0.130) |
| | | | | Weight | 0.150 (0.118) | 0.688 (0.119) |
| | | | ME | Logit | 2.288 (0.005) | 0.966 (0.018) |
| | | | | Weight | 1.866 (0.577) | 0.744 (0.146) |
| | | | QME | Logit | 2.104 (0.012) | 0.964 (0.016) |
| | | | | Weight | 1.890 (0.196) | 0.710 (0.154) |
| | | Yes | XE | Logit | 0.056 (0.042) | 0.722 (0.124) |
| | | | | Weight | 0.034 (0.030) | 0.551 (0.124) |
| | | | ME | Logit | 2.282 (0.011) | 0.967 (0.028) |
| | | | | Weight | 1.898 (0.383) | 0.748 (0.132) |
| | | | QME | Logit | 2.091 (0.013) | 0.956 (0.021) |
| | | | | Weight | 1.795 (0.379) | 0.745 (0.152) |
| | 5 | No | XE | Logit | – | 0.691 (0.052) |
| | | | | Weight | – | 0.477 (0.145) |
| | | | ME | Logit | – | 0.910 (0.010) |
| | | | | Weight | – | 0.610 (0.102) |
| | | | QME | Logit | – | 0.911 (0.005) |
| | | | | Weight | – | 0.594 (0.097) |
| | | Yes | XE | Logit | – | 0.662 (0.027) |
| | | | | Weight | – | 0.524 (0.096) |
| | | | ME | Logit | – | 0.913 (0.005) |
| | | | | Weight | – | 0.583 (0.084) |
| | | | QME | Logit | – | 0.858 (0.124) |
| | | | | Weight | – | 0.525 (0.158) |

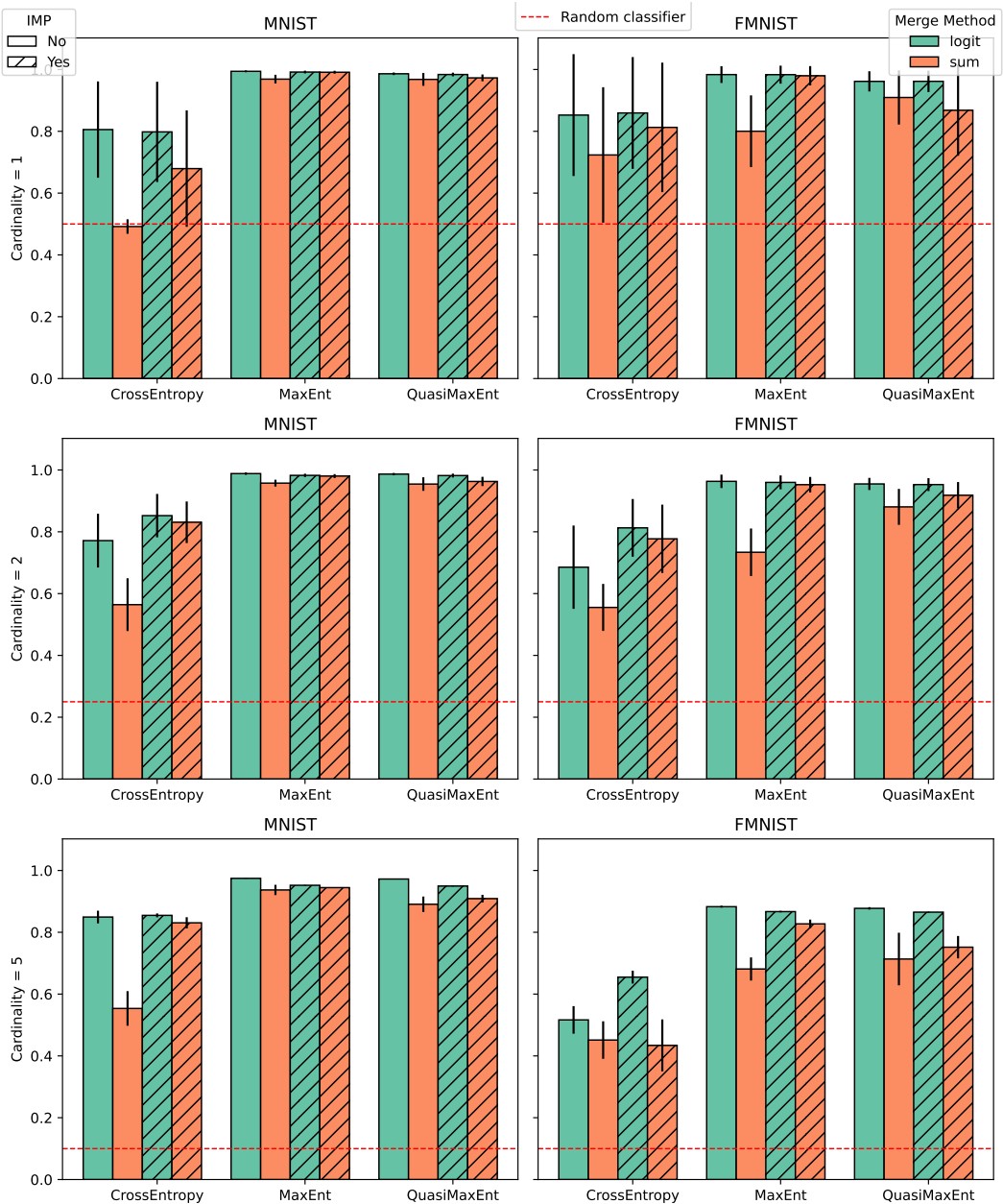

Figure 26: Rewarded accuracy of Shallow MLP on MNIST and FMNIST across cardinalities, loss functions, and merge strategies (logit vs. weight), with and without IMP.

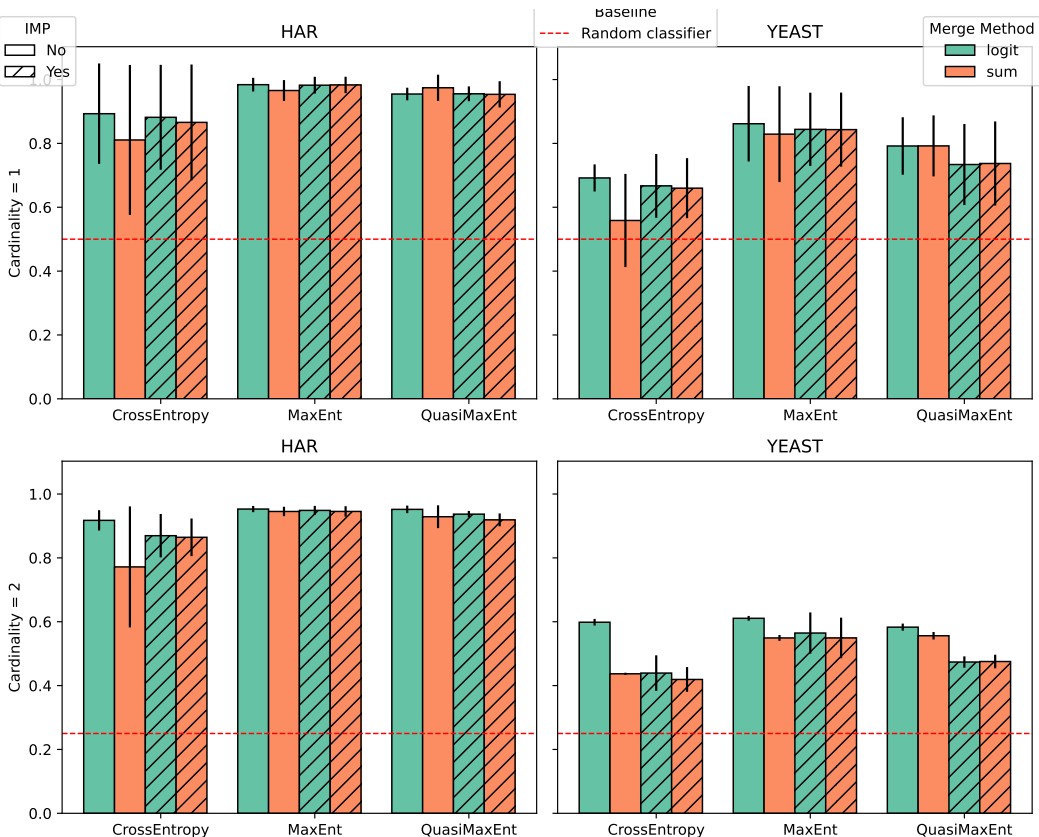

Figure 27: Rewarded accuracy of Shallow MLP on HAR and Yeast across cardinalities, loss functions, and merge strategies (logit vs. weight), with and without IMP.

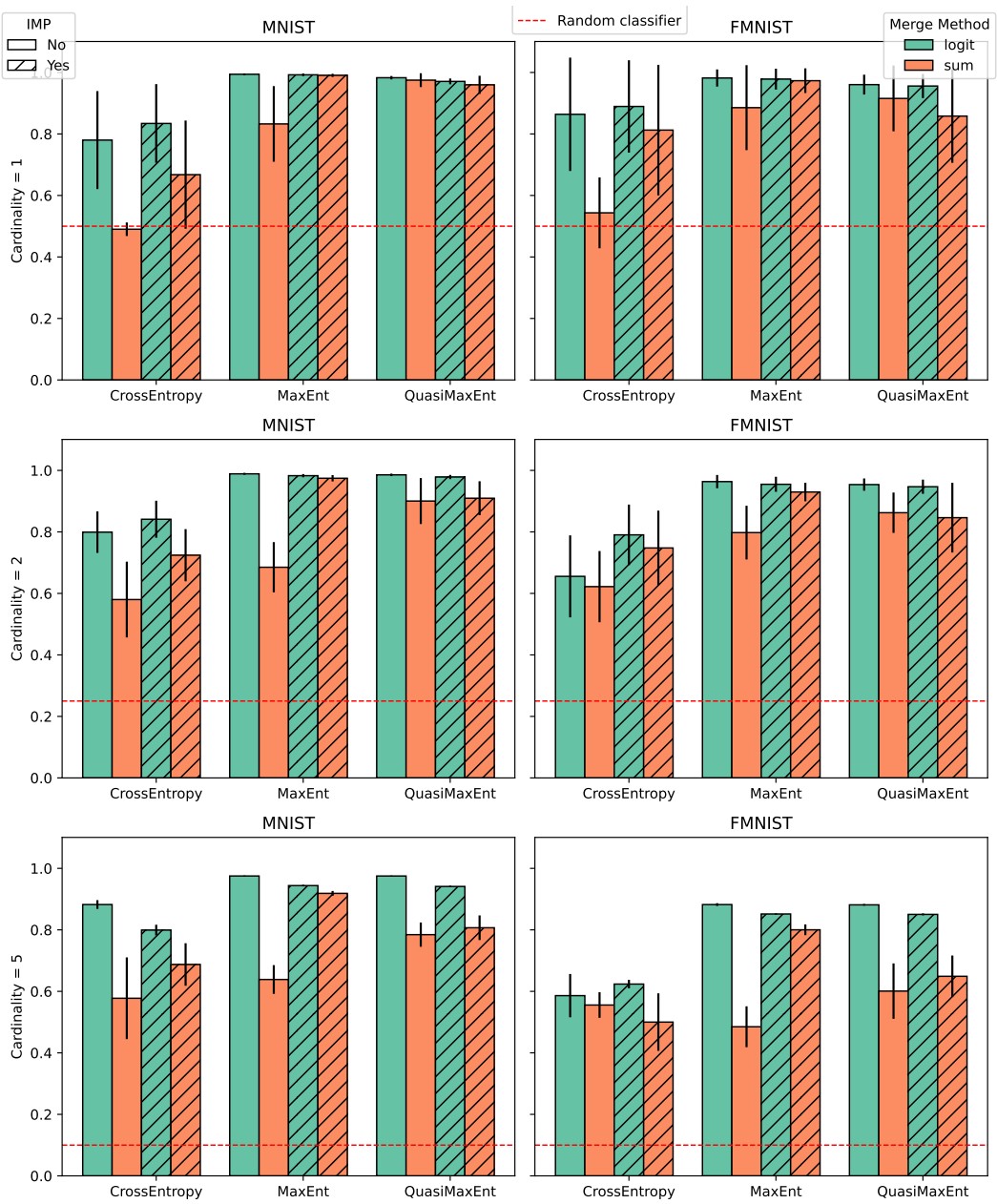

Figure 28: Rewarded accuracy of Deep MLP on MNIST and FMNIST across cardinalities, loss functions, and merge strategies (logit vs. weight), with and without IMP.

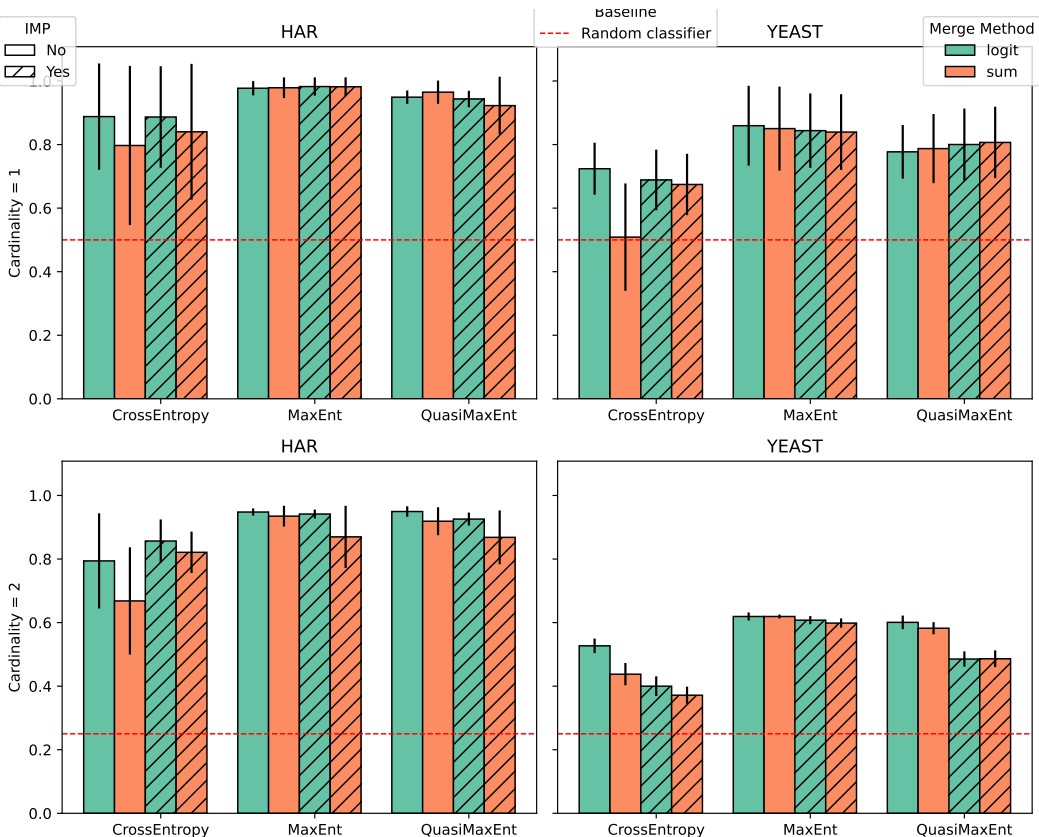

Figure 29: Rewarded accuracy of Deep MLP on HAR and Yeast across cardinalities, loss functions, and merge strategies (logit vs. weight), with and without IMP.

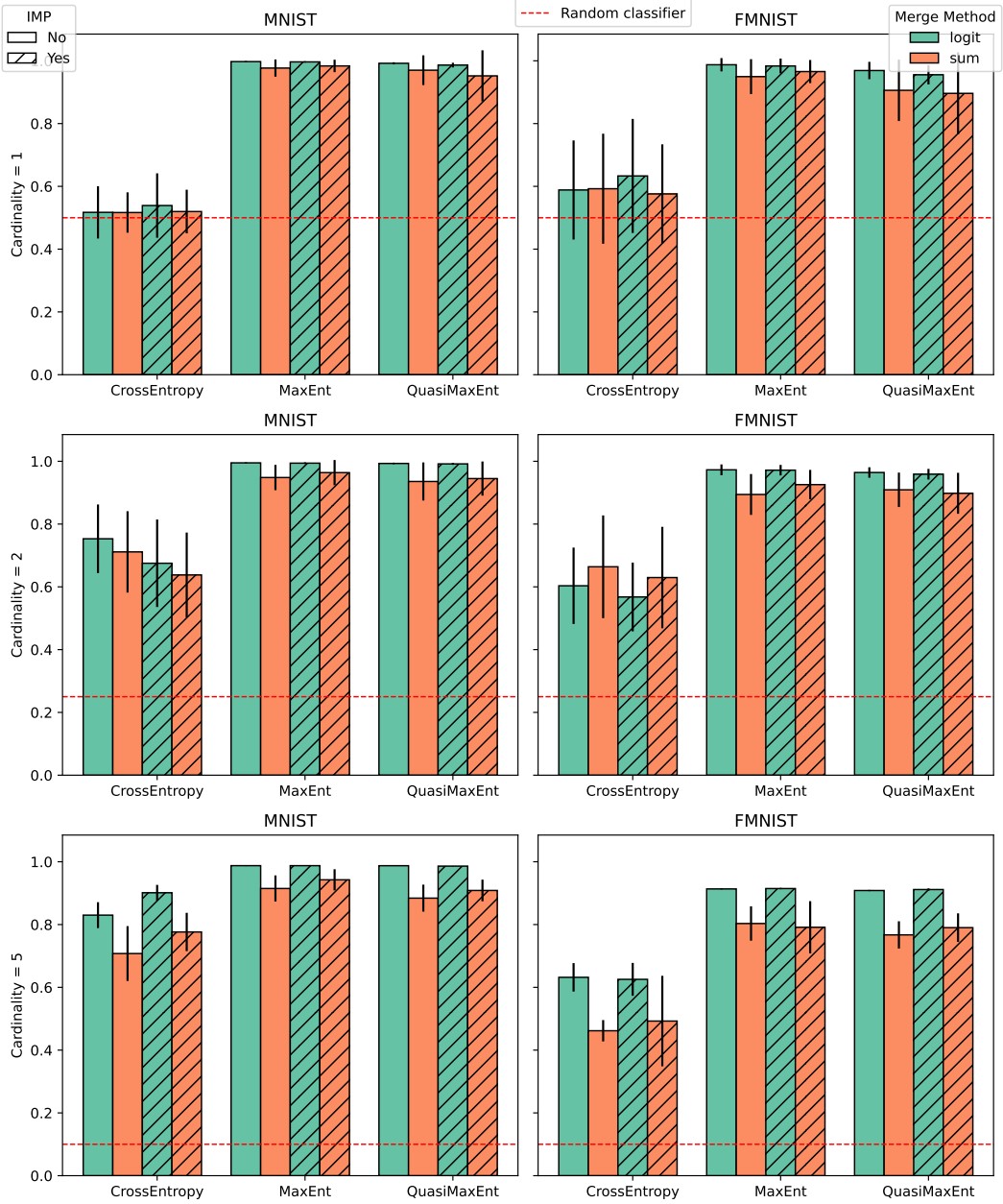

Figure 30: Rewarded accuracy of CNN on MNIST and FMNIST across cardinalities, loss functions, and merge strategies (logit vs. weight), with and without IMP.

## C.7 COMPLETE BASELINES

Table 9 reports the full comparison across all loss functions for different architectures, cardinalities, and datasets, with and without IMP. Across all settings, MaxEnt loss (ME) achieves the highest rewarded accuracy and the entropy closest to the theoretical maximum $\log(|\mathcal{C}|)$ on non-rewarded classes, confirming that its asymmetric treatment of rewarded and non-rewarded inputs is essential for isolating class-specific subnetworks. CrossEntropy with Confidence Penalty (CP) and with Label Smoothing (LS) consistently underperform ME, demonstrating that ME is not a mere re-parameterisation of standard regularisers.

Table 9: Average rewarded accuracy and non-rewarded entropy of pairwise-merged (weight-space) sub-modules across architectures, cardinalities $|\mathcal{R}|$, and loss functions: CrossEntropy (XE), Confidence Penalty (CP), Label Smoothing (LS), MaxEnt (ME), and Quasi-MaxEnt (QME), with and without IMP. Results are averaged over 5 seeds. ME consistently achieves the highest rewarded accuracy

| Model | $|\mathcal{R}|$ | IMP | loss | FMNIST Entropy | FMNIST Rewarded Acc | MNIST Entropy | MNIST Rewarded Acc | HAR Entropy | HAR Rewarded Acc | Yeast Entropy | Yeast Rewarded Acc |
|---|---|---|---|---|---|---|---|---|---|---|---|
| Shallow MLP | 1 | No | XE | 0.045 (0.039) | 0.723 (0.219) | 0.001 (0.003) | 0.492 (0.024) | 0.318 (0.399) | 0.811 (0.235) | 0.176 (0.246) | 0.559 (0.146) |
| | | | CP | 0.061 (0.040) | 0.753 (0.210) | 0.019 (0.029) | 0.506 (0.045) | 0.320 (0.401) | 0.817 (0.228) | 0.177 (0.243) | 0.539 (0.141) |
| | | | LS | 0.334 (0.272) | 0.603 (0.166) | 0.338 (0.161) | 0.519 (0.107) | – | – | 0.170 (0.224) | 0.515 (0.161) |
| | | | ME | 2.295 (0.006) | 0.800 (0.116) | 2.300 (0.002) | 0.969 (0.014) | 1.512 (0.052) | 0.966 (0.033) | 0.804 (0.190) | 0.829 (0.150) |
| | | | QME | 2.038 (0.060) | 0.909 (0.088) | 2.040 (0.091) | 0.968 (0.021) | 0.902 (0.167) | 0.974 (0.041) | 0.828 (0.092) | 0.792 (0.096) |
| | | Yes | XE | 0.269 (0.294) | 0.813 (0.209) | 0.452 (0.159) | 0.679 (0.188) | 0.571 (0.625) | 0.866 (0.181) | 1.367 (0.011) | 0.660 (0.094) |
| | | | CP | 0.306 (0.328) | 0.800 (0.217) | 0.600 (0.611) | 0.871 (0.175) | 0.504 (0.138) | 0.622 (0.161) | 1.368 (0.010) | 0.666 (0.091) |
| | | | LS | 0.786 (0.158) | 0.703 (0.204) | 0.946 (0.372) | 0.891 (0.148) | 0.742 (0.015) | 0.580 (0.137) | 1.368 (0.010) | 0.666 (0.095) |
| | | | ME | 2.279 (0.011) | 0.980 (0.031) | 2.288 (0.004) | 0.991 (0.005) | 1.555 (0.029) | 0.983 (0.026) | 1.287 (0.036) | 0.843 (0.116) |
| | | | QME | 2.047 (0.127) | 0.868 (0.147) | 2.081 (0.004) | 0.973 (0.011) | 1.076 (0.056) | 0.954 (0.041) | 1.300 (0.024) | 0.737 (0.132) |
| | 2 | No | XE | 0.064 (0.053) | 0.555 (0.076) | 0.133 (0.075) | 0.564 (0.085) | 0.152 (0.137) | 0.772 (0.189) | – | 0.437 (0.003) |
| | | | CP | 0.081 (0.058) | 0.571 (0.074) | 0.162 (0.093) | 0.577 (0.096) | 0.164 (0.150) | 0.780 (0.185) | – | 0.438 (0.004) |
| | | | LS | 0.268 (0.099) | 0.530 (0.045) | 0.749 (0.192) | 0.659 (0.094) | 0.257 (0.217) | 0.784 (0.198) | – | 0.437 (0.003) |
| | | | ME | 2.288 (0.007) | 0.734 (0.077) | 2.297 (0.002) | 0.957 (0.011) | 1.521 (0.062) | 0.945 (0.015) | – | 0.549 (0.009) |
| | | | QME | 1.999 (0.072) | 0.881 (0.058) | 1.696 (0.252) | 0.954 (0.022) | 0.286 (0.257) | 0.929 (0.036) | – | 0.556 (0.012) |
| | | Yes | XE | 0.309 (0.110) | 0.777 (0.111) | 0.334 (0.068) | 0.831 (0.067) | 0.411 (0.318) | 0.864 (0.059) | – | 0.419 (0.039) |
| | | | CP | 0.347 (0.123) | 0.773 (0.116) | 0.460 (0.339) | 0.863 (0.061) | 0.368 (0.075) | 0.822 (0.068) | – | 0.418 (0.026) |
| | | | LS | 0.778 (0.081) | 0.507 (0.152) | 0.912 (0.239) | 0.825 (0.038) | 0.887 (0.115) | 0.723 (0.063) | – | 0.418 (0.031) |
| | | | ME | 2.252 (0.012) | 0.952 (0.025) | 2.264 (0.007) | 0.980 (0.006) | 1.509 (0.034) | 0.945 (0.016) | – | 0.549 (0.064) |
| | | | QME | 1.773 (0.341) | 0.918 (0.043) | 1.698 (0.180) | 0.963 (0.015) | 0.577 (0.344) | 0.919 (0.020) | – | 0.475 (0.021) |
| | 5 | No | XE | – | 0.451 (0.061) | – | 0.554 (0.056) | – | – | – | – |
| | | | CP | – | 0.474 (0.028) | – | 0.580 (0.059) | – | – | – | – |
| | | | LS | – | 0.527 (0.059) | – | 0.690 (0.077) | – | – | – | – |
| | | | ME | – | 0.681 (0.037) | – | 0.937 (0.017) | – | – | – | – |
| | | | QME | – | 0.713 (0.085) | – | 0.891 (0.025) | – | – | – | – |
| | | Yes | XE | – | 0.434 (0.084) | – | 0.831 (0.018) | – | – | – | - |
| | | | CP | – | 0.438 (0.063) | – | 0.832 (0.007) | – | – | – | - |
| | | | LS | – | 0.521 (0.058) | – | 0.848 (0.017) | – | – | – | - |
| | | | ME | – | 0.827 (0.014) | – | 0.945 (0.001) | – | – | – | - |
| | | | QME | – | 0.752 (0.036) | – | 0.909 (0.012) | – | – | – | – |
| Deep MLP | 1 | No | XE | 0.016 (0.034) | 0.544 (0.115) | 0.000 (0.000) | 0.490 (0.022) | 0.218 (0.277) | 0.797 (0.250) | 0.019 (0.051) | 0.509 (0.169) |
| | | | CP | 0.022 (0.041) | 0.568 (0.140) | 0.000 (0.000) | 0.490 (0.022) | 0.253 (0.323) | 0.818 (0.236) | 0.020 (0.048) | 0.509 (0.169) |
| | | | LS | 0.260 (0.211) | 0.545 (0.114) | 0.333 (0.177) | 0.494 (0.032) | 0.220 (0.244) | 0.838 (0.215) | 0.181 (0.198) | 0.519 (0.155) |
| | | | ME | 2.293 (0.015) | 0.886 (0.138) | 2.300 (0.004) | 0.833 (0.123) | 1.305 (0.277) | 0.979 (0.031) | 0.865 (0.157) | 0.850 (0.132) |
| | | | QME | 1.836 (0.152) | 0.916 (0.107) | 1.996 (0.114) | 0.975 (0.023) | 0.792 (0.175) | 0.965 (0.036) | 0.594 (0.104) | 0.787 (0.109) |
| | | Yes | XE | 0.216 (0.181) | 0.813 (0.212) | 0.486 (0.149) | 0.668 (0.176) | 0.543 (0.633) | 0.840 (0.214) | 1.376 (0.008) | 0.674 (0.097) |
| | | | CP | 0.228 (0.178) | 0.788 (0.224) | 0.564 (0.622) | 0.850 (0.212) | 0.528 (0.148) | 0.682 (0.175) | 1.375 (0.009) | 0.669 (0.089) |
| | | | LS | 0.718 (0.154) | 0.655 (0.185) | 0.925 (0.361) | 0.865 (0.185) | 0.676 (0.026) | 0.553 (0.119) | - | - |
| | | | ME | 2.237 (0.073) | 0.973 (0.040) | 2.294 (0.009) | 0.991 (0.005) | 1.081 (0.432) | 0.982 (0.030) | 1.201 (0.072) | 0.839 (0.119) |
| | | | QME | 1.844 (0.282) | 0.858 (0.152) | 2.062 (0.028) | 0.960 (0.029) | 0.779 (0.237) | 0.923 (0.091) | 1.178 (0.077) | 0.807 (0.112) |
| | 2 | No | XE | 0.066 (0.045) | 0.622 (0.116) | 0.133 (0.087) | 0.580 (0.123) | 0.141 (0.157) | 0.668 (0.169) | – | 0.438 (0.035) |
| | | | CP | 0.074 (0.044) | 0.627 (0.121) | 0.133 (0.089) | 0.559 (0.113) | 0.123 (0.140) | 0.633 (0.137) | – | 0.460 (0.045) |
| | | | LS | 0.306 (0.150) | 0.575 (0.106) | 0.446 (0.219) | 0.551 (0.106) | 0.329 (0.266) | 0.805 (0.121) | – | 0.476 (0.024) |
| | | | ME | 2.291 (0.014) | 0.798 (0.087) | 2.295 (0.007) | 0.685 (0.082) | 1.219 (0.214) | 0.935 (0.033) | – | 0.619 (0.006) |
| | | | QME | 1.966 (0.256) | 0.862 (0.066) | 1.825 (0.197) | 0.900 (0.075) | 0.238 (0.187) | 0.919 (0.044) | – | 0.582 (0.019) |
| | | Yes | XE | 0.226 (0.116) | 0.748 (0.121) | 0.194 (0.059) | 0.724 (0.085) | 0.400 (0.255) | 0.821 (0.065) | – | 0.372 (0.027) |
| | | | CP | 0.227 (0.106) | 0.729 (0.132) | 0.438 (0.283) | 0.812 (0.055) | 0.213 (0.058) | 0.705 (0.142) | – | 0.405 (0.048) |
| | | | LS | 0.636 (0.094) | 0.648 (0.138) | 0.807 (0.213) | 0.831 (0.033) | 0.763 (0.137) | 0.634 (0.127) | – | 0.407 (0.034) |
| | | | ME | 2.225 (0.064) | 0.930 (0.030) | 2.272 (0.040) | 0.975 (0.010) | 0.672 (0.541) | 0.870 (0.098) | – | 0.598 (0.015) |
| | | | QME | 1.347 (0.469) | 0.846 (0.112) | 1.158 (0.368) | 0.910 (0.055) | 0.205 (0.192) | 0.868 (0.085) | – | 0.486 (0.026) |
| | 5 | No | XE | – | 0.556 (0.042) | – | 0.578 (0.133) | – | – | – | – |
| | | | CP | – | 0.558 (0.061) | – | 0.607 (0.132) | – | – | – | – |
| | | | LS | – | 0.566 (0.066) | – | 0.551 (0.066) | – | – | – | – |
| | | | ME | – | 0.485 (0.067) | – | 0.639 (0.047) | – | – | – | – |
| | | | QME | – | 0.601 (0.090) | – | 0.784 (0.039) | – | – | – | – |
| | | Yes | XE | – | 0.500 (0.094) | – | 0.687 (0.069) | – | – | – | – |
| | | | CP | – | 0.437 (0.034) | – | 0.718 (0.031) | – | – | – | - |
| | | | LS | – | 0.486 (0.032) | – | 0.806 (0.025) | – | – | – | - |
| | | | ME | – | 0.800 (0.018) | – | 0.919 (0.008) | – | – | – | – |
| | | | QME | – | 0.649 (0.068) | – | 0.807 (0.040) | – | – | – | – |
| CNN | 1 | No | XE | 0.045 (0.097) | 0.592 (0.175) | 0.077 (0.172) | 0.517 (0.064) | – | – | – | – |
| | | | CP | 0.031 (0.074) | 0.547 (0.131) | 0.064 (0.147) | 0.513 (0.053) | – | – | – | – |
| | | | LS | 0.197 (0.192) | 0.510 (0.093) | 0.225 (0.249) | 0.518 (0.073) | – | – | – | – |
| | | | ME | 2.226 (0.107) | 0.950 (0.056) | 2.102 (0.273) | 0.977 (0.027) | – | – | – | – |
| | | | QME | 1.272 (0.324) | 0.906 (0.098) | 1.210 (0.301) | 0.970 (0.047) | – | – | – | – |
| | | Yes | XE | 0.022 (0.043) | 0.576 (0.158) | 0.023 (0.104) | 0.520 (0.070) | – | – | – | – |
| | | | CP | 0.024 (0.040) | 0.575 (0.152) | 0.002 (0.006) | 0.509 (0.022) | – | – | – | – |
| | | | LS | 0.169 (0.165) | 0.582 (0.151) | 0.196 (0.219) | 0.527 (0.081) | – | – | – | – |
| | | | ME | 2.189 (0.112) | 0.966 (0.037) | 2.021 (0.357) | 0.984 (0.020) | – | – | – | – |
| | | | QME | 1.096 (0.287) | 0.896 (0.131) | 0.969 (0.371) | 0.952 (0.081) | – | – | – | – |
| | 2 | No | XE | 0.081 (0.050) | 0.664 (0.164) | 0.095 (0.050) | 0.711 (0.130) | – | – | – | – |
| | | | CP | 0.092 (0.057) | 0.671 (0.156) | 0.135 (0.067) | 0.731 (0.100) | – | – | – | – |
| | | | LS | 0.212 (0.101) | 0.618 (0.148) | 0.275 (0.126) | 0.683 (0.130) | – | – | – | – |
| | | | ME | 2.147 (0.093) | 0.894 (0.065) | 1.981 (0.204) | 0.948 (0.040) | – | – | – | – |
| | | | QME | 1.188 (0.318) | 0.909 (0.055) | 0.743 (0.376) | 0.936 (0.061) | – | – | – | – |
| | | Yes | XE | 0.047 (0.037) | 0.630 (0.162) | 0.062 (0.032) | 0.638 (0.135) | – | – | – | – |
| | | | CP | 0.054 (0.044) | 0.614 (0.158) | 0.066 (0.043) | 0.623 (0.150) | – | – | – | – |
| | | | LS | 0.183 (0.123) | 0.587 (0.120) | 0.233 (0.111) | 0.644 (0.132) | – | – | – | – |
| | | | ME | 2.055 (0.190) | 0.926 (0.047) | 1.649 (0.466) | 0.964 (0.040) | – | – | – | – |
| | | | QME | 0.694 (0.372) | 0.898 (0.065) | 0.366 (0.260) | 0.945 (0.054) | – | – | – | – |
| | 5 | No | XE | – | 0.462 (0.034) | – | 0.708 (0.087) | – | – | – | – |
| | | | CP | – | 0.475 (0.071) | – | 0.738 (0.034) | – | – | – | – |
| | | | LS | – | 0.497 (0.086) | – | 0.641 (0.113) | – | – | – | – |
| | | | ME | – | 0.803 (0.055) | – | 0.915 (0.042) | – | – | – | – |
| | | | QME | – | 0.767 (0.043) | – | 0.884 (0.043) | – | – | – | – |
| | | Yes | XE | – | 0.493 (0.144) | – | 0.776 (0.061) | – | – | – | – |
| | | | CP | – | 0.535 (0.045) | – | 0.794 (0.092) | – | – | – | – |
| | | | LS | – | 0.538 (0.069) | – | 0.554 (0.173) | – | – | – | – |
| | | | ME | – | 0.791 (0.083) | – | 0.943 (0.034) | – | – | – | – |
| | | | QME | – | 0.790 (0.046) | – | 0.909 (0.034) | – | – | – | – |

## C.8 IMAGENETTE

To evaluate the generality of our framework on large-scale image data, we apply our pipeline to **Imagenette** (Howard, 2019), a challenging 10-class subset of ImageNet (Deng et al., 2009) comprising visually diverse categories (tench, English springer, cassette player, chain saw, church, French horn, garbage truck, gas pump, golf ball, parachute). We train a ResNet18 with the MaxEnt loss, using a pruning ratio of $0.6$ whenever IMP is applied, following the hyperparametrisation recommended by the dataset authors.

**Pairwise merge.** Table 10 reports pairwise merge results across cardinalities $|\mathcal{R}| \in \{1, 2, 5\}$. Logit-space merging consistently outperforms weight-space merging across all settings. For $|\mathcal{R}| = 5$, when the union of rewarded sets covers the entire dataset, logit-merged modules *match or surpass* the CrossEntropy baseline trained on the full dataset in the standard way ($0.894 \pm 0.002$ vs. $0.866 \pm 0.011$). Weight-space merging degrades at higher cardinalities, consistent with our findings on CIFAR-10 and underscoring that logit merging is the reliable composition rule for complex architectures.

Table 10: Pairwise merge results on Imagenette (ResNet18). Mean and std over 5 seeds. The CrossEntropy baseline trained on the full dataset achieves $0.866 \pm 0.011$.

| $|\mathcal{R}|$ | IMP | Merge | Entropy | Rewarded Acc |
|---|---|---|---|---|
| 1 | No | Weights | $1.899 \pm 0.367$ | $0.628 \pm 0.105$ |
| | No | Logits | $2.293 \pm 0.002$ | $0.952 \pm 0.021$ |
| | Yes | Weights | $2.064 \pm 0.258$ | $0.641 \pm 0.086$ |
| | Yes | Logits | $2.288 \pm 0.006$ | $0.942 \pm 0.024$ |
| 2 | No | Weights | $2.152 \pm 0.154$ | $0.452 \pm 0.063$ |
| | No | Logits | $2.282 \pm 0.004$ | $0.926 \pm 0.021$ |
| | Yes | Weights | $2.127 \pm 0.186$ | $0.409 \pm 0.058$ |
| | Yes | Logits | $2.282 \pm 0.008$ | $0.928 \pm 0.020$ |
| 5 | No | Weights | – | $0.334 \pm 0.075$ |
| | No | Logits | – | $0.886 \pm 0.006$ |
| | Yes | Weights | – | $0.206 \pm 0.009$ |
| | Yes | Logits | – | $\mathbf{0.894 \pm 0.002}$ |

**Complete merge.** Table 11 shows the accuracy of the fully merged model at the final step of the complete-merge sequence (one class-specific module added at a time until all 10 classes are covered). Logit merging achieves $0.849 \pm 0.007$ with IMP and $0.845 \pm 0.005$ without, remaining only slightly below the CrossEntropy baseline ($0.866 \pm 0.011$). Weight-space merging collapses to near-random performance, confirming that the composition difficulty observed on CIFAR-10 extends to larger-scale datasets.

Table 11: Final step of the complete merge on Imagenette (ResNet18). CrossEntropy baseline: $0.866 \pm 0.011$.

| IMP | Merge | Accuracy |
|---|---|---|
| No | Weights | $0.113 \pm 0.014$ |
| No | Logits | $0.845 \pm 0.005$ |
| Yes | Weights | $0.103 \pm 0.013$ |
| Yes | Logits | $\mathbf{0.849 \pm 0.007}$ |

Taken together, these results show that MaxEnt + sparsity training reliably identifies class-specific submodules at the scale of ImageNet-derived data, and that logit-space merging produces a generalist model that closely matches a monolithic CrossEntropy baseline.

## C.9 CIFAR-100

To probe the limits of compositionality on a dataset with a *large number of classes*, we evaluate ResNet18 with MaxEnt on CIFAR-100 (Krizhevsky et al., 2009). All experiments use a pruning ratio of $0.6$ (when IMP is applied).

**Pairwise merge at cardinality 50.** Table 12 reports results for $|\mathcal{R}| = 50$, i.e., two submodules each specialising on 50 disjoint classes that are then merged. Logit merging again outperforms weight merging in both the pruned and unpruned setting. The logit-merged model (No-IMP: $0.644 \pm 0.002$; IMP: $0.521 \pm 0.005$) approaches the CrossEntropy baseline ($0.661 \pm 0.006$), whereas weight merging degrades substantially.

Table 12: Pairwise merge at $|\mathcal{R}| = 50$ on CIFAR-100 (ResNet18). CrossEntropy baseline: $0.661 \pm 0.006$.

| IMP | Merge | Rewarded Accuracy |
|-----|---------|----------------------|
| No | Weights | $0.107 \pm 0.011$ |
| No | Logits | $\mathbf{0.644 \pm 0.002}$ |
| Yes | Weights | $0.020 \pm 0.002$ |
| Yes | Logits | $0.521 \pm 0.005$ |

**Complete merge with 100 modules.** We conducted a stress test useful to detect whether we can merge more than 10 class-specific sub-modules together or if there is a bottleneck. To such an extent, we tested, for CIFAR-100, the complete merge experiment (i.e., merging 100 sub-modules each specialized on a single class). This is the most challenging scenario. Table 13 reports accuracy at representative steps of the complete-merge sequence (steps 1, 25, 50, 75, 100), run on a single seed due to computational cost. Each step adds one additional single-class module to the merged model. We observed no sudden drop in performance each time we added a new model (till reaching 100 steps). The rewarded accuracy decreases gradually from nearly 100% (initial modules) to roughly 50% at the end of the merge sequence. Importantly, logit merging maintains above 60% rewarded accuracy for roughly the first 50 merged modules before gradually decaying, reaching $0.533$ (with IMP) and $0.509$ (without IMP) at step 100 compared to the CrossEntropy baseline of $0.661$. These results confirm that MaxEnt + sparsity training scales to datasets with high visual complexity and large numbers of classes. Logit-space merging remains the reliable composition strategy, recovering a substantial fraction of monolithic performance even when composing 100 independently trained modules.

Table 13: Complete merge on CIFAR-100 (ResNet18), single seed. CrossEntropy baseline: $0.661 \pm 0.006$.

| IMP | Merge | Step | Accuracy |
|-----|-------|------|----------|
| No | Logits | 1 | 0.992 |
| | | 25 | 0.654 |
| | | 50 | 0.581 |
| | | 75 | 0.536 |
| | | 100 | 0.509 |
| | Weights | 1 | 0.992 |
| | | 25 | 0.046 |
| | | 50 | 0.019 |
| | | 75 | 0.011 |
| | | 100 | 0.009 |
| Yes | Logits | 1 | 0.985 |
| | | 25 | 0.671 |
| | | 50 | 0.606 |
| | | 75 | 0.560 |
| | | 100 | **0.533** |
| | Weights | 1 | 0.985 |
| | | 25 | 0.021 |
| | | 50 | 0.012 |
| | | 75 | 0.009 |
| | | 100 | 0.009 |

## C.10   TEXT CLASSIFICATION

To evaluate the generality of our framework beyond vision tasks, we conduct a comprehensive study on two standard text-classification benchmarks: **IMDB** (Maas et al., 2011) (binary sentiment classification) and **20 Newsgroups** (Lang, 1995) (20-way topic classification). These datasets differ substantially in size, difficulty, and label-space cardinality, allowing us to assess the robustness of modular training and merging in a qualitatively different regime.

Our evaluation mirrors the setup used in the main paper: models are trained to specialize on subsets of classes via the MaxEnt objective, optionally combined with IMP, and then merged using either **weight-space summation** or **logit-space averaging** (Table 14). We compare the resulting merged models to **CrossEntropy baselines trained on the full dataset** (Table 15).

Across both datasets and both architectures (shallow and deep MLPs), we find that merged models attain performances comparable to, and in the cases with biggest cardinality often even exceeding, the full-dataset CrossEntropy baselines. We can notice this in **IMDB** where logit-merged modules without IMP consistently reach 0.88 accuracy, on par with or slightly above the CE baseline (0.88 for shallow MLP and 0.87 for deep MLP). For **20NG**, logit-merge again produces strong merged models. For example, when merging modules with cardinality 10 and without using IMP, shallow MLP reaches 0.70 accuracy, on par with the baseline, whilst deep MLP obtains 0.68, surpassing the 0.64 of the corresponding baseline.

Table 14: Performance of merged submodules on text datasets (20NG and IMDB). Mean and std over 5 seeds.

| Dataset | Model | $|\mathcal{R}|$ | LTH | Merge | Entropy | Rewarded Acc |
|---------|-------|-----|-----|-------|---------|--------------|
| 20NG | Deep MLP | 1 | No | Logit | 2.986 (0.004) | 0.878 (0.069) |
| | | | No | Weight | 2.833 (0.032) | 0.703 (0.091) |
| | | | Yes | Logit | 2.977 (0.007) | 0.886 (0.083) |
| | | | Yes | Weight | 2.856 (0.090) | 0.879 (0.083) |
| | | 5 | No | Logit | 2.917 (0.016) | 0.739 (0.024) |
| | | | No | Weight | 2.248 (0.128) | 0.403 (0.035) |
| | | | Yes | Logit | 2.905 (0.019) | 0.729 (0.022) |
| | | | Yes | Weight | 2.678 (0.064) | 0.719 (0.022) |
| | | 10 | No | Logit | – | 0.677 (0.005) |
| | | | No | Weight | – | 0.368 (0.021) |
| | | | Yes | Logit | – | 0.667 (0.004) |
| | | | Yes | Weight | – | 0.654 (0.008) |
| 20NG | Shallow MLP | 1 | No | Logit | 2.984 (0.003) | 0.872 (0.074) |
| | | | No | Weight | 2.764 (0.051) | 0.810 (0.096) |
| | | | Yes | Logit | 2.981 (0.005) | 0.859 (0.085) |
| | | | Yes | Weight | 2.946 (0.009) | 0.857 (0.083) |
| | | 5 | No | Logit | 2.926 (0.012) | 0.756 (0.020) |
| | | | No | Weight | 2.126 (0.126) | 0.528 (0.060) |
| | | | Yes | Logit | 2.933 (0.013) | 0.734 (0.024) |
| | | | Yes | Weight | 2.753 (0.043) | 0.728 (0.025) |
| | | 10 | No | Logit | – | 0.704 (0.002) |
| | | | No | Weight | – | 0.599 (0.020) |
| | | | Yes | Logit | – | 0.673 (0.002) |
| | | | Yes | Weight | – | 0.670 (0.001) |
| IMDB | Deep MLP | 1 | No | Logit | – | 0.877 (0.001) |
| | | | No | Weight | – | 0.684 (0.049) |
| | | | Yes | Logit | – | 0.848 (0.003) |
| | | | Yes | Weight | – | 0.849 (0.005) |
| IMDB | Shallow MLP | 1 | No | Logit | – | 0.880 (0.001) |
| | | | No | Weight | – | 0.870 (0.002) |
| | | | Yes | Logit | – | 0.848 (0.002) |
| | | | Yes | Weight | – | 0.849 (0.002) |

Table 15: CrossEntropy baselines for reference in text classification. Mean and std over 5 seeds.

| Dataset | Model | $|\mathcal{R}|$ | Loss | Acc | Std |
|---------|-------|-----|------|-----|-----|
| 20NG | Deep MLP | 20 | XE | 0.639 | 0.005 |
| 20NG | Shallow MLP | 20 | XE | 0.702 | 0.001 |
| IMDB | Deep MLP | 2 | XE | 0.867 | 0.002 |
| IMDB | Shallow MLP | 2 | XE | 0.876 | 0.006 |

