# OpenReview forum: "Composable Sparse Subnetworks via Maximum-Entropy Principle"
_ICLR.cc/2026/Conference — ICLR 2026 Poster_

### Official Review · Reviewer_EE1e · 2025-10-20

**Soundness:** 2
**Presentation:** 3
**Contribution:** 3
**Rating:** 4
**Confidence:** 4

**Summary:**

This paper introduces a method for training sparse networks that accurately classify only a designated subset of classes while remaining deliberately uncertain on all others, functioning as class-specific subnetworks.

**Strengths:**

The topic this paper focused on is very interesting.

**Weaknesses:**

1. The motivation of using Iterative Magnitude Pruning is not clear enough. Authors should explain clearly why using such pruning is a necessity.

2. Authors only conduct experiments on simple networks such as MLP and LeNet, making results not convincing enough. Can the proposed method be applied to other widely-used DNNs, such as ResNets, transformer-based models, VIT?

3. Can the proposed method be applied to large-scale and complex image dataset, such as imagenet or CUB200? MNIST is too simple.

4. Can the proposed method be generalized to other tasks, such as text classification and other tasks rather than classification?

5. What is the pratical usage of the proposed method? Authors can add a discussion for this.

**Questions:**

Please refer to the weaknesses, especially more experiments on larger models (e.g., ResNet, ViT) and more complex tasks (e.g., CIFAR, ImageNet).
If all my problems are addressed, I will raise my rating.

---

> ### Author Response · Authors · 2025-11-25
> **Response 1/2 to Reviewer EE1e**
>
> # **Deer Reviewer EE1e,**
>
> We thank you for your careful reading of our work and for finding the topic of our work **“very interesting”**. Your concerns helped us significantly improve the manuscript. Below, we address each point in depth.
>
> ---
>
> **Q**: *The motivation of using Iterative Magnitude Pruning is not clear enough. Authors should explain clearly why using such pruning is a necessity.*
>
> **RE**: We appreciate the opportunity to clarify this point. Our **MaxEnt objective** is designed to train only the **functional submodule/circuit** associated with the **rewarded classes**. However, if we didn’t use any pruning strategy, we would retain many weights that are **not essential** to the prediction of the rewarded set, i.e. are not in their circuits.
> IMP is therefore not a generic sparsification step, but a critical mechanism to **isolate class-specific circuits**: MaxEnt encourages **functional isolation at the logit level**, but IMP **removes weights that do not carry reward-set information**, yielding subnetworks that are both **sparse and functionally separated**.
>
> As shown in Appendix from Table 4 to Table 9 (and more easily visualizable from Figure 26 to Figure 30), MaxEnt alone already improves module isolation, but the combination **MaxEnt + sparsity training** consistently yields the best merging behaviour for simpler datasets and architectures.
>
> On more complex models (i.e. **ResNet**) and more complex datasets (i.e. **CIFAR-10**) the situation changes since simple **weight-space merge** does not reach optimal performance. We instead resolve to a **logit-space merge** (suggested by Reviewer gGJL). In such a regime, the pruning strategy is still fundamental to identify a quite small (wrt to the full model size) circuit.
>
> The focus of our present work is to show that MaxEnt + sparsity training reliably identifies class-specific circuits,  which can be composed into a generalist model, either through weight-space merge (simple settings) or logit-space merge (complex settings). This provides a principled and practically useful tool for studying and composing circuits.
>
> ---
>
> **Q**: *Authors only conduct experiments on simple networks such as MLP and LeNet, making results not convincing enough. Can the proposed method be applied to other widely-used DNNs, such as ResNets, transformer-based models, VIT?*
>
> **RE**: We have now added over **10,000 new runs** on: **ResNet-18**, **VGG11** (with and without **BatchNorm**), considering FMNIST, MNIST, **CIFAR-10**.  Following Reviewer gGJL’s suggestion, we now analyse not only the original **weight-space merge**, but also a new **logit-space merge**.
>
> Our **pipeline (MaxEnt + IMP)** remains unchanged, and the results offer a multifaceted picture:
> - For **simple datasets** and **simple or some complex models** (e.g., ResNet on MNIST and FMNIST), plain **weight summation** still works, once BatchNorm running statistics are re-estimated.
> - For **complex datasets + complex models** (e.g., ResNet-18 on CIFAR-10), naive weight-sum merging is insufficient. However, the **logit-level merging recovers most of the monolithic performance**, and confirms that **the procedure successfully learns the class-specific submodules**. This holds also for complete merging.
> The results are displayed in **Table 2 (new)**, **Figure 4 (new)** and **Figure 5 (new)**, as well as in the Appendix.
>
> ---
>
> **Q**: *Can the proposed method be applied to large-scale and complex image datasets, such as imagenet or CUB200? MNIST is too simple.*
>
> **RE**: Our experiments show that the **circuit-identification stage scales (on CIFAR-10) well**, while **circuit-composition** through summing the weights must be replaced, e.g. through **logit merge**. The results and their comments are the same as the previous answer.
>
> ---

---

> > ### Author Response · Authors · 2025-11-25
> > **Response 2/2 to Reviewer EE1e**
> >
> > **Q**: *Can the proposed method be generalized to other tasks, such as text classification and other tasks rather than classification?*
> >
> > **RE**: We appreciate the Reviewer’s insightful comment. Our current work indeed focuses on **classification tasks**. To the best of our knowledge, training **composable, isolated task-specifc submodules/circuits** is an interesting direction that hasn’t been explored enough in the existing literature. The two lines of work that are most closely related to ours are (i) **post-hoc circuit analysis** and (ii) canonical **weight-space model merging**. However, both are fundamentally different from our setting: we train the circuits from scratch and explicitly aim to merge circuits that are **functionally different**, whereas weight-space model merging typically considers models that solve the same task and differ mainly in parameter space.
> >
> > Moreover, the foundational works in both of these related areas, as well as a substantial fraction of the broader literature, primarily focus on **classification benchmarks on standard computer vision datasets**. For instance, see [1] as an example of post-hoc circuit analysis. In this sense, our experimental setup is **well aligned with prevailing practice in neighboring domains**.
> >
> > We agree that extending our approach beyond classification (e.g., to text classification, sequence modeling, or more general prediction tasks) is an important and interesting direction. At this stage, we do not yet have a definitive answer regarding how well our method generalizes to non-classification tasks. We view this as a **promising avenue for future work**, and, given that this is, to our knowledge, the first study in this new setting of training and merging composable circuits, we believe it is reasonable for the initial exploration to adopt assumptions similar to those made in closely related literature.
> >
> > **[1]** *Olah, et al., "Zoom In: An Introduction to Circuits", Distill, 2020.*
> >
> > ---
> >
> > **Q**: *What is the practical usage of the proposed method? Authors can add a discussion for this.*
> >
> > **RE**: Thanks for the comment. While we would like to clarify that the aim of this work is primarily to be of foundational nature, we can clearly identify some practical usages, expanding this discussion in the paper (see **Section 5**):
> >
> > - **Selective unlearning by design**. Removing or down-weighting a module corresponds to removing a class, without retraining. This is relevant in: safety constraints, privacy-driven machine unlearning, and data governance.
> > - **Federated and distributed settings**. Clients can train concept-specific modules locally and aggregate them without sharing data. This is especially appealing in medical or privacy-sensitive contexts.
> > - **Mechanistic interpretability**. Unlike post-hoc circuit extraction techniques, our method provides a principled a priori way to learn circuits before they are entangled by joint training, enabling cleaner mechanistic analyses.
> > - **Verification of Neural Networks**. Formal verification applied to neural networks currently is limited in its practical adoption by its enormous computational cost when scaling to networks and dataset of meaningful sizes. With our approach, thanks to IMP and class-specificity, individual submodules have a scale much more approachable for SOTA verification techniques.
> >
> > ---
> >
> > ## Recap
> > In conclusion, in response to the Reviewer’s concerns, we have:
> > - clarified the **necessity and role of IMP**,
> > - added **extensive experiments (more than 10,000 runs)** on **widely used DNNs (ResNet-18, VGG11)**, and
> > - **more complex datasets (CIFAR-10)**,
> > - outlined our position in the literature and thus justified the choice of the tasks,
> > - and expanded the section describing **practical applications**.
> >
> > We sincerely appreciate the Reviewer’s feedback and hope that these additions fully address the raised concerns.

---

> > > ### Comment · Reviewer_EE1e · 2025-11-27
> > > **Response**
> > >
> > > Thanks for authors' response. However, some of my questions are not addressed. For example, CIFAR-10 is not a dataset with large-scale images. Besides, authors do not clarify whether their methods can be generalized to other tasks, such as text classification and other tasks rather than classification. The most direct way to answer this question is to conduct an experiments to verify its generalization capacity.

---

> ### Author Response · Authors · 2025-12-03
> **Second Response 1/3 to Reviewer EE1e**
>
> # **Dear EE1e,**
>
> We thank the Reviewer for the follow-up and for highlighting the need to more clearly demonstrate the generalization capacity of our approach beyond the settings originally included in the submission. In particular, the Reviewer requests evidence that our method can extend both to **large scale image datasets** and to **non-vision tasks** *such as* **text classification** (that also complement the two tabular datasets in our original submission).
> In our new experiments, we directly address both of these points, showing that our method continues to perform robustly and effectively in these broader scenarios.
>
> ## **Large Scale Images**
>
> We evaluate **ResNet18 trained with MaxEnt on Imagenette** (using a pruning ratio of 0.6 whenever pruning is applied and following the hyperparametrization suggested in  \[1\]) to test our framework on a large scale image dataset. Imagenette \[1\] is a subset of 10 classes from Imagenet (tench, English springer, cassette player, chain saw, church, French horn, garbage truck, gas pump, golf ball, parachute).
>
> Overall, we observe that merging in **logit-space consistently outperforms merging in weight-space** (see Table below). Moreover, in both merging regimes, **pruning does not significantly degrade performance**, with the only notable exception being the weight-space merge at cardinality 5\. All these results are consistent with our original findings and with all the additional results we derived through the rebuttal period, and that we have already included in the manuscript, as we mentioned in our previous reply (see Figures 4, 5, 26-30, Table 2, updated with logit-merge, and Tables 6, 7, 8).
> A striking result emerges in the logit-space merge: when the **union of the rewarded sets covers the entire dataset**, the merged models **match or even surpass the baseline trained on the whole dataset in the standard way with CrossEntropy** (0.894 ± 0.002 vs 0.866 ± 0.011).
>
> We also report results for the **final step of the complete merge experiments on Imagenette**, which is the most challenging setting (submodules of a single class merged one after the other untill we get a merged model representing all the classes). This allows another direct comparison to the baseline. Even here, the **fully merged model** (0.849 ± 0.007) **remains only slightly below the CrossEntropy baseline** (0.866 ± 0.011), which is a strong indicator of the robustness of our approach.
>
> **\[1\]** [https://github.com/fastai/imagenette](https://github.com/fastai/imagenette), by J. Howard
>
> **Pairwise Merge Results**
>
> | Card | IMP | Merge | Entropy | Rew. Acc. |
> | :---- | :---- | :---- | :---- | :---- |
> | 1  | No | Weights | 1.899 ± 0.367 | 0.628 ± 0.105 |
> | 1 | No | Logits | 2.293 ± 0.002 | 0.952 ± 0.021 |
> | 1 | Yes | Weights | 2.064 ± 0.258 | 0.641 ± 0.086 |
> | 1 | Yes | Logits | 2.288 ± 0.006 | 0.942 ± 0.024 |
> | 2 | No | Weights | 2.152 ± 0.154 | 0.452 ± 0.063 |
> | 2 | No | Logits | 2.282 ± 0.004 | 0.926 ± 0.021 |
> | 2 | Yes | Weights | 2.127 ± 0.186 | 0.409 ± 0.058 |
> | 2 | Yes | Logits | 2.282 ± 0.008 | 0.928 ± 0.020 |
> | 5 | No | Weights | None | 0.334 ± 0.075 |
> | 5 | No | Logits | None | 0.886 ± 0.006 |
> | 5 | Yes | Weights | None | 0.206 ± 0.009 |
> | 5 | Yes | Logits | None | 0.894 ± 0.002 |
>
> **Final Step of Complete Merge Results**
>
> | IMP | Merge | Accuracy |
> | :---- | :---- | :---- |
> | No | Weights | 0.113 ± 0.014  |
> | No | Logits | 0.845 ± 0.005 |
> | Yes | Weights | 0.103 ± 0.013  |
> | Yes | Logits | 0.849 ± 0.007 |
>
> **Baseline(trained with CrossEntropy on full Imagenette): 0.866 ± 0.011**

---

> ### Author Response · Authors · 2025-12-03
> **Second Response 2/3 to Reviewer EE1e**
>
> ## **Text Classification**
>
> To evaluate the generality of our framework beyond vision and tabular-data tasks, we conduct a comprehensive study on two standard text-classification benchmarks: **IMDB** \[2\] (binary sentiment classification) and **20 Newsgroups** \[3\] (20-way topic classification). These datasets differ substantially in size, difficulty, and label-space cardinality, allowing us to assess the robustness of modular training and merging in a qualitatively different regime.
>
> Our evaluation mirrors the setup already described in the main paper: models are trained to specialize on subsets of classes via the MaxEnt objective, optionally combined with IMP, and then merged using either **weight-space summation** or **logit-space averaging**. We compare the resulting merged models to **CrossEntropy baselines trained in the standard way on the full dataset.**
>
> Across both datasets and both architectures (shallow and deep MLPs), we find that merged models attain performances comparable to, and in the cases with biggest cardinality often even exceeding, the full-dataset CrossEntropy baselines.
> We can notice this in **IMDB**, where logit-merged modules without IMP consistently reach 0.88 accuracy, on par with or slightly above the CrossEntropy baseline (0.88 for shallow MLP and 0.87 for deep MLP). For **20NG**, logit-merge again produces strong merged models. For example, when merging modules with cardinality 10 and without using IMP, shallow MLP reaches 0.70 accuracy, on par with the baseline, whilst deep MLP obtains 0.68, surpassing the 0.64 of the corresponding baseline. Again these results are consistent with our original findings and support the idea that our approach is general and not limited to vision or tabular modality.
>
> **\[2\]** Maas, A. L., Daly, R. E., Pham, P. T., Huang, D., Ng, A. Y., & Potts, C. (2011).  Learning Word Vectors for Sentiment Analysis. *Proceedings of the 49th Annual Meeting of the Association for Computational Linguistics: Human Language Technologies*, 142–150.
>
> **\[3\]** Lang, K. (1995). Newsweeder: Learning to Filter Netnews.  Available at: http://qwone.com/\~jason/20Newsgroups/
>
> **Table: Performance of merged submodules on text datasets (20NG and IMDB)**
>
> *Mean and standard deviation over 5 seeds.*
>
> **20 Newsgroups — Deep MLP**
> | Cardinality | IMP | Merge  | Entropy       | Rewarded Acc  |
> | ----------- | --- | ------ | ------------- | ------------- |
> | 1           | No  | Logit  | 2.986 ± 0.004 | 0.878 ± 0.069 |
> | 1           | No  | Weight | 2.833 ± 0.032 | 0.703 ± 0.091 |
> | 1           | Yes | Logit  | 2.977 ± 0.007 | 0.886 ± 0.083 |
> | 1           | Yes | Weight | 2.856 ± 0.090 | 0.879 ± 0.083 |
> | 5           | No  | Logit  | 2.917 ± 0.016 | 0.739 ± 0.024 |
> | 5           | No  | Weight | 2.248 ± 0.128 | 0.403 ± 0.035 |
> | 5           | Yes | Logit  | 2.905 ± 0.019 | 0.729 ± 0.022 |
> | 5           | Yes | Weight | 2.678 ± 0.064 | 0.719 ± 0.022 |
> | 10          | No  | Logit  | —             | 0.677 ± 0.005 |
> | 10          | No  | Weight | —             | 0.368 ± 0.021 |
> | 10          | Yes | Logit  | —             | 0.667 ± 0.004 |
> | 10          | Yes | Weight | —             | 0.654 ± 0.008 |
>
> **20 Newsgroups — Shallow MLP**
> | Cardinality | IMP | Merge  | Entropy       | Rewarded Acc  |
> | ----------- | --- | ------ | ------------- | ------------- |
> | 1           | No  | Logit  | 2.984 ± 0.003 | 0.872 ± 0.074 |
> | 1           | No  | Weight | 2.764 ± 0.051 | 0.810 ± 0.096 |
> | 1           | Yes | Logit  | 2.981 ± 0.005 | 0.859 ± 0.085 |
> | 1           | Yes | Weight | 2.946 ± 0.009 | 0.857 ± 0.083 |
> | 5           | No  | Logit  | 2.926 ± 0.012 | 0.756 ± 0.020 |
> | 5           | No  | Weight | 2.126 ± 0.126 | 0.528 ± 0.060 |
> | 5           | Yes | Logit  | 2.933 ± 0.013 | 0.734 ± 0.024 |
> | 5           | Yes | Weight | 2.753 ± 0.043 | 0.728 ± 0.025 |
> | 10          | No  | Logit  | —             | 0.704 ± 0.002 |
> | 10          | No  | Weight | —             | 0.599 ± 0.020 |
> | 10          | Yes | Logit  | —             | 0.673 ± 0.002 |
> | 10          | Yes | Weight | —             | 0.670 ± 0.001 |
>
> **IMDB — Deep MLP** (Cardinality is 1)
> | IMP | Merge  | Rewarded Acc  |
> | --- | ------ | ------------- |
> | No  | Logit  | 0.877 ± 0.001 |
> | No  | Weight | 0.684 ± 0.049 |
> | Yes | Logit  | 0.848 ± 0.003 |
> | Yes | Weight | 0.849 ± 0.005 |
>
> **IMDB — Shallow MLP** (Cardinality is 1)
> | IMP | Merge  | Rewarded Acc  |
> | --- | ------ | ------------- |
> | No  | Logit  | 0.880 ± 0.001 |
> | No  | Weight | 0.870 ± 0.002 |
> | Yes | Logit  | 0.848 ± 0.002 |
> | Yes | Weight | 0.849 ± 0.002 |

---

> ### Author Response · Authors · 2025-12-03
> **Second Response 3/3 to Reviewer EE1e**
>
> **CrossEntropy Baselines (text classification)**
> | Dataset | Model       | Cardinality | Loss | Accuracy |
> | ------- | ----------- | ----------- | ---- | -------- |
> | 20NG    | Deep MLP    | 20          | CE   | 0.639 ± 0.005 |
> | 20NG    | Shallow MLP | 20          | CE   | 0.702 ± 0.001 |
> | IMDB    | Deep MLP    | 2           | CE   | 0.867 ± 0.002 |
> | IMDB    | Shallow MLP | 2           | CE   | 0.876 ± 0.006 |
>
> In summary, we provide new experiments that directly address the Reviewer’s concern about the scope and applicability of our method:
>
> * On **Imagenette**, a large-scale image dataset, MaxEnt-trained modules composed via logit merging perform extremely well, matching or surpassing a monolithic baseline, and remaining robust even in the complete-merge regime.
> * On **text classification**, we demonstrate that our approach generalizes beyond image tasks: across IMDB and 20NG, logit-merged modules (both with and without pruning) approach or exceed the performance of models trained with CrossEntropy in the standard way on the full dataset.
>
> These additional results (which will be included in the main body of the paper), once more, show that MaxEnt \+ sparsity training reliably identifies class-specific submodules also in **large-scale vision** and **non-vision classification** domains.
> We believe that, taken together with our previous experiments, these additions fully address the Reviewer’s concerns.

---

### Official Review · Reviewer_gGJL · 2025-10-29

**Soundness:** 4
**Presentation:** 4
**Contribution:** 3
**Rating:** 8
**Confidence:** 4

**Summary:**

The proposed method creates modular NNs for classification by training each subnetwork to recognize only a subset of the classes. When the input is from another class the network is trained to output a uniform probability vector. Sparsity is achieved by iterative pruning. Subnetworks are then combined by weight summation, which is justified by a modified mode-connectivity analysis. Experiment results show the subnetworks learn their tasks, the merged models succeed at the combined tasks, and pruning leads to better merging.

**Strengths:**

Clever training objective to make each subnet an expert on its classes but to be agnostic (i.e., not interfere) on other classes. The combined loss is elegant, relative entropy from either a one-hot or a uniform. Experiment results show the method works as intended.

**Weaknesses:**

The promise of modular and interpretable networks makes sense, but the paper would be stronger with some direct demonstrations of downstream applications that could not be done with a standard monolithic network.

The mode connectivity analysis is not well justified since (as the paper notes) it evaluates a combined loss that neither model was trained on. The results in fig 5 are not surprising, because each separate model does terribly on the other model’s task. The more important question is how the combined model’s loss for classes in $\mathcal{R}\_{\rm A}$ compares to model A’s loss on those classes. In the ideal case, where summing parameters results in summing the logits, these losses will be equal because model B’s logits will be uniform for items in $\mathcal{R}\_{\rm A}$. This is the rationale for the method, but how close to you come to that?

**Questions:**

Would there be a benefit of end-to-end fine-tuning after merging? Fine-tuning just the last linear layer could reduce interference between subnets and wouldn't impact modularity or interpretability.

The Quasi-MaxEnt loss will bias predictions of merged models for the reason explained in the final paragraph of sec 3.1. As an alternative baseline, what about giving each subnetwork an “other class” output for $\mathcal{C}\setminus\mathcal{R}$ which is then discarded in the merged model?

Combining the above two comments, I bet last-layer fine-tuning would eliminate the difference between QME and ME, since the targets of these two loss functions are linear combinations of each other (in logit space). The same should be true of one-vs-all formulations. If the primary question is whether an effective modular representation can be learned then it seems fair to free up the final weight layer, which is just an affine map from this representation to logits.

Merging models requires a choice of correspondence between neurons (or filters) because of permutation invariance. The naive approach of identifying neurons in different models that have the same nominal index amounts to a random permutation. Can you do better by choosing a permutation, for example to minimize overlap in weights?

Rather than merging models, you could compute each model separately and sum their logits. This would avoid questions of mode connectivity. In fact you’re effectively doing this already when pruning is high (e.g., the 99% pruning for the MLPs).

L193: I'm not certain what is meant by the first term on the RHS. Is it $\mathcal{L}(f\_{\theta\_1}(\mathcal{D}))$ or is $(1-t)$ included? Why the asymmetry, basing the threshold only on model 1?

Table 1: It would help to state the number of classes for each dataset (I had to look up HAR (6) and yeast (10)). Why is entropy so low for yeast when it is near maximum for the other datasets? The dataset is imbalanced (I calculated entropy 1.729) but the target should still be uniform implying entropy 2.302. Also it would help to state $|\mathcal{R}|=1$ since that is given only later in the text.

Fig 2: what are the units of the entries? I would have guessed they’re predicted probability summed over the test set, but the MNIST test set is balanced and the row sums here differ across rows (though they look to be consistent across tables).

There’s a claim that merged models maintain near-maximal entropy for non-rewarded classes (line 308) but Fig 2 suggests a significant bias toward rewarded classes. I calculated the entropy for classes 1 and 2 in fig 2a and got 1.97 and 1.78, much lower than the reported 2.28. Is this an atypical example?

---

> ### Author Response · Authors · 2025-11-25
> **Response 1/2 to Reviewer gGJL**
>
> # **Dear Reviewer gGJL,**
>
> We sincerely thank you for your **detailed and generous review**, and for rating both the **soundness** and **presentation** of the paper as **excellent**. We are pleased that you found our **training objective “clever”**, the **combined loss “elegant”**, and that overall the **experiments convinced you that the method works** as intended. Your comments are deeply appreciated and have helped us clarify several aspects that can further strengthen our work. Below, we address your questions point by point.
>
> ---
>
> **Q**: *Would there be a benefit of end-to-end fine-tuning after merging? Fine-tuning just the last linear layer could reduce interference between subnets and wouldn't impact modularity or interpretability.*
>
> **RE**: We have conducted preliminary experiments on **ResNet + FMNIST** (weight-space merge at cardinality 5), fine-tuning:
> - only the **last linear layer**,
> - or the **last residual block + the last linear layer**.
>
> While finetuning only the **last linear layer** doesn’t significantly improve the accuracy of the merged model (from 74.0 (1.9) to 76.9 (1.4) ) we did notice **significant improvements** when finetuning the **last residual block too** (86.2 (0.3) ), although results are not yet extensive enough to draw strong conclusions.
>
> ---
>
> **Q**: *The Quasi-MaxEnt loss will bias predictions of merged models for the reason explained in the final paragraph of sec 3.1. As an alternative baseline, what about giving each subnetwork an “other class” output for  which is then discarded in the merged model?*
>
> **RE**: We agree that this is an interesting baseline. Our intuition is that explicitly allocating an **“other” neuron** may create undesired dependencies, especially in deep networks, overall steering the flow of information in a **preferential direction**, a priori unjustified. Nevertheless, we view this as a meaningful **future direction**. Your suggestion also connects naturally to your next point on last-layer fine-tuning.
>
> ---
>
> **Q**: *Combining the above two comments, I bet last-layer fine-tuning would eliminate the difference between QME and ME, since the targets of these two loss functions are linear combinations of each other (in logit space). The same should be true of one-vs-all formulations. If the primary question is whether an effective modular representation can be learned then it seems fair to free up the final weight layer, which is just an affine map from this representation to logits.*
>
> **RE**: This is a sharp observation. Our preliminary experiments with **logit-level merging** support this intuition: **summing logits** reduces the gap between ME and QME more than **summing weights does**. We will try to report on the experiments about the fine-tuning but we already show the results of the merge in the **logit space** that we discuss below.
>
> ---
>
> **Q**: *Merging models requires a choice of correspondence between neurons (or filters) because of permutation invariance. The naive approach of identifying neurons in different models that have the same nominal index amounts to a random permutation. Can you do better by choosing a permutation, for example to minimize overlap in weights?*
>
> **RE**: There is already a research branch that deals with **permutations** and alignment for weight merging, but we should clearly position our work in this literature with more extended and specific experiments. We keep this as a direction for future works.
> However, it is worth mentioning that there is not much literature on our specific scenario (**submodules/circuits** trained **independently** from scratch to solve **different tasks**), and much of the alignment literature rests on assumptions that don’t hold in our scenario.
>
> ---
>
> **Q**: *Rather than merging models, you could compute each model separately and sum their logits. This would avoid questions of mode connectivity. In fact you’re effectively doing this already when pruning is high (e.g., the 99% pruning for the MLPs).*
>
> **RE**: We followed your suggestion and experimented with **averaging logits** instead of summing weights. For **simple datasets**, the two operations behave similarly. For **complex datasets**, **logit merge performs much better** (see also the new complete merge results in Figure 5), reinforcing the point that our training identifies the submodules/circuits. We have added extensive experiments using logit merge, and we have rephrased many parts of the paper following this suggestion. As a consequence, **we further thank the Reviewer for this constructive feedback**.
>
> ---
>
> **Q**: *L193: I'm not certain what is meant by the first term on the RHS. Is it $\mathcal{L}(f_{\theta_1}(\mathcal{D}))$ or is $(1 - t)$ included? Why the asymmetry, basing the threshold only on model 1?*
>
> **RE**: Eq. (193): We have **clarified the notation** in the paper. Please, if there are other doubts don’t hesitate to ask.
>
> ---

---

> > ### Author Response · Authors · 2025-11-25
> > **Response 2/2 to Reviewer gGJL**
> >
> > **Q**: *Table 1: It would help to state the number of classes for each dataset (I had to look up HAR (6) and yeast (10)). Why is entropy so low for yeast when it is near maximum for the other datasets? The dataset is imbalanced (I calculated entropy 1.729) but the target should still be uniform implying entropy 2.302. Also it would help to state  since that is given only later in the text.*
> >
> > **RE**: We now explicitly state the *number of classes* for each dataset in Section 4.2 (near Table 1) and clearly indicate that **Yeast** results refer to a **balanced subset of 4 classes** (this information was in the appendix but should have been in the main text too).
> >
> > ---
> >
> > **Q**: *Fig 2: what are the units of the entries? I would have guessed they’re predicted probability summed over the test set, but the MNIST test set is balanced and the row sums here differ across rows (though they look to be consistent across tables).*
> >
> >
> > **RE**: Fig. 2: Units correspond to total **predicted counts per class**; we now clarify this in the caption.
> >
> > ---
> >
> >
> > **Q**: *There’s a claim that merged models maintain near-maximal entropy for non-rewarded classes (line 308) but Fig 2 suggests a significant bias toward rewarded classes. I calculated the entropy for classes 1 and 2 in fig 2a and got 1.97 and 1.78, much lower than the reported 2.28. Is this an atypical example?*
> >
> >
> > **RE**:
> > You are correct: MaxEnt modules are **nearly uniform** but retain a slight **bias toward rewarded classes**. This bias is enough to shift the distribution of predictions when these are produced via argmax. As stated above, we have clarified that the matrices show **predicted counts** per class, so hopefully this clarification resolved the misunderstanding.
> >
> > ---
> >
> > ## Recap
> >
> > In the present work, we chose to focus on **isolating task-specific submodules** and demonstrating that they can be **recombined without joint training**. We believe this may find application in **unlearning** (i.e., removing or re-training just the single module), **neural network verification** (where formally verifying an entire network is often intractable), **federated learning** (where merging different networks is a key step), **mechanistic interpretability** by design, and many other settings. Showing concrete downstream uses is indeed valuable, and we plan to explore this in multiple future works (we now emphasize this point more clearly in the conclusion Section 5).
> >
> > We truly appreciate your constructive feedback and your positive assessment of our work. For this first response we focused on new experiments on **more complex datasets and models**, **baselines** proposed by Reviewer g43A and your suggestion of the **merge in the logit space**, to explore the directions of all the Reviewers (in total **more than 10,000 new runs**).
> >
> > So thank you very much for the precious feedback, and thank you even more if you would like to continue to give us feedback.

---

### Official Review · Reviewer_oJ2u · 2025-11-01

**Soundness:** 3
**Presentation:** 3
**Contribution:** 2
**Rating:** 4
**Confidence:** 4

**Summary:**

The paper examines whether independently trained class-specialized sub-networks can be combined into a single network that accurately identifies all classes.

The authors propose a method combining the maximum entropy principle with iterative magnitude pruning. The approach defines an objective function that uses KL divergence during module training to identify specific class subsets, while enforcing uniform distribution predictions for samples from other classes. The method employs iterative magnitude pruning with weight rewinding to enhance both functional specialization and module composability.

The authors evaluates the method using two image classification and two tabular classification datasets. The evaluation encompasses three architectures: a shallow MLP, a deep MLP, and a CNN. Results demonstrate that the specialized sub-network modules successfully identify specific class subsets and perform well when combined to handle all classes. The authors also conduct mode-connectivity analysis, revealing that the modules are mode-connected with a local minimum at the linear path's midpoint.

**Strengths:**

- The paper presents its hypotheses, questions, details, and conclusions with exceptional clarity.
- The proposed approach offers a novel and well-justified contribution.
- The results convincingly demonstrate the method's advantages over traditional training approaches, with mode-connectivity analysis providing valuable insights into local minima and composability outcomes.

**Weaknesses:**

- The method's application is restricted to classification tasks.
- The approach only identifies end-to-end modules and requires manual class assignment, limiting its utility for complex tasks where classification is just one component and intermediate labels may be unavailable.
- The experiments are confined to small-scale settings, and composability deteriorates with increased network depth, raising concerns about the method's scalability to deeper networks.

**Questions:**

- What factors contribute to the degradation of composability in deeper networks? This issue requires further investigation and discussion to understand its implications.
- Why does iterative magnitude pruning show greater effectiveness in deeper networks compared to shallow MLPs? Additional insights would be valuable.
- How does this work relate to broader research trends addressing increasingly complex tasks beyond classification?

---

> ### Author Response · Authors · 2025-11-25
> **Response 1/2 to Reviewer oJ2u**
>
> # **Dear Reviewer oJ2u,**
>
> We thank you for your thoughtful review and for highlighting the **clarity**, **novelty**, and **convincing nature** of our approach. We appreciate your constructive comments, which helped us further refine our work. Below, we address each of your points.
>
> ---
>
> **Q**: *What factors contribute to the degradation of composability in deeper networks? This issue requires further investigation and discussion to understand its implications.*
>
> **RE**: To shed light on this, we have conducted additional experiments with **ResNet18** on MNIST, FMNIST and **CIFAR-10** and with **VGG11** on FMNIST, which complement those of Deep MLP and CNN on simpler datasets.
> For deeper models, the **negative effects of weight-space merging** or wrong isolation are propagated and increased throughout the whole architecture, in a **cascade effect**. This might be alleviated using **BatchNorm**, as suggested by our good results on simple datasets using ResNet18 and by the comparison between VGG and VGG with BatchNorm (Figure 4). In particular, this latter clearly outperforms the standard one.
> On more complex datasets, the use of BatchNorm is not sufficient, suggesting that lower performances are to be imputed to the merging procedure. In fact, for ResNet18 on CIFAR-10,  aggregation in **weight-space** seems disruptive, while it is not in **logit-space** (new methodology introduced following Reviewer gGJL’s suggestion). This suggests that MaxEnt is able to isolate **task-specific functional modules** (otherwise logit-based merge would fail as well), but that they need to be carefully aggregated.
>
> ---
>
> **Q**: *Why does iterative magnitude pruning show greater effectiveness in deeper networks compared to shallow MLPs? Additional insights would be valuable.*
>
> **RE**: As discussed in the previous response, composability of deeper models requires additional care when done through the **weight-space merge**. Thus it is crucial to properly isolate the submodules before composing them through sum: small errors can propagate through the aforementioned **cascade effect**. It is clear that IMP helps in weight-space composability by the comparison between Shallow and Deep MLPs (See new Figure 26, 27, 28 and 29 for easier visualization).
> However, this is not always the case for deeper models such as **ResNet** and **VGG**. In these scenarios, as previously discussed, the negative effects can be conveniently overcome using the **BatchNorm** layer and/or the merge in **logit-space**. We believe that studying the best **merging techniques for submodules** is a fascinating and still underexplored direction that we are excited to investigate in future works.
>
> ---
>
> **Q**: *How does this work relate to broader research trends addressing increasingly complex tasks beyond classification?*
>
> **RE**: We appreciate the Reviewer’s insightful comment. Our current work indeed focuses on classification tasks. To the best of our knowledge, training **composable, isolated task-specifc submodules/circuits** is an interesting direction that hasn’t been explored enough in the existing literature. The two lines of work that are most closely related to ours are (i) **post-hoc circuit analysis** and (ii) canonical **weight-space model merging**. However, both are fundamentally different from our setting: we train the circuits from scratch and explicitly aim to merge circuits that are **functionally different**, whereas weight-space model merging typically considers models that solve the same task and differ mainly in parameter space.
>
> Moreover, the foundational works in both of these related areas, as well as a substantial fraction of the broader literature, primarily focus on **classification benchmarks on standard computer vision datasets**. For instance, see [1] as an example of post-hoc circuit analysis. In this sense, our experimental setup is well aligned with prevailing practice in neighboring domains.
>
> We agree that extending our approach beyond classification (e.g., to text classification, sequence modeling, or more general prediction tasks) is an important and interesting direction. At this stage, we do not yet have a definitive answer regarding how well our method generalizes to non-classification tasks. We view this as a promising avenue for future work, and, given that this is, to our knowledge, the first study in this new setting of training and merging composable circuits, we believe it is reasonable for the initial exploration to adopt assumptions similar to those made in closely related literature.
>
> **[1]** *Olah, et al., "Zoom In: An Introduction to Circuits", Distill, 2020.*
>
> ---

---

> ### Author Response · Authors · 2025-11-25
> **Response 2/2 to Reviewer oJ2u**
>
> ## Recap
>
> In summary, we are grateful for your comments. Our results indicate that:
> - **Composability** becomes more challenging in deeper networks for **cascade effects** that can be alleviated through **BatchNormalization**.
> - **IMP** is beneficial to isolate the minimal circuits, reducing interference in **weight-space**. However, optimal performances can always be recovered through merge in the **logit space**.
> - Overall, the **MaxEnt** objective + **sparsity training** consistently isolates **task-specific submodules**.
>
> These findings help us clarify the scope of our method, confirming its **novelty** and **usefulness** for more general models and datasets both for compositionality and interpretability. We hope our response fully addresses the Reviewer’s concerns and we are happy to continue the discussion in case further concerns might arise.

---

### Official Review · Reviewer_g43A · 2025-11-01

**Soundness:** 3
**Presentation:** 3
**Contribution:** 3
**Rating:** 6
**Confidence:** 4

**Summary:**

The authors describe a method for composing independently trained sparse neural subnetworks into a single model without joint training or fine-tuning. Each subnetwork (“module”) is trained on a subset of classes called the rewarded set, using a modified loss based on the maximum-entropy principle: the model is encouraged to make confident predictions only for its rewarded classes and uniform (high-entropy) predictions for all others. Sparsity is induced through IMP to limit parameter overlap between modules. Once trained, these subnetworks (each specialized to different classes) are combined by simple weight summation, producing a merged model that can classify across all classes while preserving much of each specialist’s performance.

The authors empirically evaluate this approach on MNIST, Fashion-MNIST, and two tabular datasets, using small MLPs and a LeNet-style CNN. They report that modules trained with the MaxEnt objective achieve functional isolation, and that merged networks retain high per-class accuracy and high entropy on non-rewarded classes, indicating low interference. Mode-connectivity analyses further suggest that module parameters lie in nearby basins, explaining why their linear combination yields coherent behavior. The work claims that enforcing entropy separation, combined with sparsity, enables modular composition of neural networks in weight space.

**Strengths:**

- simple, interpretable composition mechanism, avoiding complex re-alignment or joint optimization.
- clear/principled loss function that explicitly enforces confident predictions on rewarded classes and uniform predictions elsewhere, promoting functional isolation
- demonstrates that iterative pruning improves mergeability, showing a plausible connection between sparsity and reduced interference across modules
- mode-connectivity analysis indicating that isolated modules occupy nearby low-loss basins, lending some theoretical (but limited) support for weight-space composition
- related to modular and compositional learning. In my opinion, this is an important and promising direction in DL these days.

**Weaknesses:**

Major issue: experiments are limited to simple datasets and small architectures (MNIST, Fashion-MNIST, shallow MLPs, LeNet)
- The scalability and effectivenes to realistic settings (e.g., CIFAR-10/100, ImageNet, Transformers) is unclear (probably it would not work in my opinion)

- no comparison to strong baselines such as model soups, Fisher-weighted merging etc
- The interaction between pruning and the MaxEnt loss is not disentangled. It is unclear which component drives functional isolation and mergeability
- Summation without alignment or normalization can be tricky to work
- the MaxEnt objective may suppress shared features and degrade generalization when classes share structure, limiting applicability beyond disjoint one-vs-all setups
- Claims of “modular structure” are not supported by structural analysis. There is no measurement of mask overlap, activation similarity, or connectivity between class modules
- The method lacks quantitative ablations and uncertainty metrics (e.g., calibration, leakage, loss barriers vs. number of merged modules).
- The writing and figure clarity varies a bunch. Some tables lack legends and experiment protocols (e.g., dataset reuse and merge order) are not sufficiently described.

**Questions:**

-  Please, if there is enough time in the rebuttal phase, compare the MaxEnt loss directly to label smoothing and confidence-penalty objectives, tuned so that non-rewarded classes approach a uniform distribution. This is essential to verify that MaxEnt provides more than a re-parameterization of existing regularization methods
- Conduct a 2×2 ablation (MaxEnt vs. standard cross-entropy) × (with vs. without IMP) while keeping final sparsity fixed. This would clarify whether isolation arises mainly from the loss, from pruning, or from their interaction.
- can the method compose modules on datasets with shared features, such as CIFAR-10 or CIFAR-100? Can you have some results on a small ResNet at similar sparsity?
- As you know weight summation can amplify parameter norms. Did you test weight rescaling, re-centering, etc before merging? Please report any normalization strategy (or justify why none is needed?)
- Quantify inter-module overlap. for example Jaccard similarity of masks, cosine similarity of activations, or participation coefficients. This will help support your claim of isolated modules.
- in the real world we often have overlapping or hierarchical class groups. Can this approach handle partially overlapping rewarded sets without large interference?
- How would the method extend to architectures with BatchNorm or LayerNorm? Did you attempt merging such models?
- beyond non-rewarded entropy, report per-class confusion or leakage rates, calibration error and loss-barrier heights as quantitative measures of interference after merging
- I would love to see some comparisons against model soups, Fisher-weighted merging, and Git Re-Basin under the same initialization and pruning regime to position your approach within current weight-space composition literature.
- Add legends and unit definitions in Table 1. specify the dataset split and permutation protocol in complete-merge experiments. plz give implementation details for IMP (rounds, pruning ratios, weight resets) and temperature settings for all architectures

---

> ### Author Response · Authors · 2025-11-25
> **Response 1/3 to Reviewer g34A**
>
> # **Dear Reviewer g43A,**
> We thank you for your thorough and insightful feedback, and for recognizing the **clarity** and **potential** of our approach. We greatly appreciate your detailed comments, which already helped us strengthen the paper both experimentally and conceptually. Below, we address each main concern.
>
> ---
>
> **Q**: *Please, if there is enough time in the rebuttal phase, compare the MaxEnt loss directly to label smoothing and confidence-penalty objectives, tuned so that non-rewarded classes approach a uniform distribution. This is essential to verify that MaxEnt provides more than a re-parameterization of existing regularization methods*
>
> **RE**:
> We thank the Reviewer for this excellent suggestion. We have now implemented both **label-smoothing** and **confidence-penalty** baselines, as they can bring valuable comparisons with respect to our proposed methodology. Results, which can be found in Table 9 (in the Appendix), indicate that both baselines perform significantly worse than our MaxEnt objective, suggesting that label-smoothing and confidence-penalty objectives do **not** work well for isolating task-specific submodules, while our proposed methodology is specifically designed to do so. The **asymmetric treatment** of MaxEnt (it differentiates between rewarded and non-rewarded sets, applying strong regularization only to the latter) appears essential for isolating class-specific subnetworks and we couldn’t find it in previous work.
>
> ---
>
> **Q**: *Conduct a 2×2 ablation (MaxEnt vs. standard cross-entropy) × (with vs. without IMP) while keeping final sparsity fixed. This would clarify whether isolation arises mainly from the loss, from pruning, or from their interaction.*
>
> **RE**:
> As shown in Appendix from Table 4 to Table 9 (and more easily visualizable from Figure 26 to Figure 30), **MaxEnt alone already improves module isolation**, but the combination MaxEnt + sparsity training consistently yields the best merging behaviour for simpler datasets and architectures. In particular, in this scenario results show that: MaxEnt allows better merged accuracy than CrossEntropy even without pruning, confirming that the loss itself encourages isolation; **IMP always improves CrossEntropy**, and **MaxEnt + IMP is the strongest combination**.
> On more complex models (i.e. ResNet) and more complex datasets (i.e. CIFAR-10) the situation changes since **simple weight-space merge** does not reach optimal performance. We instead resolve to a **logit-space merge** (suggested by Reviewer gGJL). In such a regime, the pruning strategy is still fundamental to identify a quite small (wrt to the full model size) circuit.
> The focus of our present work is to show that **MaxEnt + sparsity training reliably identifies class-specific circuits**, which can be composed into a generalist model, either through weight-space merge (simple settings) or logit-space merge (complex settings). This provides a principled and practically useful tool for studying and composing circuits.
>
> ---
>
> **Q**: *Can the method compose modules on datasets with shared features, such as CIFAR-10 or CIFAR-100? Can you have some results on a small ResNet at similar sparsity?*
>
> **RE**:
> We have added an extensive set of experiments (more than **10,000 runs**) on **ResNet** across MNIST, FMNIST and **CIFAR-10** and **VGG** on FMNIST. Following Reviewer gGJL’s suggestion, we now analyse not only the original **weight-space** merge, but also a new **logit-space** merge.
> Overall, the results reveal that: for **simple datasets** (MNIST, FMNIST) and **small models** and for **ResNet**, the modules produced by our pipeline compose effectively in **weight space**, confirming that the identified modules behave as independent functional submodules whose weights can be added directly; for more complex datasets such as CIFAR-10, weight-space summation is no longer sufficient, whereas **logit-space summation remains highly accurate**. This holds also in the complete merge setting. The results are displayed in **Table 2 (new)**, **Figure 4 (new)** and **Figure 5 (new)**.
> Overall, our findings consistently support the core claim of the paper: **MaxEnt + sparsity training reliably identifies class-specific functional modules**, and these modules are composable by design, either directly in weight space for simple datasets, or in logit space for complex ones.
>
> ---

---

> > ### Author Response · Authors · 2025-11-25
> > **Response 2/3 to Reviewer g43A**
> >
> > **Q**: *As you know weight summation can amplify parameter norms. Did you test weight rescaling, re-centering, etc before merging? Please report any normalization strategy (or justify why none is needed?)*
> >
> > **RE**: We have not extensively tried any normalization technique, as it did not seem to affect the generalization particularly during preliminary experiments. Further investigation on the merging phase, alongside normalization studies on the weights are in plan for future works.
> > Nevertheless, we have conducted some deeper investigations on the role of the **BatchNorm**, that makes the network **scale-invariant** to the norm of the weights, while producing a significantly smoother optimization landscape [1]; we show in Figure 4 that it **improves the weight-space mergeability** of our modules for complex models and datasets. This is an interesting behavior that requires further investigation that is planned for future works.
> > We would like to mention that our work focuses on proposing a whole new **training pipeline** (MaxEnt + sparsity training) for finding composable modules, rather than optimizing each of its components, such as pruning techniques or merging strategies.
> >
> > **[1]** *Santurkar, Shibani, et al. "How does batch normalization help optimization?." Advances in neural information processing systems 31 (2018).*
> >
> > ---
> >
> > **Q**: *Quantify inter-module overlap. for example Jaccard similarity of masks, cosine similarity of activations, or participation coefficients. This will help support your claim of isolated modules.*
> >
> > **RE**: Unfortunately, for this first part of the rebuttal, we had to focus on other aspects to accommodate as much as possible all the comments raised by all the Reviewers. This surely remains a very interesting point. Thus, for the remainder of the rebuttal period, we will try to provide some of these metrics, like Mask overlap (**Jaccard index**) between both MaxEnt and Cross-Entropy subnetworks (with and without IMP).
> >
> > ---
> >
> > **Q**: *In the real world we often have overlapping or hierarchical class groups. Can this approach handle partially overlapping rewarded sets without large interference?*
> >
> > **RE**: We appreciate this question. While our datasets (e.g., FMNIST and **CIFAR-10**) contain partially overlapping visual features, we have not yet explicitly tested hierarchical class groups or multi-label settings. This is an interesting direction for future work, as our approach could naturally extend to hierarchical modularization by training subnetworks on nested rewarded sets.
> >
> > ---
> >
> > **Q**: *How would the method extend to architectures with BatchNorm or LayerNorm? Did you attempt merging such models?*
> >
> > **RE**: We thank the Reviewer for this insightful question, since BatchNorm (and its variants) is quite widespread in modern NNs. We confirm that **our method works with BatchNorm** layers as mentioned in the previous answers and as shown by the experiments (Table 7, Table 8 and Figure 4) that we have added to this revision. When merging with BatchNorm, it suffices to **re-estimate the running statistics** on data from the union of rewarded classes, keeping all learned affine parameters frozen. This simple step enables successful **weight-space merges** in ResNet architectures on FMNIST and MNIST. Clearly, no issues are present when performing **logits-based merging** when using MaxEnt (see very good results for ResNet on CIFAR-10 (Figure 4) and VGG on FMNIST (Table 7)). LayerNorm merges should be analogous, but we keep this investigation for future works.
> >
> > ---

---

> > > ### Author Response · Authors · 2025-11-25
> > > **Response 3/3 to Reviewer g43A**
> > >
> > > **Q**: *I would love to see some comparisons against model soups, Fisher-weighted merging, and Git Re-Basin under the same initialization and pruning regime to position your approach within current weight-space composition literature.*
> > >
> > > **RE**:
> > > Existing weight-space merging methods (e.g., Git Re-Basin, Fisher weighted averaging, model soups) are designed to merge models trained on **similar tasks and/or datasets**. These methods typically assume some functional similarity between the models to ensure meaningful merging. This is **not** the case for our class-specific modules. In our preliminary experiments on CIFAR-10, these techniques did **not** seem to recover the accuracy of the monolithic model, because the assumptions they rely on (e.g., shared training objectives) do not hold in our setting. This highlights a broader point: **merging class-specific submodules requires composition rules fundamentally different from those used for standard methods**.
> > >
> > > Currently, we rely on the most basic merging strategies. However, as pointed out, designing smarter ways of merging might lower the interference during merging in weight-space, and this is, in fact, a very valuable future direction. We would like to mention that our work focuses on proposing a whole new **training pipeline** (MaxEnt + sparsity training) for finding composable modules, rather than optimizing each of its components, such as pruning techniques or merging strategies. Nonetheless, we thank the Reviewer for these insights and, if we have time, we can present further merging strategies.
> > >
> > > ---
> > >
> > > **Q**: *Add legends and unit definitions in Table 1. specify the dataset split and permutation protocol in complete-merge experiments. plz give implementation details for IMP (rounds, pruning ratios, weight resets) and temperature settings for all architectures*
> > >
> > > **RE**:
> > > We have improved the presentation of Table 1, following the Reviewer’s suggestions. Appendix C describes our IMP procedure (rounds, ratios, weight rewinding steps, and temperature parameters).
> > >
> > > ---
> > >
> > > ## Recap
> > > We are grateful for your constructive feedback. In summary:
> > > - Label-smoothing and confidence-penalty baselines **underperform** compared to MaxEnt.
> > > - The loss and pruning contributions are disentangled as follows: **MaxEnt only trains the relevant circuits** whilst **IMP is useful for not carrying useless weights in the modules**, thus either obtaining the same accuracy of the final merged models or improving it, when keeping useless weights would have caused interference between modules.
> > > - ResNet and VGG results confirm that **our method scales** with minimal adaptation (BN re-estimation suffices for simple datasets) also on more complex datasets (CIFAR-10).
> > > - We improved the presentation following the Reviewer’s suggestions.
> > >
> > > These additions reinforce that core message of the paper that our **MaxEnt + sparsity training uncovers functional task-specific submodules, both in simple and complex scenarios**, representing a novel and useful tool. We once again thank the Reviewer for the precious feedback, hoping to be able to continue this fruitful discussion.

---

> ### Author Response · Authors · 2025-12-03
> **Second Response to Reviewer g43A**
>
> # **Dear Reviewer g43A,**
>
> We wanted to follow up with a study on the limits of compositionality that the Reviewer requested. In particular, the Reviewer asked whether our approach can handle datasets with richer intra-class variability and greater overlap in visual features, such as CIFAR-100.
>
> ## **CIFAR-100**
>
> We further evaluate **ResNet18 trained with MaxEnt on CIFAR-100**, using a **cardinality of 50** (i.e., training 2 sub-modules with 50 disjoint classes each and then merging them together) and a **pruning ratio of 0.6** whenever pruning is applied.
>
> This experiment is designed to probe the **limits of compositionality when the number of classes becomes large**. We again find that **logit-space merging consistently outperforms weight-space merging**, both with and without pruning. Indeed, the **pairwise logit merge** achieves performance only slightly below the CrossEntropy baseline trained in the standard way on the full dataset, while the **weight merge** degrades.
>
> We also conducted a final experiment which aims to be a **stress test** useful to detect whether we can merge more than 10 class-specific sub-modules together or if there is a bottleneck. To such an extent, we tested, for CIFAR-100, the **complete merge experiment** (i.e., merging 100 sub-modules each specialized on a single class). This is the most challenging scenario. We observed no sudden drop in performance each time we added a new model (till reaching 100 steps), suggesting that our approach can generalize to larger settings. The rewarded accuracy decreases gradually from nearly **100% (initial modules)** to roughly **50% at the end of the merge sequence**. Importantly, the merged model maintains **above 60% rewarded accuracy for the first ~50 merged modules** before eventually decaying. Of course, as the final merged 100-class model reached 0.533 accuracy compared to the optimal case of 0.661 achieved by the CrossEntropy baseline (trained in the standard way on the full dataset), this suggests, on one hand, that composition becomes increasingly difficult when merging dozens of modules, and on the other hand, that **MaxEnt successfully isolates class-specific functionality even in a high-cardinality setting**.
>
> **Pairwise Merge Results at cardinality 50**
>
> | IMP | Merge | Rewarded Accuracy |
> | :---- | :---- | :---- |
> | No | Weights | 0.107 ± 0.011 |
> | No | Logits | 0.644 ± 0.002 |
> | Yes | Weights | 0.020 ± 0.002 |
> | Yes | Logits | 0.521 ± 0.005 |
>
> **Complete Merge Results**
> *(trained on a single seed due to computational cost)*
>
> | IMP | Step | Merge | Accuracy |
> | :---- | :---- | :---- | :---- |
> | No | 1 | Weights | 0.992 |
> | No | 1 | Logits | 0.992 |
> | Yes | 1 | Weights | 0.985 |
> | Yes | 1 | Logits | 0.985 |
> | No | 25 | Weights | 0.046 |
> | No | 25 | Logits | 0.654 |
> | Yes | 25 | Weights | 0.021 |
> | Yes | 25 | Logits | 0.671 |
> | No | 50 | Weights | 0.019 |
> | No | 50 | Logits | 0.581	 |
> | Yes | 50 | Weights | 0.012 |
> | Yes | 50 | Logits | 0.606	 |
> | No | 75 | Weights | 0.011	 |
> | No | 75 | Logits | 0.536	 |
> | Yes | 75 | Weights | 0.009 |
> | Yes | 75 | Logits | 0.560 |
> | No | 100 | Weights | 0.009  |
> | No | 100 | Logits | 0.509 |
> | Yes | 100 | Weights | 0.009  |
> | Yes | 100 | Logits | 0.533 |
>
> **Baseline (trained with CrossEntropy on full CIFAR-100): 0.661 ± 0.006**
>
> Together with the extensive CIFAR-10, Imagenette, and text-classification experiments provided elsewhere in the rebuttal (see Figures 4, 5, 26-30, Table 2, updated with logit-merge, and Tables 6, 7, 8, and the new answer to the Reviewer EE1e), these CIFAR-100 results further strengthen and confirms our central claim: **MaxEnt \+ sparsity training reliably identifies class-specific submodules and supports principled composition in both simple and challenging regimes**. We hope these additions fully address the Reviewer’s question regarding modularity and composability on complex, shared-feature datasets.

---

### Author Response · Authors · 2025-11-25
**General Response 1/2**

# **Dear Reviewers, ACs, and PCs,**

We **sincerely thank** all Reviewers for their thoughtful, constructive, and encouraging feedback.
We are delighted that several Reviewers highlighted the **clarity, novelty, and elegance** of our training objective, and that **the experiments convinced them that our method**, which consists of a new loss function, MaxEnt, inspired by the max entropy principle, paired with iterative magnitude pruning (IMP), **indeed isolates class-specific submodules/circuits and enables their composition without joint training**.

Over the past days we have carried out a substantial number of new experiments, **more than 10,000 additional runs**, specifically aimed at addressing the Reviewers’ main concerns regarding **scalability, motivation of IMP, comparisons with baselines, and structural analysis** (first revision).

To fully address the concerns raised across reviews, we conducted additional experiments after the first revision. Nonetheless, to make the evaluation process easier for the AC, we decided to update this same high-level summary, while per-reviewer responses contain detailed descriptions and tables, as replies added to those provided at the first round.


Below we summarize the key findings and improvements to the paper.

---

1. **Logit-level merging (suggested by gGJL)**

Following Reviewer gGJL’s excellent insight, we implemented **logit merge**, finding that: in **simple settings (simple model + simple dataset)**, weight-merge is similar to logit-merge; in more **complex settings (complex model + complex dataset)**, logit-merge is significantly stronger, recovering better accuracies even when weight-space merging underperforms.
This strongly supports the Reviewer’s intuition and provides a mechanistic explanation: **our method identifies circuits**, but deeper architectures require a different composition rule.

---

2. **New large-scale experimental section (ResNet18, VGG11, CIFAR-10, CIFAR-100, Imagenette)**

Across most Reviewers, the primary concern was whether our **MaxEnt + sparsity training** pipeline would generalize beyond models and datasets. In response, we added extensive experiments on **ResNet18, VGG11 (with and without BatchNorm)**, as new models, and **CIFAR-10, CIFAR-100, and Imagenette**, as new datasets. These experiments lead to a clear picture: **task-specific submodule identification** generalizes extremely well across models and datasets. For all architectures, including ResNet and VGG, the **MaxEnt loss consistently isolates the functional circuits** of the rewarded classes, and **IMP reliably removes irrelevant weights**. In addition, we show that submodule **composition in weight space**, depends on dataset/architecture complexity. For **simple datasets** (MNIST, FMNIST), and even some complex models on simple datasets (ResNet-18 on FMNIST), **simple weight summation works** (once BatchNorm statistics are re-estimated). For **complex datasets + complex models** (ResNet-18 on CIFAR-10, CIFAR-100, Imagenette), naive weight-sum merging is insufficient, but **logit-level merging** recovers most of the monolithic model’s performance, confirming that our method still discovers the correct circuits even when the composition rule is not trivial. This holds also in the case of **complete merge** (see the new **Figure 5**).
These results address the core **scalability concern** raised by Reviewers g43A and EE1e.

We also noticed that on Imagenette and CIFAR-100 merged models were on par or even better than the models trained with CrossEntropy in the standard way on the full dataset (see Second Responses to EE1e and g43A).

---

3. **BatchNorm role in merging with deeper architectures (requested by g43A)**

The added experiments shed light on the applicability of our framework to all architectures using **BatchNorm**. Weight-space merging models with BN works by simply **re-estimating running statistics** on the union of the rewarded classes, with frozen weights. This enabled **successful weight-space merging** in ResNet architectures on FMNIST and MNIST.
We also show that BatchNorm can alleviate how negative effects of weight-space merging or wrong isolation **propagate through deep models**. This is supported by strong results on simple datasets using ResNet18 (that employs BatchNorm) even by just using weight-space merge and by the comparison between **VGG and VGG with BatchNorm** (Figure 4). In particular, this latter clearly outperforms the standard one.

---

4. **Label smoothing and confidence-penalty baselines (requested by Reviewer g43A)**

We implemented both baselines (in addition to the CrossEntropy and Quasi-MaxEnt ones) and **they both perform significantly worse than MaxEnt** (see Table 9), showing that MaxEnt is fundamentally different from these regularizers.

---

---

> ### Author Response · Authors · 2025-11-25
> **General Response 2/2**
>
> 5. **Impact and role of IMP**
>
> As shown in Appendix from Table 4 to Table 9 (and more easily visualizable from Figure 26 to Figure 30), **MaxEnt alone already improves module isolation**, but the combination **MaxEnt + sparsity training** consistently yields **the best merging behaviour for simpler datasets and architectures**.
> On **more complex models** (i.e. ResNet) and **more complex datasets** (i.e. CIFAR-10) the situation changes since simple weight-space merge does not reach optimal performance. We instead resolve to a **logit-space merge** (suggested by Reviewer gGJL). In such a regime, the pruning strategy is still fundamental to identify a quite small (wrt to the full model size) circuit.
>
> Overall, results show that **MaxEnt + sparsity training reliably identifies class-specific circuits,  which can be composed into a generalist model**, either through weight-space merge (simple settings) or logit-space merge (complex settings). This provides a principled and practically useful tool for studying and composing circuits.
>
> ---
>
> 6. **Practical usage of our method (requested by EE1e)**
>
> We now include a dedicated discussion (see Section 5) describing how our framework enables:
> - **Selective unlearning by design**,
> - **Federated training of class-specific modules**,
> - Principled circuit extraction for **mechanistic interpretability**,
> - Adoption of costly SOTA **formal verification techniques**, by the decomposition of the network into modules.
>
> ---
>
> 7. **Generalization beyond vision (requested by EE1e)**
>
> To address concerns from Reviewer EE1e, we added new experiments on **text classification** (IMDB, 20NG). Across both shallow and deep MLPs, MaxEnt modules merged in logit-space achieve performance comparable to or exceeding models trained with CrossEntropy in the standard way on the full dataset, demonstrating that the method transfers beyond image classification.
>
> ---
>
> A more detailed discussion of each point is provided in the **per-reviewer responses**, while the Recap section distills the main takeaways of the entire rebuttal.
>
> We hope this consolidated summary, together with the extensive new experiments and clarifications, makes it straightforward for the AC to assess the improvements and confirm that the key concerns raised by the Reviewers have been thoroughly addressed.
>
> ---
>
> ## Recap
>
> Across all Reviewers, several positive aspects were highlighted:
> - the **clarity of the approach** (Reviewers g43A and oJ2u) and the **excellent presentation** (Reviewer gGJL),
> - **the interesting topic** (Reviewer EE1e),
> - **the potential** (Reviewer g43A) and **novelty** of the approach (Reviewer oJ2u),
> - as well as the **“clever” and “elegant” solution** that we proposed (Reviewer gGJL),
> - but more importantly the **convincing empirical demonstration** that MaxEnt subnetworks behave as intended (Reviewers gGJL and oJ2u).
>
> Our **new experimental results** significantly strengthen the paper, showing that **the same simple pipeline** (MaxEnt + sparsity training):
> - **finds class-specific submodules in all models (MLPs, CNNs, VGG, ResNet)**,
> - composes them through **weight summation** in **simple** regimes (Yeast, HAR, MNIST, FMNIST),
> - still **exposes the correct circuits in complex regimes**, where **logit merging** is highly effective (CIFAR-10, CIFAR-100, Imagenette),
> - and **generalizes beyond vision**, where logit-merged modules achieve competitive performance on text classification tasks (IMDB, 20NG).
>
>
> We are deeply grateful for the Reviewers’ feedback and for the opportunity to greatly improve our work. We believe that the new experiments (**more than 10,000 runs**), **clarifications**, and **analyses have responded to most, if not all, of the Reviewers’ concerns**. We regret the unusual circumstances of this review cycle, as we would have welcomed a longer exchange, but we are glad that your feedback enabled us to substantially strengthen the paper.
>
>
> **Thank you** for your time, insights, and constructive engagement.

---

### Author Response · Authors · 2025-12-04
**Final Comment**

We **thank** once more the Reviewers for the feedback. We just wanted to clarify that the results from the **first revision round are already included in the revised PDF**, while the new results from this **second round are currently only on OpenReview** (added as new replies) due to timing; these **will be integrated into the paper**, and fully in the camera-ready version (if the paper will be accepted).

We have also updated the **General Response** (below), which provides a broad description of what occurred in the rebuttal, along with a brief **Recap** highlighting the most important points.

**Thank you** again for your time.

---

### Meta-Review · Area_Chair_3kK4 · 2026-01-06

**Summary:**

This paper presents a clean and principled approach for training class-specialized sparse subnetworks that behave like reusable modules and can be composed without joint training. The key idea—a maximum-entropy / KL-based objective combined with iterative magnitude pruning that makes each module confident on its rewarded classes and deliberately uncertain on all others—is elegant, easy to interpret, and supported by clear presentation and convincing evidence that the method isolates functional modules and enables compositionality.

The initial concerns were mainly about scalability beyond simple datasets and architectures, the specific role of IMP, and whether weight-summation merging would hold in realistic settings. The rebuttal substantially strengthens the paper by adding extensive new experiments on ResNet/VGG and harder datasets (CIFAR-10/100, Imagenette), plus text classification (IMDB, 20NG), showing the module-identification pipeline generalizes beyond the original small-scale setting. During the rebuttal, the authors introduced logit-level merging, which helps clarify that weight-space merging works well in simpler regimes, while logit-level merging is consistently strong in more complex architectures and datasets, often recovering most of the monolithic baseline and in some cases matching it. They also add comparisons to label smoothing and confidence penalty, which underperform MaxEnt for isolating modules, suggesting the method is not a repackaging of standard regularizers. The rebuttal also clarifies the role of BatchNorm and provides a coherent explanation for the failure modes of weight-space merging in deep models.

Overall, this is a strong contribution to modular and compositional learning, and it opens an interesting path toward neural network modules trained by design rather than extracted post hoc. The approach is simple, well-motivated, and now supported by substantially broader experimental validation. While additional structural analyses and broader merging baselines would be valuable future work, the revised evidence base addresses the central concerns about scope and robustness. I therefore recommend acceptance.

**Reviewer Concerns:**

Reviewer g43A: Major concerns about scalability to deeper architectures and more realistic datasets, disentangling the role of MaxEnt vs. IMP, and comparisons to alternative regularizers were mostly addressed. The rebuttal adds extensive experiments on ResNet/VGG and CIFAR-10/100/Imagenette, includes label smoothing and confidence-penalty baselines, and provides clearer ablations showing the complementary roles of the loss and pruning. Concerns about weight-space merging in complex settings are clarified via the introduction of logit-level merging. More detailed structural analyses of module overlap and broader comparisons to sophisticated weight-merging methods remain open but are no longer blocking.

Reviewer gGJL: Most concerns were addressed. In particular, the introduction of logit-level merging directly responds to questions about composition without relying on weight-space mode connectivity and clarifies the relationship between weight merging and logit summation. Presentation issues and metric clarifications were also resolved. Suggestions regarding neuron alignment and downstream applications were acknowledged but remain future work.

Reviewer oJ2u: Concerns about compositionality degradation in deeper networks were addressed through new ResNet/VGG experiments and the explanation of cascade effects under weight-space merging, as well as mitigation via BatchNorm and logit-level merging. Questions about broader relevance beyond small-scale vision tasks were partially addressed through additional experiments and discussion, though extensions beyond classification remain future work.

Reviewer EE1e: Initial concerns about limited experimental scope, applicability to larger-scale datasets, and generalization beyond vision were addressed by new experiments on Imagenette and text classification benchmarks (IMDB, 20NG), as well as an expanded discussion of practical use cases. Extensions to very large-scale datasets (e.g., full ImageNet) and non-classification tasks beyond text remain open directions.

**Reviewer Scores:**

Reviewer g43A: My estimation is that the main concerns on scalability, ablations, and baselines were addressed; the assessment would likely remain positive with increased confidence.

Reviewer gGJL: I expect that the rebuttal’s incorporation of the reviewer’s suggestions and strengthened empirical evidence would leave a strong positive assessment unchanged.

Reviewer oJ2u: It seems likely that the additional experiments and clarifications mitigate the main concerns about depth and compositionality, shifting the assessment toward neutral to mildly positive, with some limitations remaining.

Reviewer EE1e: I estimate that the new results on larger-scale datasets and text classification address key scope concerns, leading to a more balanced assessment, while very large-scale settings remain open.

---

### Decision · Program_Chairs · 2026-01-26

Accept (Poster)